# Integrated multi-system Prediction via Equilibrium State Evaluation

## Abstract

This study presents a new paradigm of prediction, Equilibrium State Evaluation (ESE), which excels in multi-system prediction where systems interact with each other and every system needs its own prediction. Unlike mainstream prediction approaches, ESE views each system as an integral part under one structure and predicts all systems simultaneously in one go. It evaluates these systems' equilibrium state by analyzing the dynamics of their attributes in a holistic manner, instead of treating each system as an individual time series. The effectiveness of ESE is verified in synthetic and real world scenarios, in particular COVID-19 transmission, where each geographic region can be viewed as a system. So cases spreading across regions against the medical competency and demographic traits of these regions can be considered as an equilibrium problem rather than a time series problem. Extensive analysis and experiments show that ESE is linear in complexity and can be 10+ times faster than SOTA methods, yet achieving comparable or better prediction accuracy. More importantly, ESE can be integrated with these prediction methods to achieve both high accuracy and high speed, making it a powerful prediction mechanism, especially for scenarios that involve multiple systems. When the dimensionality of the multi-system increases, e.g. more systems joining, the advantages of ESE becomes even more apparent. [1].

## 1 Introduction

This study establishes a new prediction method to address tasks involving multiple systems that may interact with each other. While predicting individually for each system is possible, we enable a holistic approach based on the concept of equilibrium, so all of the systems can be predicted at once without any repetition. Equilibrium by nature is not for prediction but describes a state or condition in which all competing influences in a system remain relatively balanced Daskalakis et al. (2009). If the equilibrium no longer holds, the system will start to destabilize. For example, a system in thermal equilibrium will not be so when a hot object appears. A new equilibrium may emerge after going through certain changes and adjustments. By viewing systems as components of a super-system and evaluating the equilibrium state of the super-system, we can estimate the interactions between these systems. Hence the tendency of changes between systems can be revealed. Subsequently, the future states of all these systems in the integrated super-system could be predicted. Based on this hypothesis, we propose a multi-system prediction approach, Equilibrium State Evaluation (ESE).

Equilibrium can be easily multi-dimensional and multifaceted. So equilibrium-based prediction can also go beyond single-target prediction, unlike typical time-series approaches. For example, considering an epidemic that spreads across different regions of a big area, each region itself is a system. Regions do influence their adjacent regions and are also influenced by their neighbours. So the area can be viewed as a case of integrated multi-system. Predicting the state of each region can be achieved by predicting the whole area with only one run of our proposed ESE, instead of multiple prediction runs for individual regions. ESE consists of three major components: (1) Equilibrium State Estimation, (2) Equilibrium Index, to measure the deviation of the current state from equilibrium, and (3) Predictor, to forecast the states of all systems based on the deviation. Moreover, ESE can act alone but can also be integrated with other methods, enabling these conventional methods to perform multi-system prediction as well. In such cases, these methods predict the overall trend

---

[1]The source code and the full data sets are viewable at: https://anonymous.4open.science/r/ESE-6432

while ESE takes care of the distribution across different systems. **Contributions** are summarized into three folds as follows:

- We propose a new integrated multi-system prediction mechanism, ESE,based on the concept of equilibrium. Unlike current time series prediction approaches, ESE predicts all systems in one run. Due to its linear complexity, ESE is more advantageous compared to other approaches when the number of involved systems is large.
- We conduct extensive experiments across a range of synthetic and real-world COVID-19 data, with various input lengths, prediction distances and granularities, demonstrating that ESE is not just fast and accurate, but also flexible, compared to SOTA prediction methods.
- We integrate the proposed ESE with SOTA prediction methods so the integrated models can achieve multi-system prediction with higher accuracy when maintaining high speed.

The epidemic data highly represents the definitions of integrated multi-system (introduced at Section 2. Therefore, Our case study on real-world data uses COVID-19 Cucinotta & Vanelli (2020), a type of epidemic, that refers to infectious diseases that occur in specific groups of people or regions under common causes Dicker et al. (2006). Elderly and people with underlying diseases such as cardiovascular disease and diabetes are more susceptible to COVID-19 Bajaj et al. (2021); Emami et al. (2020); Javanmardi et al. (2020); Richardson et al. (2020). Hence demographic attributes are important in COVID-19 prediction. Susceptible affected recovered (SIR) EPIC model, a predictive method in epidemiology, has been used to simulate the spread of dengue fever disease Side & Noorani (2013) and COVID-19 Cooper et al. (2020). The spatial distribution of confirmed cases can be predicted by using population mobility data Jia et al. (2020), and by analysing the spatio-temporal trends and characteristics of the pandemic Huang et al. (2020); Rex et al. (2020). Common approaches for prediction of epidemic spreading are mainly based on time series or related methods Kumar et al. (2020); Perone (2020). In particular, ARIMA ArunKumar et al. (2021); Benvenuto et al. (2020) and LSTM/GRU Feng et al. (2022); Omran et al. (2021); Sah et al. (2022); Shahid et al. (2020) are the mainstream methods to forecast the number of new cases. A few variations have been proposed to improve accuracy, such as EVDHM-ARIMA, which was used to predict new daily cases of the COVID-19 pandemic in India, the United States, and Brazil Sharma et al. (2021). CNN-LSTM has been also applied to predict new cases and to analyze the status of medical resources availability Ketu & Mishra (2021).

## 2 EQUILIBRIUM AND INTEGRATED MULTI-SYSTEM

**Equilibrium** Equilibrium state is common in the real world, such as isostatic equilibrium in mechanics Hemingway & Matsuyama (2017) and homeostasis in biology Hegyi et al. (2012). Equilibrium also plays an essential role in economics, e.g. equilibria in the large market Cole & Tao (2016). For example, market competition can be predicted by the regression equilibrium Ben-Porat & Tennenholtz (2019). The equilibrium state of decision-making behaviours can be viewed as Nash equilibrium Farina & Sandholm (2021). Nagurney et al. utilize Nash equilibrium to analyze the impact of the pandemic on business competition and other socioeconomic activities Nagurney & Salarpour (2021). Bairagi et al. use equilibrium-based game theory to design the optimization scheme for social distancing to minimize the COVID-19 situation Bairagi et al. (2020). Equilibrium-based approaches have not yet been widely adopted in numerical analysis or prediction, possibly because these are not the intended purposes of equilibrium. It is however increasingly more valued in learning as it can effectively improve computational efficiency when facing complex operations. For example, equilibrium is used to solve the problem of decentralized learning in Markov games Foster et al. (2023). The deep equilibrium model, which incorporates the concept of equilibrium into deep learning, makes it possible to perform training and prediction without the need for increased memory, regardless of the network's effective "depth" Bai et al. (2019); Yang et al. (2023). This technique is also a success in computer vision Bai et al. (2022; 2020); Graf et al. (2022).

**Multi-system** Although no universal definition, multi-system is widely mentioned in literature from different fields, such as Pathology Haslak et al. (2021), Sociology Andersen & Geels (2023), and marketing Gilliland (2023). For this study, we define integrated multi-system as below and illustrated in Figure 1. It is particularly important to distinguish multi-system from similar-looking terms related to prediction, such as multi-target, multi-variate, multi-objective, and multi-compartment.

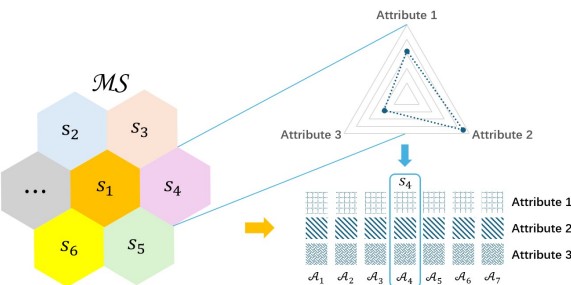

Figure 1: Illustration of a multi-system $\mathcal{MS}$, which contains systems $s_1, s_2, s_3, s_4, s_5, s_6$ and more. Each system $s_i$ contains three attributes, $Attribute\ 1$ to $Attribute\ 3$. The attribute set of system $s_4$ is denoted as $\mathcal{A}_4$. The task is to predict the future state of all systems in $\mathcal{MS}$ simultaneously.

**Definition 1:** An integrated multi-system $\mathcal{MS}$ contains $n$ systems as set $s_{1:n} = [s_1, \ldots, s_n],\ n \in \mathbb{N}^*;\ n \geq 2$, each $s_i$ is an individual system.

This definition describes an integrated multi-system consisting of 2 or more systems. Note, to simplify the representation, we use $s_i$ to denote the target variable of System $s_i$ as well. For example in the scenario of epidemic prediction of new cases, $s_i$ represents the daily new cases in System $s_i$, or region $i$. Similarly, we use $\mathcal{MS}$ to represent the entire multi-system, as well as the sum of target variables of all systems, so $\mathcal{MS} = \sum_{i=1}^{n} s_i\ n \geq 2$.

Each system can be viewed as an individual component, which is not independent of other systems. Systems do influence each other. A change in one system is the result of changes in other systems. Based on this, we can have Definition 2.

**Definition 2:** When multi-system $\mathcal{MS}$ is viewed in the entirety, the change in proportion in one system complements the total changes in proportion from other systems as expressed in Equation 1,

$$\triangle\gamma_i = -\sum_{j=1}^{n} \triangle\gamma_j\ ; n \neq i, \tag{1}$$

Equation 1 derives from zero-sum games Eatwell et al. (1989); Von Neumann & Morgenstern (2007) and describes multi-systems that are suitable for this study. In the equation, $\gamma_i$ is the proportion of system $s_i$ to $\mathcal{MS}$, as described in Equation 2:

$$\gamma_i = \frac{s_i}{\mathcal{MS}} \tag{2}$$

Definition 2 describes the relationship between each system in $\mathcal{MS}$, that is, the change in one system equals the total changes in other systems. Note the entire integrated multi-system should be complete. For example, if we consider a nation as an integrated multi-system $\mathcal{MS}$ with its states as the systems, no state shall be removed from $\mathcal{MS}$. Thereby constraints need to be imposed for ESE:

**Constraint 1:** When considering an integrated multi-system, no system or component is excluded. If Constraint 1 is satisfied, $\mathcal{MS} - \sum_{i=1}^{n} s_i = 0\ ; n \geq 2,$, so the constraint can be expressed as:

$$\sum_{i=1}^{n} \gamma_i = 1\ ; n \geq 2, \tag{3}$$

**Constraint 2:** In an integrated multi-system $\mathcal{MS}$, the sum of changes of all systems in proportion is zero, as expressed in Equation 4. This is derived from Equation 1, as $\triangle\gamma_i + \sum_{j=1}^{n} \triangle\gamma_j = 0\ ; n \neq i$.

$$\sum_{i=1}^{n} \triangle\gamma_i = 0\ ; n \geq 2, \tag{4}$$

With Constraints 1 and 2, changes in the systems can be presented in proportion to $\mathcal{MS}$. Any change in one system will impact all other systems. Therefore, our equilibrium based method can apply as detailed in Section 3.

**Definition 3:** Systems $(s_{1:n})$ of multi-system $\mathcal{MS}$ collectively determine the state of the entire super-system. The attribute set $\mathcal{A}$ of every system is identical. The values of $\mathcal{A}$ for a system in $\mathcal{MS}$ can be expressed as in Equation 5, where $\mathcal{A}_i$ is the set $\alpha_{i,1:m}$, and $m$ is the number of attributes in $(s_i)$.

$$\mathcal{A}_i = \{\alpha_{i,j} | j \in [1:m],\ m \in \mathbb{N}^*\}. \tag{5}$$

## 3 EQUILIBRIUM STATE

As described in the introduction section, our equilibrium state evaluation method (ESE) analyzes whether the multi-system is in a state of equilibrium. The core idea is similar to Nash equilibrium Kreps (1989), performing state evaluation by analyzing the internal competitive relationship between systems. ESE also relates to the concept of deep equilibrium model Bai et al. (2019) and image information transformation Xu & Song (2022). That is, the changes in the state of a multi-system can be obtained by studying the internal competitive relationship between "internal" systems. Thus we can evaluate the overall state based on equilibrium.

**Equilibrium Conditions**

$$\begin{aligned} u_1(\alpha_1^*, \alpha_2^*) &\geq u_1(\alpha_1, \alpha_2^*); \\ u_2(\alpha_1^*, \alpha_2^*) &\geq u_2(\alpha_1^*, \alpha_2), \end{aligned} \tag{6}$$

To evaluate the equilibrium state of multi-system, the relationships between the systems need to be analyzed. According to Nash equilibrium's basic payoff function, we can have Equation 6 Eatwell et al. (1989); Von Neumann & Morgenstern (2007). They are for a two-player game with a single decision-making point, e.g. betray or not. Function $u_i()$ represents the payoff of player $i$, and $\alpha_i$ represents the decision of player $i$. Note that $\alpha_1$ and $\alpha_2$ are in the same decision space. That is if $\alpha_1$ is a decision $\{yes, no\}$, $\alpha_2$ is also of decision $\{yes, no\}$. Further, $(\alpha_1^*, \alpha_2^*)$ are the decisions under Nash Equilibrium. Thus, we can obtain the conditions for being in an equilibrium state as below.

**Lemma 1:** The equilibrium state of the integrated multi-system $\mathcal{MS}$ means that all systems of $\mathcal{MS}$ reach their maximum benefit, or proportion in the system $\mathcal{MS}$, under mutual influence based on their attributes $\mathcal{A}$ (as the decisions in Nash equilibrium).

$$\begin{aligned} U_1(\mathcal{A}_1^*, \mathcal{A}_2^*, \dots, \mathcal{A}_n^*) &\geq U_1(\mathcal{A}_1, \mathcal{A}_2^* \dots, \mathcal{A}_n^*) \\ U_2(\mathcal{A}_1^*, \mathcal{A}_2^*, \dots, \mathcal{A}_n^*) &\geq U_2(\mathcal{A}_1^*, \mathcal{A}_2 \dots, \mathcal{A}_n^*) \\ &\vdots \\ U_n(\mathcal{A}_1^*, \mathcal{A}_2^*, \dots, \mathcal{A}_n^*) &\geq U_n(\mathcal{A}_1^*, \mathcal{A}_2^*, \dots, \mathcal{A}_n), \end{aligned} \tag{7}$$

In the case of multi-system with a group of attributes, we can define the equilibrium conditions of $\mathcal{MS}$ as in Equation 7. It is a generalization of Equation 6. As shown in Equation 7, changes in any attribute of any system will lead to changes in other systems. Based on Constraints 1 and 2 in Section 2, we can disregard the development of $\mathcal{MS}$ but only focus on the relationships between systems. Therefore, the payoff function $U()$ can be unified, as Equation 8[2]. In this way, internal variations within the system, such as interactions and feedback loops between systems, can be transformed into a distribution of proportions, significantly reducing the complexity. The trends of each $s_i$ can also be shown more clearly and consistently, regardless of the tendency of the $\mathcal{MS}$. The stronger the $s_i$, the greater the proportion. No matter how the $\mathcal{MS}$ develops, the proportions of the $s_{1:n}$ will not change if no change in attributes of $s_{1:n}$. Note, ESE does not require $\mathcal{MS}$ to reach a true equilibrium but estimates its equilibrium state under the assumption of zero-sum.

$$U(\mathcal{A}_1^*, \mathcal{A}_2^*, \dots, \mathcal{A}_n^*) \geq \cdots \geq U(\mathcal{A}_1, \mathcal{A}_2 \dots, \mathcal{A}_n). \tag{8}$$

Now the equilibrium state of an integrated multi-system can be evaluated by estimating the proportion of each system based on the attribute set $\mathcal{A}$ of these systems, as presented in Section 4.1.

---

[2]The deductive reasoning process of the payoff function is presented in **Appendix B**.

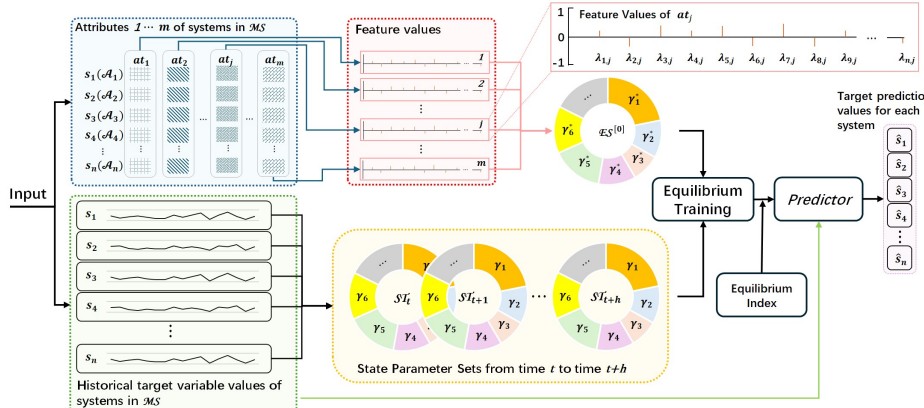

Figure 2: Illustration of the overall process of ESE. Multi-system $\mathcal{MS}$ consists of systems $s_1$ to $s_n$. The attribute set of $s_i$, $\mathcal{A}_i$ is converted to feature values $\lambda_{i,j}$. So the state parameter sets from $\mathcal{ST}_t$ to $\mathcal{ST}_{t+h}$ can be obtained, where $h$ is the input length. The feature values can generate the initial equilibrium state parameter set $\mathcal{ES}^{[0]}$. With all the above, followed by long-run equilibrium training, multiple predictions $\hat{s}_i$ can be performed through the predictor with equilibrium index.

## 4 EQUILIBRIUM STATE ESTIMATION METHODOLOGY

ESE is a dynamic framework with time-dependent interaction similar to time-varying models Gao et al. (2024). It consists of three main parts: (1) equilibrium state estimation, (2) equilibrium index, and (3) predictor, for estimating the equilibrium parameter set, evaluating the system's equilibrium level, and predicting the future states respectively. The overall process is illustrated in Figure 2.

### 4.1 ESTIMATING THE EQUILIBRIUM PARAMETER SET

**State Parameter Set**  Every system is always in a certain state $\mathcal{ST}$, which is the set of $\gamma_i$ introducing at Section 2, $\gamma_{1:n} = [\gamma_1, \ldots, \gamma_n]$, $n \in \mathbb{N}^*$.

**Equilibrium State Parameter Set**  This is the state parameter set when all systems are in an estimated equilibrium. In other words, the equilibrium state parameter set can be expressed as $\mathcal{ES} = \gamma^*_{1:n} = [\gamma^*_1, \ldots, \gamma^*_n]$, $n \in \mathbb{N}^*$. By comparing $\mathcal{ES}$ and $\mathcal{ST}$, we can determine whether the current state is in equilibrium or not, and hence estimate how far from the current state to the equilibrium state if the system is not yet in equilibrium. Similar to $\mathcal{ST}$, $\mathcal{ES}$ contains the same parts $\gamma^*_i$, which represents the corresponding $s_i$ when the multi-system $\mathcal{MS}$ reaches equilibrium. Hereby $\gamma^*_{1:n}$ is the mapping of the states of all systems $s_i$ of $\mathcal{MS}$.

$$\lambda_{i,j} = \frac{\alpha_{i,j} - \bar{at}_j}{upperbound(at_j) - lowerbound(at_j)}, \tag{9}$$

$$
\begin{array}{cccc}
s_1 & s_2 & \cdots & s_n \\
\lambda_{1,1} & \lambda_{2,1} & \cdots & \lambda_{n,1} & \sum_{i=1}^n \lambda_{i,1} = 0 \\
\lambda_{1,2} & \lambda_{2,2} & \cdots & \lambda_{n,2} & \sum_{i=1}^h \lambda_{i,2} = 0 \\
\vdots & \vdots & \ddots & \vdots & \vdots \\
\lambda_{1,m} & \lambda_{2,m} & \cdots & \lambda_{n,m} & \sum_{i=1}^n \lambda_{i,m} = 0 \\
\Downarrow & \Downarrow & \Downarrow & \Downarrow & \\
\Lambda_1 & \Lambda_2 & \cdots & \Lambda_n & \sum_{i=1}^n \Lambda_i = 0
\end{array}
\tag{10}
$$

In Figure 2, the columns in the blue box represent attributes from the attribute set $[at_1..at_m]$. The rows are the attributes set $\mathcal{A}$ for each system. We can extract the feature information $\lambda_{i,j}$ of attributes from all systems by using the feature measure of $\alpha_i$ as in Equation 9, where $\lambda_{i,j}$ is the feature information of $s_i$'s attribute $j$, $\bar{at}_j$ is the average of attribute $at_j$, $upperbound(at_j)$ and $lowerbound(at_j)$

are the upper bound and lower bound values of attribute $at_j$. The upper and lower bounds are usually the maximum and minimum values in $at_j$. Because the system is assumed to be a zero-sum system, the aggregation of all $\lambda_{i,j}$, e.g. the feature of a particular attribute of all systems, is always summed to zero, e.g. $\sum_{j=1}^{m} \lambda_{i,j} = 0$. That is applicable for all features/attributes, hence we can have Equation 10. In addition, the value of $\lambda_{i,j}$ would indicate the magnitude of the change in $s_i$ and whether the change is positive or negative.

$\Lambda_i$ in the Equation 10 is the total feature values of $s_i$. It is an intermediate parameter prior to the calculation of the initial equilibrium parameters $\mathcal{ES}^{[0]}$, as shown in Equation 11. Note, the total feature values of all $s_i$ is 0, which conforms to the zero-sum game theory.

$$\mathcal{ES}^{[0]} = Equi(\lambda_{1:m,1:n}; \Lambda_{1:m}) = \frac{1 + \Lambda_i}{n} = \frac{1 + \frac{1}{m}\sum_{j=1}^{m}\psi_j\lambda_{i,j}}{n} \tag{11}$$

In addition to the above parameters, a new parameter set ($\mathcal{L} = [l_1, \ldots, l_n]$) is needed to facilitate the subsequent predictor training. It is a $n$-element vector, which stores the progress of training. The elements of $\mathcal{L}$ map to the elements of $\mathcal{ES}$. The training process of $\mathcal{ES}$, e.g. the "Equilibrium Training" block of Fig. 2, is detailed in Algorithm 1. The progression of $\mathcal{L}$ over time, e.g. from $t$ to $t+1$ can be described in the following Equation 12.

$$\mathcal{L}^{[t+1]} = f(\mathcal{L}^{[t]}; \gamma_{1:n}^{*[t]}; \gamma_{1:n}^{[t]}), \ t \in [0:L-1], \mathcal{L}^{[0]} = 0 \tag{12}$$

---

**Algorithm 1** The equilibrium training process of $\mathcal{ES}$

---

**Input:** $\mathcal{ST}_{1:t} = \gamma_{1:n,1:t}$, $\mathcal{ES}^{[0]}$
**Output:** $\mathcal{ES}_{output}$
1: $\mathcal{ES}^{[0]}$ as $\mathcal{ES}_{output}$; $i = 0$; $\mathcal{L}_0 = np.ones(n)$ /* Initialization, all $n$ elements of $\mathcal{L}$ set to 1. */
2: **while** $\mathcal{ST}_{1:t} = \gamma_{1:n,1:t}$ and $\mathcal{ES}_{output} = \gamma_{output,1:n}^*$ are not cointegrated **do**
3:     **for** $s$ in [1, t] **do**
4:         $\mathcal{L}_s = (\mathcal{ES}_s^{[i]} - \mathcal{ST}_s + \mathcal{L}_{s-1})/2$
5:     **end for**
6:     $\mathcal{ES}_{output} = \mathcal{ES}_t^{[i]} - (\mathcal{L}_{output}/2)$
7:     i+=1
8: **end while**

---

Cointegration in Algorithm 1 is often used in statistics to test long-run equilibrium Abadir (2004); Enders & Siklos (2001). In this study, we assume the existence of long-run equilibrium, if $\mathcal{ST}_{1:t}$ and $\mathcal{ES}_{output}$ are cointegrated. Otherwise, $\mathcal{ES}_{output}$ will keep converging until cointegration is reached. The cointegration equations are in Equation 13, where $\Phi$ are parameters used for Ordinary Least Square (OLS) estimation. When $E_t$ are sequences of $I(0)$s, e.g. no differences, there is a cointegration relationship between $\mathcal{ST}_t$ and $\mathcal{ES}_t$. The convergence analysis of ESE training can be found in in **Appendix C**, which shows the progression of p-values from the cointegration test at each training step. Once the p-value reaches 0.05, the existence of long-run equilibrium is considered true, hence the training terminates. Then ESE prediction can proceed as shown below.

$$\mathcal{ST}_t = \Phi_0 + \Phi_1\mathcal{ES}_t + E_t$$
$$E_t = \mathcal{ST}_t - \hat{\mathcal{ST}}_t = \mathcal{ST}_t - \hat{\Phi}_0 + \hat{\Phi}_1\mathcal{ES}_t, \tag{13}$$

### 4.2 EQUILIBRIUM INDEX

The second key ingredient of ESE, Equilibrium Index (EI), is to measure the current equilibrium level of system $\mathcal{MS}$. In this study, Transformed Euclidean Distance (Equ 14) is introduced. The range of EI values is normalized in the range of $[0, 1]$. When EI approaches 0, the system becomes closer to its estimated equilibrium state. On the contrary, an EI value closer to 1 means the state is approaching extreme imbalance.

**Transformed Euclidean Distance (TED)** The core of TED is the Euclidean distance between states but with a further square root. That is to standardize the output. TED can better measure the difference between $\mathcal{ST} = \gamma_{1:n}$ and $\mathcal{ES} = \gamma_{1:n}^*$. It is more sensitive, especially for small distances.

$$EI_{TED} = \left( \frac{\sum_{i=1}^{n} (\gamma_i - \gamma_i^*)^2}{2} \right)^{1/4} \tag{14}$$

## 4.3 PREDICTOR

The equilibrium state by nature is not for pre-
diction. We need a predictor to utilize the cal-
culated equilibrium state parameter set. Equa-

$$\hat{s}_{1:n,t+i} = \theta_{t+i} \mathcal{MS}_t \mathcal{ES}_t + e_{t+i}, \tag{15}$$

tion 15 is our autoregression-based predictor, where $s_{t,1:n}$ are all systems at time $t$, $\mathcal{MS}_t$ is the total value of multi-system at time $t$, $\mathcal{ES}_t$ is the equilibrium state set at time $t$, $\theta_t$ is the coefficient obtained by log maximum likelihood at time $t$, and $e_t + i$ is the residual. The detailed process of Equation 15 is in **Appendix D**. If a system's attributes do no change over time, $\mathcal{ES}$ will be a constant. Because each equilibrium state parameter represents the proportion of the corresponding system relative to the $\mathcal{MS}$, a parameter can be regarded as the weight of that system in $\mathcal{MS}$. So this predictor can easily integrate with any existing prediction tool, such as LSTM, DeepAR, etc. When using other tools with ESE, we only need the simplified version: $\hat{s}_{1:n,t+i} = \hat{\mathcal{MS}}_t \mathcal{ES}_t$ to get the prediction of each system, where $\hat{\mathcal{MS}}_t$ is the predicted overall value obtained by other tools at time $t$.

## 5 EXPERIMENTS

Our experiments involve two parts, synthetic data and real-world COVID-19 data. They will be made publicly available as currently there is no benchmark for multi-system prediction. Common time series prediction benchmarks are not suitable [3].

## 5.1 SYNTHETIC DATASETS

Three sets of synthetic data are generated as detailed in Appendix E. Six SOTA time series prediction methods are selected in the comparison here: ARIMA ArunKumar et al. (2021), LSTM Feng et al. (2022), Dlinear Zeng et al. (2023), Informer Zhou et al. (2021), DeepAR Le Guen & Thome (2020), and PatchTST[4] Nie et al. (2023). Table 1 shows the performance in average RMSE and MAE (LHS) as well as the costs (RHS). SOTA methods either predict alone (No ESE) or are combined with ESE (With ESE). The full results for other input lengths and prediction steps are shown in **Appendix F.1** (Tables 4, 5 and 6). We can observe that **(1)** ESE by itself is competitive in performance, never being the worst; **(2)** With ESE, all predictors can perform better or at least maintain the performance; **(3)** The best of each column is either by ESE alone or a SOTA predictor but with ESE. **(4)** The cost of ESE is much lower than other predictors alone, except ARIMA. **(5)** When combined with ESE, the cost of SOTA predictors can be significantly reduced from $1/2$ to $1/10$, except ARIMA. **(6)** More systems lead to high cost. ARIMA does not involve any training, hence is fast on small data sets, but not on large data. The complete results are shown in **Appendix F.2** (Tables 7, 8 and 9).

## 5.2 READ-WORLD COVID-19 DATA

Epidemic transmission across different regions, like the COVID-19 pandemic, is a real-world sce-
nario that can be viewed in equilibrium. Prediction is needed for each region which can be viewed as a system. Hence we use this task to validate ESE. The data are the daily case data collected by us from the state[5] government of Victoria, Australia, ranging from January 25, 2020, to Septem-
ber 16, 2022[6]. The information about the regions in Victoria and other epidemic related data are collected from two main resources: (1) government agencies, e.g. the health department and the Australian Bureau of Statistics; (2) media reports such as the Australian Broadcasting Corporation (ABC), which reports epidemic related news, e.g. large scale crowd gathering, an announcement of new government policies and mishandling in COVID-19 handling (details in **Appendix G**).

---

[3]In our experiments, all computing costs presented here are measured on AMD CPU Ryzen 9 7950X 16-Core 4.50 GHz, 64 GB memory, and GPU NVIDIA 4090 with 24 GB memory.

[4]The PatchTST used in this study are all PatchTST/64.

[5]We use "state" for two unrelated concepts: the condition of a system, and a constituent unit of a nation.

[6]After Sep/16/22, data are no longer published daily but weekly, making it unsuitable for this study.

Table 1: Comparison on three synthetic datasets of 5 systems, 10 systems and 20 systems (input size = 20 steps, prediction step = 1). The results on the left are prediction accuracies . Highlighted are the best of those columns. On the right are the costs with different input lengths.

| Models | | Performance on 5/10/20 Systems | | Cost on 5/10/20 Systems (mins) | | |
|---|---|---|---|---|---|---|
| | | RMSE | MAE | $Input = 10$ | $Input = 20$ | $Input = 50$ |
| ESE | – | 0.252 / 0.248 / 0.255 | **0.210** / 0.228 / 0.216 | 0.19 / 0.24 / 0.27 | 0.20 / 0.23 / 0.29 | 0.20 / 0.23 / 0.30 |
| ARIMA | No ESE | 0.243 / 0.249 / 0.262 | 0.229 / 0.243 / 0.235 | 0.04 / 0.09 / 0.17 | 0.04 / 0.09 / 0.17 | 0.04 / 0.08 / 0.18 |
| | With ESE | **0.240** / 0.247 / 0.261 | 0.228 / 0.243 / 0.233 | 0.20 / 0.25 / 0.28 | 0.21 / 0.24 / 0.30 | 0.21 / 0.24 / 0.31 |
| LSTM | No ESE | 0.258 / 0.263 / 0.284 | 0.212 / 0.216 / 0.216 | 1.27 / 2.34 / 5.00 | 1.22 / 2.47 / 4.69 | 1.18 / 2.35 / 4.94 |
| | With ESE | 0.255 / 0.263 / 0.280 | 0.212 / 0.212 / 0.213 | 0.44 / 0.47 / 0.52 | 0.44 / 0.48 / 0.53 | 0.43 / 0.47 / 0.55 |
| Dlinear | No ESE | 0.257 / 0.256 / 0.264 | 0.214 / 0.221 / 0.204 | 1.47 / 2.93 / 5.53 | 1.39 / 2.93 / 5.80 | 1.41 / 2.75 / 5.68 |
| | With ESE | 0.254 / 0.264 / 0.260 | **0.210** / 0.221 / 0.204 | 0.48 / 0.53 / 0.55 | 0.47 / 0.52 / 0.58 | 0.48 / 0.51 / 0.59 |
| Informer | No ESE | 0.248 / 0.244 / 0.252 | 0.213 / 0.236 / 0.245 | 0.90 / 1.71 / 3.44 | 0.86 / 1.69 / 3.40 | 0.83 / 1.66 / 3.38 |
| | With ESE | 0.246 / **0.241** / 0.251 | 0.239 / 0.232 / 0.234 | 0.37 / 0.41 / 0.45 | 0.37 / 0.40 / 0.46 | 0.36 / 0.40 / 0.47 |
| DeepAR | No ESE | 0.252 / 0.271 / 0.279 | 0.215 / 0.218 / 0.214 | 1.17 / 2.35 / 4.54 | 1.10 / 2.29 / 4.53 | 1.16 / 2.37 / 4.76 |
| | With ESE | 0.248 / 0.271 / 0.278 | 0.214 / **0.210** / **0.211** | 0.61 / 0.51 / 0.57 | 0.51 / 0.53 / 0.52 | 0.51 / 0.64 / 0.53 |
| PatchTST | No ESE | 0.263 / 0.263 / 0.247 | 0.218 / 0.224 / 0.218 | 0.90 / 1.87 / 3.76 | 0.90 / 1.84 / 3.70 | 0.94 / 1.83 / 3.67 |
| | With ESE | 0.266 / 0.265 / **0.246** | 0.219 / 0.224 / 0.218 | 0.37 / 0.43 / 0.46 | 0.38 / 0.41 / 0.46 | 0.38 / 0.41 / 0.49 |

Table 2: Comparing prediction performance with 12 SOTA methods, in RMSE, MAE, and DILATE, without ESE and with ESE, with input size = 50 steps and prediction step = 1, for 20/79/320 regions.

| Models | | Prediction Performance | | |
|---|---|---|---|---|
| | | RMSE | MAE | DILATE |
| ESE | – | 62.16 / 54.52 / 47.34 | 51.34 / 49.94 / 45.67 | 77.64 / 94.52 / 78.24 |
| VAR | – | 77.19 / 84.94 / 89.09 | 73.55 / 82.26 / 83.26 | 110.48 / 118.93 / 119.60 |
| ARIMA | No ESE | 69.84 / 72.56 / 76.42 | 68.05 / 66.87 / 65.44 | 82.41 / 102.43 / 92.31 |
| | With ESE | 61.34 / 55.46 / 48.97 | 50.34 / 50.45 / 44.74 | 87.03 / 93.56 / 74.64 |
| LSTM | No ESE | 57.69 / 60.83 / 55.47 | 47.64 / 55.87 / 52.90 | 79.99 / 93.01 / 81.25 |
| | With ESE | 60.20 / 55.77 / 47.31 | 47.37 / 50.47 / 42.48 | 83.92 / 91.45 / 73.91 |
| Dlinear | No ESE | 57.32 / 55.15 / 51.32 | 49.63 / 52.95 / 48.39 | 80.41 / 94.24 / 82.43 |
| | With ESE | 58.13 / 53.44 / 47.06 | 46.82 / 50.42 / 43.60 | 71.91 / 93.33 / 72.47 |
| Nlinear | No ESE | 56.74 / 54.22 / 49.74 | 47.41 / 51.95 / 48.01 | 78.84 / 91.45 / 77.31 |
| | With ESE | 58.14 / 55.01 / 47.13 | 45.84 / 50.34 / 42.45 | 70.45 / 93.66 / 72.04 |
| Informer | No ESE | 58.31 / 61.23 / 57.14 | 46.72 / 58.85 / 52.56 | 79.45 / 95.31 / 83.50 |
| | With ESE | 59.42 / 55.17 / 48.01 | 48.83 / 49.47 / 44.06 | 72.14 / 94.77/ 72.94 |
| FiLM | No ESE | 55.33 / 55.57 / 50.98 | 47.45 / 48.60 / 45.79 | 83.64 / 95.14 / 80.77 |
| | With ESE | 57.93 / 55.31 / 46.94 | **45.83** / 49.66 / 43.22 | **69.73** / 95.54 / 72.12 |
| SCINet | No ESE | 58.94 / 59.80 / 60.33 | 54.79 / 51.31 / 55.74 | 81.74 / 95.14 / 85.91 |
| | With ESE | 58.88 / 54.12 / **46.34** | 46.73 / 48.14 / **41.64** | 71.06 / **90.12** / **71.46** |
| DeepAR | No ESE | 61.78 / 54.64 / 61.74 | 50.74 / 50.31 / 56.41 | 81.74 / 95.65 / 85.93 |
| | With ESE | 60.03 / 52.43 / 48.52 | 48.34 / 51.02 / 43.82 | 75.06 / 95.14 / 73.64 |
| KVAE | No ESE | 54.36 / 52.41 / 49.74 | 45.96 / 50.34 / 41.90 | 78.71 / 92.14 / 77.06 |
| | With ESE | 58.22 / **52.11** / 47.42 | 46.56 / 51.32 / 43.77 | 72.41 / 93.03 / 73.92 |
| TPGNN | No ESE | **53.74** / 56.65 / 52.31 | 48.71 / 54.79 / 45.70 | 83.44 / 94.65 / 79.74 |
| | With ESE | 57.65 / 55.16 / 46.82 | 46.07 / 52.96 / 43.64 | 71.62 / 93.45 / 72.49 |
| PatchTST | No ESE | 55.43 / 54.49 / 50.74 | 49.34 / 53.35 / 42.94 | 86.41 / 89.86 / 79.46 |
| | With ESE | 59.12 / 52.58 / 46.54 | 48.49 / **47.99** / 43.51 | 79.54 / 92.19 / 79.97 |

In Victoria, there are 79 municipalities. The Victorian government reports the epidemic status of these 79 regions daily. Hence each $\mathcal{ST} = \gamma_{1:n}$ and $\mathcal{ES} = \gamma_{1:n}^*$ contain 79 systems respectively. To verify the prediction results under different granularity, we merged the 79 regions into 20 systems and also divided these regions into 320 systems, according to postcodes. The rules are that (1) merged regions must be geographically adjacent; (2) the total population of the merged regions cannot be higher than twice that of any neighboring regions; (3) the merged attribute data is the sum of the merging regions. At the level of 320 regions, the only attributes are population and band.

Twelve SOTA predictors are involved in this part of comparison: six used for synthetic data, plus VAR Hyndman & Athanasopoulos (2018), Nlinear Zeng et al. (2023), FiLM Zhou et al. (2022), SCINet Liu et al. (2022a), KVAE Tang & Matteson (2021), and TPGNN Liu et al. (2022b). Three metrics are in use: RMSE, MAE and DILATE Le Guen & Thome (2019). Their prediction performance on three levels of granularity, 20 regions, 79 regions and 320 regions, is shown in Table 2. Note VAR can predict multiple systems based on cross-system correlation so not suitable to be combined with ESE. Overall, ESE shows excellent performance as **(1)** ESE improves SOTA performance in most cases; **(2)** the best results of each column are mostly with ESE, except RMSE of 20 regions, topped by TPGNN alone; **((3)** ESE alone outperforms other predictors alone in many cases, especially under 320 regions. The full comparisons in RMSE, MAE, and DILATE, are viewable in **Appendix J.1**. To further illustrate ESE's advantage over an increasing number of systems, we plot RMSE of these methods with 20, 79 and 320 regions in Figure 3. ESE can perform better with more regions. In comparison, other methods either deteriorate or do not improve as much. Another point

to highlight is that when combined with ESE, these 12 methods also show a similar trend as that of ESE alone. Figure 4 illustrates how ESE handles different input sizes, ranging from 10 to 100.It clearly shows that ESE can handle large inputs as most of the lowest RMSE with input over 50 are either from ESE or SOTA methods combined with ESE.

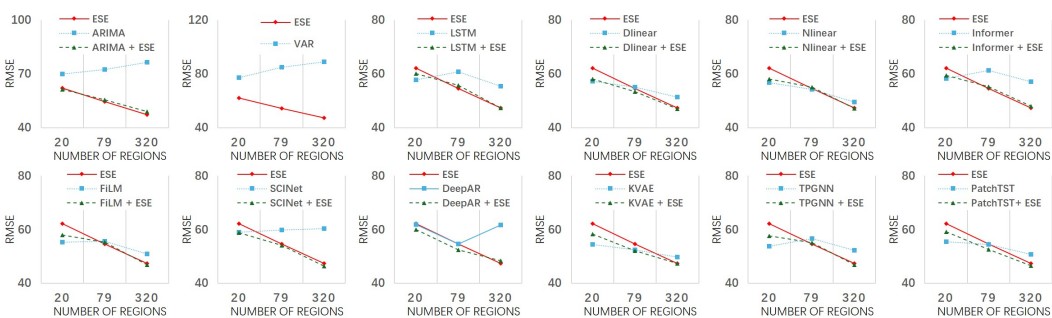

Figure 3: Comparing with 12 SOTA methods in RMSE on different numbers of systems, 20, 79 and 320 (input size = 50, step = 1 ).

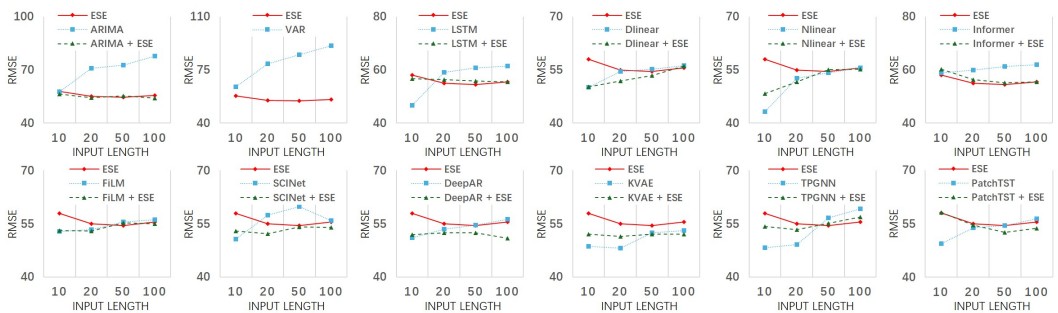

Figure 4: Comparing with 12 SOTA methods in RMSE on different input sizes, 10, 20, 50 and 100 (79 regions, step = 1 )

## 6 COMPUTATIONAL COST AND COMPLEXITY ANALYSIS

ESE has a significant cost advantage in multi-system prediction as it requires no repetition for predicting each system separately. Table 3 shows the computational cost of ESE vs. 12 SOTA methods for predicting 20/79/320 regions with input lengths of 10, 20, 50 and 100 respectively. The full comparison on costs is in **Appendix J.2**. ESE's cost is significantly lower than other methods, especially with longer inputs and more regions. Note, ARIMIA is based on least squares Singh et al. (2020), hence of low cost, but still slower than ESE on 320 regions. More importantly, ESE can greatly reduce costs for all these twelve methods when combined. In the case of FiLM and SCINet on 320 regions, the acceleration enabled by ESE is 70+ times (bold in Table 3).

Section 4 shows there are no costly operations in ESE. As shown in Eq. 11, $\psi_j$ and $\lambda_{i,j}$ reflect the number of systems and attributes respectively. Also, as shown in Line 3 of Algorithm 1, the number of iterations is proportional to time step $t$. That means Algorithm 1 is of linear complexity. The computational cost is linear to the number of systems, the number of attributes and the time steps. That is consistent with the analysis using COVID data, shown in Fig. 5. The X-axis represents the number of regions, ranging from 1 to 79, and the y-axis on the left represents the number of inputs, ranging from 5 to 100. All points are coloured in four bands. A similar linear trend can also be observed in the right of Fig. 5, which shows a linear increase in cost with the number of regions and the number of attributes.

## 7 CONCLUSION

This study proposes ESE, a new paradigm of prediction method to handle multi-system prediction. Unlike conventional methods, ESE is based on the concept of equilibrium. It does not treat multi-

Table 3: Comparing computational cost with 12 SOTA methods, with no ESE and with ESE, with 10, 20, 50, 100 steps of input, 1 step output, for 20/79/320 regions.

| Models | | Computational Costs (mins) | | | |
|---|---|---|---|---|---|
| | | $InputLength = 10$ | $InputLength = 20$ | $InputLength = 50$ | $InputLength = 100$ |
| ESE | – | 1.19 / 1.49 / 1.71 | 1.23 / 1.43 / 1.82 | 1.22 / 1.45 / 1.97 | 1.31 / 2.10 / 2.28 |
| ARIMA | No ESE | 0.18 / 0.70 / 2.81 | 0.22 / 0.89 / 3.59 | 0.27 / 1.09 / 4.11 | 0.33 / 1.31 / 5.07 |
| | With ESE | 1.20 / 1.50 / 1.72 | 1.24 / 1.44 / 1.83 | 1.23 / 1.46 / 1.99 | 1.33 / 2.12 / 2.30 |
| LSTM | No ESE | 5.06 / 20.68 / 86.88 | 6.14 / 27.76 / 109.29 | 7.77 / 33.81 / 131.74 | 9.23 / 39.96 / 149.26 |
| | With ESE | 1.46 / 1.77 / 1.98 | 1.57 / 1.74 / 2.13 | 1.61 / 1.84 / 2.41 | 1.80 / 2.59 / 2.72 |
| Dlinear | No ESE | 6.20 / 23.96 / 96.44 | 7.07 / 31.61 / 132.31 | 10.28 / 38.58 / 159.46 | 10.40 / 40.98 / 163.57 |
| | With ESE | 1.48 / 1.79 / 2.03 | 1.62 / 1.81 / 2.18 | 1.69 / 1.90 / 2.49 | 1.86 / 2.68 / 2.87 |
| Nlinear | No ESE | 6.04 / 24.91 / 99.01 | 7.40 / 31.37 / 135.11 | 9.57 / 37.83 / 155.00 | 11.56 / 45.67 / 160.09 |
| | With ESE | 1.50 / 1.81 / 2.01 | 1.60 / 1.80 / 2.23 | 1.72 / 1.93 / 2.41 | 1.87 / 2.70 / 2.86 |
| Informer | No ESE | 3.52 / 13.24 / 57.48 | 4.56 / 17.16 / 74.00 | 5.38 / 20.58 / 93.92 | 5.95 / 26.97 / 100.08 |
| | With ESE | 1.36 / 1.67 / 1.88 | 1.43 / 1.67 / 2.03 | 1.48 / 1.73 / 2.25 | 1.66 / 2.40 / 2.62 |
| FiLM | No ESE | 6.63 / 26.22 / 108.97 | 7.62 / 34.91 / 143.09 | 9.66 / 40.59 / **171.36** | 11.70 / 48.55 / 181.06 |
| | With ESE | 1.50 / 1.82 / 2.02 | 1.65 / 1.87 / 2.24 | 1.73 / 1.96 / **2.44** | 1.86 / 2.74 / 2.84 |
| SCINet | No ESE | 7.98 / 30.62 / 127.32 | 10.37 / 40.89 / 151.48 | 12.24 / 49.96 / 189.39 | 14.18 / 62.27 / **206.06** |
| | With ESE | 1.57 / 1.88 / 2.12 | 1.70 / 1.92 / 2.31 | 1.78 / 2.11 / 2.60 | 2.04 / 2.82 / **2.94** |
| DeepAR | No ESE | 5.11 / 19.57 / 76.72 | 6.16 / 23.56 / 94.99 | 7.34 / 31.55 / 131.00 | 9.22 / 37.25 / 130.67 |
| | With ESE | 1.43 / 1.74 / 1.95 | 1.52 / 1.72 / 2.15 | 1.63 / 1.87 / 2.39 | 1.77 / 2.52 / 2.73 |
| KVAE | No ESE | 4.37 / 17.03 / 67.34 | 5.21 / 22.17 / 90.23 | 7.19 / 28.31 / 96.23 | 6.80 / 32.41 / 109.62 |
| | With ESE | 1.41 / 1.71 / 1.92 | 1.49 / 1.69 / 2.07 | 1.53 / 1.78 / 2.30 | 1.67 / 2.50 / 2.63 |
| TPGNN | No ESE | 5.87 / 23.56 / 97.40 | 7.90 / 27.60 / 119.72 | 9.85 / 35.00 / 152.22 | 10.58 / 42.92 / 158.84 |
| | With ESE | 1.49 / 1.80 / 2.00 | 1.62 / 1.82 / 2.19 | 1.72 / 1.91 / 2.48 | 1.81 / 2.65 / 2.86 |
| PatchTST | No ESE | 3.71 / 15.55 / 61.08 | 4.60 / 18.68 / 82.90 | 5.85 / 23.96 / 94.47 | 6.92 / 27.70 / 109.89 |
| | With ESE | 1.38 / 1.69 / 1.90 | 1.48 / 1.69 / 2.08 | 1.53 / 1.78 / 2.29 | 1.79 / 2.46 / 2.66 |

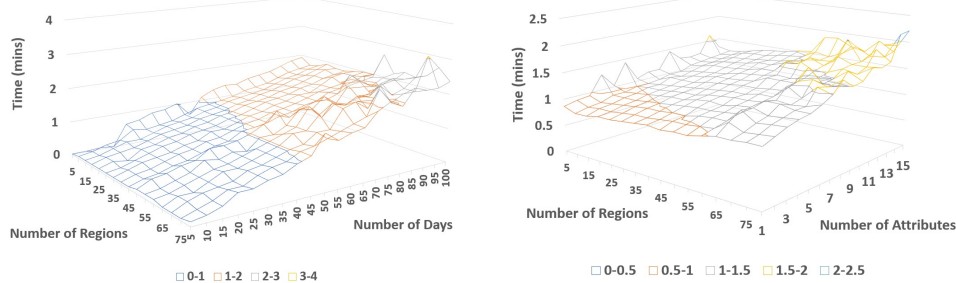

Figure 5: Left: ESE's cost relative to the number of regions and the number of days (with 9 attributes). Right: to the number of regions and the number of attributes (input size = 150 ).

systems as multiple time series but as a body of interacting systems. By analyzing the equilibrium state holistically, ESE can forecast the development of the whole group and all the systems. Hence it can perform integrated multi-system prediction with just one run. More importantly, ESE can act alone or integrate with existing prediction methods. Our extensive experiments demonstrate its effectiveness on three sets of synthetic data and large real-world COVID-19 data. ESE can achieve an equivalent level of performance with SOTA methods but with much less cost. When integrated with other methods, ESE can improve performance yet significantly reduce the cost. Furthermore, it can easily handle different granularities, especially large-scale multi-systems with no negative impact on prediction performance, yet with no significant cost increase due to its low complexity.

Hence, we conclude that ESE is an effective and efficient integrated multi-system prediction mechanism. It can bring significant value to the real world, as it can be a powerful tool to predict not just COVID-19 but also other types of epidemic spreading and complex economic and finance analysis.

**Further Discussion** ESE method does have limitations. (1) It is based on equilibrium, so when encountering a scenario with no equilibrium state, or the collected data are incomplete, ESE will not be suitable because Nash equilibrium and zero-sum conditions are not met. (2) ESE is more suitable for handling prediction with long inputs. For input of short-length, ESE may not be able to obtain sufficient information to estimate the equilibrium state, as shown in Appendix J.1, Tables 12 -15.

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

# A   THE PROOF PROCESS OF REMOVING THE INTERFERENCE CAUSED BY THE TREND OF INTEGRATED MULTI-SYSTEM

$$
\begin{aligned}
\triangle \mathcal{M}\mathcal{S}_t &= \mathcal{M}\mathcal{S}_t - \mathcal{M}\mathcal{S}_{t-1} \\
&= \sum_{i=1}^{n} s_{i,t} - \sum_{i=1}^{n} s_{i,t-1} \\
&= \sum_{i=1}^{n} \gamma_{i,t} \mathcal{M}\mathcal{S}_t - \sum_{i=1}^{n} \gamma_{i,t-1} \mathcal{M}\mathcal{S}_{t-1},
\end{aligned}
\tag{16}
$$

where $\triangle \mathcal{M}\mathcal{S}_t$ is the change in the system. According to Equation 16, no matter how $\mathcal{M}\mathcal{S}_t$ changes, $\sum_{i=1}^{n} \gamma_{i,t}$ must and always equals to 1.

# B   SIMPLIFIED EQUILIBRIUM STATE IN ESE

The concept of equilibrium in this study derives from the original definition of Nash equilibrium. In its original form with no zero-sum game assumption, Equation 6 can be extended as **Lemma 1: Equation 7**, which is also shown below.

$$
\begin{aligned}
U_1(\mathcal{A}_1^*, \mathcal{A}_2^*, \ldots, \mathcal{A}_n^*) &\geq U_1(\mathcal{A}_1, \mathcal{A}_2^* \ldots, \mathcal{A}_n^*) \\
U_2(\mathcal{A}_1^*, \mathcal{A}_2^*, \ldots, \mathcal{A}_n^*) &\geq U_2(\mathcal{A}_1^*, \mathcal{A}_2 \ldots, \mathcal{A}_n^*) \\
&\vdots \\
U_n(\mathcal{A}_1^*, \mathcal{A}_2^*, \ldots, \mathcal{A}_n^*) &\geq U_n(\mathcal{A}_1^*, \mathcal{A}_2^*, \ldots, \mathcal{A}_n),
\end{aligned}
\tag{17}
$$

With zero-sum assumption, the formulation can be simplified. Using a two-player scenario as an example, the payoff functions for two players can be expressed as Equation 18:

$$
\begin{aligned}
U_1(\mathcal{A}_1, \mathcal{A}_2) &= \sum_{j=1}^{J} \sum_{k=1}^{K} u_1(\theta_{1,j} \cdot \alpha_{1,j}, \ \phi_{1,k} \cdot \alpha_{2,k}); \\
U_2(\mathcal{A}_1, \mathcal{A}_2) &= \sum_{j=1}^{J} \sum_{k=1}^{K} u_2(\theta_{2,j} \cdot \alpha_{1,j}, \ \phi_{2,k} \cdot \alpha_{2,k}),
\end{aligned}
\tag{18}
$$

where $U_i()$ is the payoff function for player $i$ under multiple decisions (attributes). $\mathcal{A}_1$ is the decision (attribute) set of player 1, containing $J$ different decisions, $(\alpha_{1,1}, \ldots, \alpha_{1,J})$. $\mathcal{A}_2$ is the decision (attribute) set of player 2, containing $K$ different decisions, $(\alpha_{2,1}, \ldots, \alpha_{2,K})$. All attributes, e.g. $\alpha_{1,j}$ and $\alpha_{2,k}$ are not independent and may influence each other. In the equations, $\theta_{1,j}$ is the coefficient on attribute $\alpha_{1,j}$ of player 1, while $\phi_{1,k}$ is the coefficient on attribute $\alpha_{1,k}$ of player 2, both on the payoff function of player 1. Similarly, $\theta_{2,j}$ and $\phi_{2,k}$ are the corresponding coefficients on the payoff function of player 2.

For ESE, we assume zero-sum for the equilibrium. With this assumption, the attributes of the players will be independent of each other. One attribute only affects the same attribute of other players. Therefore there is no need to compute full interactions and feedback loops, which can be exponentially expensive. Furthermore, as set in Definition 3, the attribute set $\mathcal{A}$ of every player is identical. Therefore, we can greatly simplify Equation 18 to Equation 19, as shown below:

$$
\begin{aligned}
U_1(\mathcal{A}_1, \mathcal{A}_2) &= \sum_{j=1}^{J} u(\psi_j \cdot \alpha_{1,j}, \ \psi_j \cdot \alpha_{2,j}); \\
U_2(\mathcal{A}_1, \mathcal{A}_2) &= \sum_{j=1}^{J} u(\psi_j \cdot \alpha_{1,j}, \ \psi_j \cdot \alpha_{2,j}),
\end{aligned}
\tag{19}
$$

The payoff function $u()$ in both $U_1()$ and $U_2()$ are identical. Since attributes of the same type are independent under the zero-sum assumption, the coefficients $\psi_j$ for attribute $j$ on $u()$ are identical for all players. Therefore, $U_1(\mathcal{A}_1, \mathcal{A}_2)$ and $U_2(\mathcal{A}_1, \mathcal{A}_2)$ are the same and can be combined as $U(\mathcal{A}_1, \mathcal{A}_2)$. The payoff functions for all players can be calculated by just one payoff function $U(\mathcal{A}_1, \mathcal{A}_2)$. Subsequently, the equilibrium state can be simplified as below, also Equation. 8 in the main paper:

$$
U(\mathcal{A}_1^*, \mathcal{A}_2^*, \ldots, \mathcal{A}_n^*) \geq \cdots \geq U(\mathcal{A}_1, \mathcal{A}_2 \ldots, \mathcal{A}_n).
\tag{20}
$$

## C  ESE CONVERGENCE PROCESS

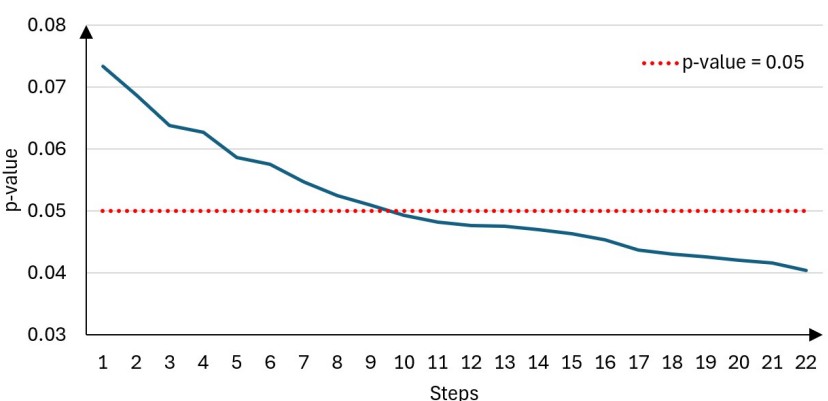

Figure 6: Convergence of ESE training on Synthetic Data, 20 Systems. The blue line represents the p-values obtained at each step of ESE training. The red dotted line represents a p-value of 0.05, the threshold for rejecting the null hypothesis for the existence of a long-run equilibrium.

Figures 6 and 7 show the analysis on the convergence during ESE training, on synthetic data and COVID data respectively. The p-values are from the cointegration test (Step 2, Algorithm 1), where the null hypothesis is rejected if the value is lower than 0.05. For illustration purposes, we allow the convergence continues beyond 0.05 on these two figures. During an actual training, it will stop once the p-value reaches 0.05, showing the existence of a long-run equilibrium. From the figures we can see the ESE convergence process is steady and effective. With this, we don't need to be too concerned about stochastic Fleming & Rishel (2012) and oscillation behaviors Morin (2008) which can often be observed in real world data, like COVID. More details about conintegration and long-run equilibrium can be found in Maki & Kitasaka (2006); Chen et al. (2009).

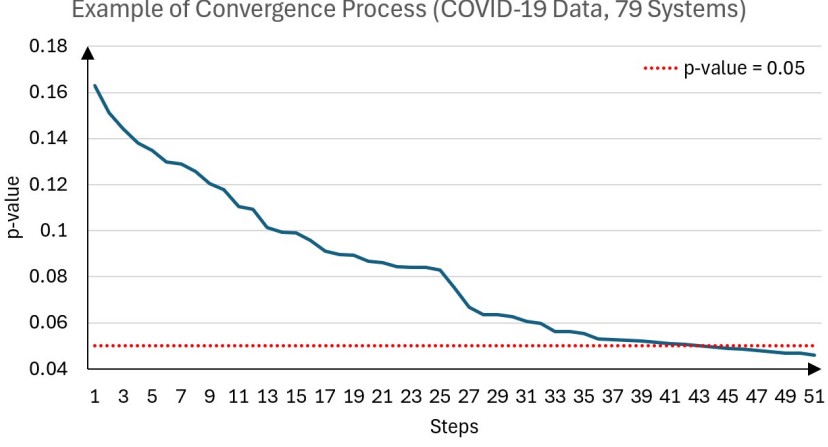

Figure 7: Convergence of ESE training on COVID-19 Data, 79 Systems. The blue line represents p-values obtained from the cointegration test at each step of ESE training. The red dotted line is the threshold for rejecting the null hypothesis for the existence of a long-run equilibrium.

# D PREDICTOR

Equations 21, 22, 23 show the estimation process of Equation 15 by log maximum likelihood.

$$\mathcal{MS}_{t+i} = \theta_{t+i}\mathcal{MS}_t + e_{t+i},$$
$$e_{t+i} \overset{i.i.d.}{\sim} N(0, \sigma^2); \quad \mathcal{MS}_t = \sum_{i=1}^{n} c_{t,i} \tag{21}$$

$$\hat{\theta_{mle}} = argmax\ logL(\theta), \tag{22}$$

$$L(\theta) \overset{def}{=} p(\mathcal{MS}_1, \ldots, \mathcal{MS}_k|\theta) =$$
$$p(\mathcal{MS}_1)\left(\frac{1}{\sigma\sqrt{2\pi}}\right)^{k-1} exp\left\{-\frac{1}{2\sigma^2}\sum_{t=2}^{k}(\mathcal{MS}_t - \theta\mathcal{MS}_{t-1})^2\right\}, \tag{23}$$
$$logL(\theta) =$$
$$logp(\mathcal{MS}_1) - (k-1)log(\sigma\sqrt{2\pi}) - \frac{1}{2\sigma^2}(\mathcal{MS}_t - \theta\mathcal{MS}_{t-1})^2$$

where $\mathcal{MS}_t$ is the total value of the system at time $t$. $\theta_t$ is the parameter of the model at time $t$, which was estimated by log maximum likelihood.

## E  SYNTHETIC DATA

To validate ESE, three systems are synthesized, consisting of 5, 10, and 20 systems respectively. Each system contains a series of targets and two series of attributes for 1000 time points. They are generated by Equation 24,

$$y_t = log(C + \beta y_{t-1} + e_t) \tag{24}$$

where $\beta$ is an adjustable coefficient, with a value of $\beta = 1.2$ in this study. $C$ is the intercept, which is randomly chosen from $[50, 100]$ for the target and from $[1, 10]$ for the attributes. To add some white noise, a random number ($e_t$) is added in the range of $[-1, 1]$ based on Gaussian distribution.

## F  FULL COMPARISONS ON SYNTHETIC DATA

### F.1  THE COMPUTATIONAL COST (MINUTE) FOR PREDICTING SYNTHETIC DATA

Table 4: Comparison of prediction results for 5/10/20 systems by using synthetic data. RMSE and MAE are means of prediction results based on different input lengths (10,20,50), and the prediction target is fixed at 1 step.

| Models | | Metric | Predicting 1 Step | | |
| --- | --- | --- | --- | --- | --- |
| | | | *Input Length = 10* | *Input Length = 20* | *Input Length = 50* |
| ESE | | RMSE | 0.246±0.01 / 0.246±0.01 / 0.245±0.01 | 0.252±0.01 / 0.248±0.01 / 0.255±0.01 | 0.295±0.01 / 0.270±0.01 / 0.282±0.01 |
| | | MAE | 0.208±0.01 / 0.207±0.01 / 0.207±0.01 | 0.210±0.01 / 0.228±0.01 / 0.216±0.01 | 0.229±0.01 / 0.261±0.01 / 0.257±0.01 |
| ARIMA | No ESE | RMSE | 0.241±0.01 / 0.241±0.01 / 0.241±0.01 | 0.243±0.02 / 0.249±0.02 / 0.262±0.01 | 0.276±0.01 / 0.289±0.01 / 0.314±0.02 |
| | | MAE | 0.222±0.01 / 0.222±0.01 / 0.222±0.01 | 0.229±0.01 / 0.243±0.01 / 0.235±0.01 | 0.248±0.01 / 0.283±0.01 / 0.246±0.01 |
| | With ESE | RMSE | 0.243±0.02 / 0.243±0.01 / 0.239±0.02 | 0.240±0.02 / 0.247±0.01 / 0.261±0.02 | 0.276±0.02 / 0.291±0.01 / 0.312±0.02 |
| | | MAE | 0.221±0.01 / 0.220±0.01 / 0.223±0.01 | 0.228±0.01 / 0.243±0.01 / 0.233±0.01 | 0.247±0.01 / 0.285±0.01 / 0.248±0.01 |
| LSTM | No ESE | RMSE | 0.250±0.01 / 0.250±0.01 / 0.260±0.01 | 0.258±0.01 / 0.263±0.01 / 0.284±0.01 | 0.298±0.01 / 0.297±0.01 / 0.297±0.01 |
| | | MAE | 0.203±0.01 / 0.203±0.01 / 0.202±0.01 | 0.212±0.01 / 0.216±0.01 / 0.216±0.01 | 0.226±0.01 / 0.245±0.01 / 0.247±0.01 |
| | With ESE | RMSE | 0.249±0.01 / 0.250±0.01 / 0.259±0.01 | 0.255±0.01 / 0.263±0.01 / 0.280±0.01 | 0.298±0.01 / 0.296±0.01 / 0.296±0.01 |
| | | MAE | 0.204±0.01 / 0.202±0.01 / 0.202±0.01 | 0.212±0.01 / 0.212±0.01 / 0.213±0.01 | 0.224±0.01 / 0.247±0.01 / 0.246±0.01 |
| Dlinear | No ESE | RMSE | 0.244±0.01 / 0.244±0.01 / 0.243±0.01 | 0.257±0.01 / 0.256±0.01 / 0.264±0.01 | 0.309±0.01 / 0.289±0.01 / 0.267±0.01 |
| | | MAE | 0.203±0.01 / 0.205±0.01 / 0.211±0.01 | 0.214±0.01 / 0.221±0.01 / 0.204±0.01 | 0.223±0.01 / 0.230±0.01 / 0.234±0.01 |
| | With ESE | RMSE | 0.242±0.01 / 0.241±0.01 / 0.245±0.01 | 0.254±0.01 / 0.264±0.01 / 0.260±0.01 | 0.309±0.01 / 0.287±0.01 / 0.265±0.01 |
| | | MAE | 0.215±0.01 / 0.214±0.01 / 0.203±0.01 | 0.215±0.01 / 0.221±0.01 / 0.204±0.01 | 0.225±0.01 / 0.232±0.01 / 0.232±0.01 |
| Informer | No ESE | RMSE | 0.243±0.01 / 0.243±0.01 / 0.249±0.01 | 0.248±0.01 / 0.244±0.01 / 0.252±0.01 | 0.283±0.01 / 0.283±0.01 / 0.281±0.01 |
| | | MAE | 0.225±0.01 / 0.237±0.01 / 0.233±0.01 | 0.213±0.01 / 0.236±0.01 / 0.245±0.01 | 0.227±0.01 / 0.278±0.01 / 0.268±0.01 |
| | With ESE | RMSE | 0.242±0.01 / 0.242±0.01 / 0.251±0.01 | 0.246±0.01 / 0.241±0.01 / 0.251±0.01 | 0.283±0.01 / 0.282±0.01 / 0.279±0.01 |
| | | MAE | 0.226±0.01 / 0.237±0.01 / 0.246±0.01 | 0.239±0.01 / 0.232±0.01 / 0.234±0.01 | 0.228±0.01 / 0.256±0.01 / 0.271±0.01 |
| DeepAR | No ESE | RMSE | 0.248±0.01 / 0.248±0.01 / 0.255±0.01 | 0.252±0.01 / 0.271±0.01 / 0.279±0.01 | 0.281±0.01 / 0.215±0.01 / 0.335±0.01 |
| | | MAE | 0.203±0.01 / 0.204±0.01 / 0.209±0.01 | 0.215±0.01 / 0.218±0.01 / 0.214±0.01 | 0.251±0.01 / 0.244±0.01 / 0.257±0.01 |
| | With ESE | RMSE | 0.249±0.01 / 0.250±0.01 / 0.257±0.01 | 0.248±0.01 / 0.271±0.01 / 0.278±0.01 | 0.280±0.01 / 0.314±0.01 / 0.334±0.01 |
| | | MAE | 0.202±0.01 / 0.205±0.01 / 0.204±0.01 | 0.214±0.01 / 0.210±0.01 / 0.211±0.01 | 0.253±0.01 / 0.243±0.01 / 0.256±0.01 |
| PatchTST | No ESE | RMSE | 0.241±0.01 / 0.241±0.01 / 0.241±0.01 | 0.251±0.01 / 0.262±0.01 / 0.247±0.01 | 0.294±0.01 / 0.263±0.01 / 0.291±0.01 |
| | | MAE | 0.207±0.01 / 0.207±0.01 / 0.208±0.01 | 0.208±0.01 / 0.219±0.01 / 0.225±0.01 | 0.211±0.01 / 0.233±0.01 / 0.229±0.01 |
| | With ESE | RMSE | 0.240±0.01 / 0.243±0.01 / 0.243±0.01 | 0.250±0.01 / 0.260±0.01 / 0.246±0.01 | 0.292±0.01 / 0.261±0.01 / 0.291±0.01 |
| | | MAE | 0.209±0.01 / 0.208±0.01 / 0.208±0.01 | 0.207±0.01 / 0.218±0.01 / 0.225±0.01 | 0.211±0.01 / 0.232±0.01 / 0.229±0.01 |

Table 5: Comparison of prediction results for 5/10/20 systems by using synthetic data. RMSE and MAE are means of prediction results based on different input lengths (10,20,50), and the prediction target is fixed at 2 steps.

| Models | | Metric | Predicting 2 Steps | | |
| --- | --- | --- | --- | --- | --- |
| | | | Input Length = 10 | Input Length = 20 | Input Length = 50 |
| ESE | | RMSE | 0.269±0.01 / 0.267±0.01 / 0.266±0.01 | 0.284±0.01 / 0.276±0.01 / 0.285±0.01 | 0.336±0.01 / 0.283±0.01 / 0.320±0.01 |
| | | MAE | 0.218±0.01 / 0.218±0.01 / 0.215±0.01 | 0.234±0.01 / 0.236±0.01 / 0.228±0.01 | 0.264±0.01 / 0.266±0.01 / 0.266±0.01 |
| ARIMA | No ESE | RMSE | 0.254±0.01 / 0.244±0.01 / 0.253±0.02 | 0.246±0.01 / 0.256±0.02 / 0.258±0.02 | 0.293±0.02 / 0.270±0.01 / 0.279±0.02 |
| | | MAE | 0.216±0.01 / 0.215±0.01 / 0.213±0.01 | 0.235±0.01 / 0.220±0.01 / 0.221±0.01 | 0.246±0.01 / 0.235±0.01 / 0.261±0.01 |
| | With ESE | RMSE | 0.258±0.01 / 0.256±0.02 / 0.255±0.02 | 0.259±0.01 / 0.253±0.01 / 0.256±0.02 | 0.290±0.01 / 0.272±0.02 / 0.281±0.02 |
| | | MAE | 0.218±0.01 / 0.215±0.01 / 0.216±0.01 | 0.234±0.01 / 0.219±0.01 / 0.217±0.01 | 0.245±0.01 / 0.233±0.01 / 0.263±0.01 |
| LSTM | No ESE | RMSE | 0.269±0.01 / 0.267±0.01 / 0.266±0.01 | 0.292±0.01 / 0.282±0.01 / 0.278±0.01 | 0.337±0.01 / 0.290±0.01 / 0.304±0.01 |
| | | MAE | 0.215±0.01 / 0.214±0.01 / 0.213±0.01 | 0.226±0.01 / 0.223±0.01 / 0.219±0.01 | 0.268±0.01 / 0.266±0.01 / 0.252±0.01 |
| | With ESE | RMSE | 0.270±0.01 / 0.267±0.01 / 0.269±0.01 | 0.289±0.01 / 0.277±0.01 / 0.275±0.01 | 0.341±0.01 / 0.292±0.01 / 0.305±0.01 |
| | | MAE | 0.215±0.01 / 0.217±0.01 / 0.216±0.01 | 0.222±0.01 / 0.221±0.01 / 0.215±0.01 | 0.268±0.01 / 0.264±0.01 / 0.243±0.01 |
| Dlinear | No ESE | RMSE | 0.257±0.01 / 0.251±0.01 / 0.250±0.01 | 0.281±0.01 / 0.259±0.01 / 0.272±0.01 | 0.282±0.01 / 0.268±0.01 / 0.296±0.01 |
| | | MAE | 0.214±0.01 / 0.224±0.01 / 0.213±0.01 | 0.226±0.01 / 0.219±0.01 / 0.216±0.01 | 0.256±0.01 / 0.244±0.01 / 0.246±0.01 |
| | With ESE | RMSE | 0.260±0.01 / 0.256±0.01 / 0.253±0.01 | 0.278±0.01 / 0.256±0.01 / 0.268±0.01 | 0.284±0.01 / 0.268±0.01 / 0.295±0.01 |
| | | MAE | 0.215±0.01 / 0.214±0.01 / 0.213±0.01 | 0.227±0.01 / 0.216±0.01 / 0.214±0.01 | 0.255±0.01 / 0.245±0.01 / 0.246±0.01 |
| Informer | No ESE | RMSE | 0.254±0.01 / 0.244±0.01 / 0.253±0.01 | 0.272±0.01 / 0.273±0.01 / 0.270±0.01 | 0.293±0.01 / 0.316±0.01 / 0.283±0.01 |
| | | MAE | 0.218±0.01 / 0.217±0.01 / 0.227±0.01 | 0.228±0.01 / 0.218±0.01 / 0.234±0.01 | 0.262±0.01 / 0.235±0.01 / 0.242±0.01 |
| | With ESE | RMSE | 0.255±0.01 / 0.254±0.01 / 0.258±0.01 | 0.270±0.01 / 0.266±0.01 / 0.276±0.01 | 0.296±0.01 / 0.318±0.01 / 0.283±0.01 |
| | | MAE | 0.218±0.01 / 0.221±0.01 / 0.222±0.01 | 0.226±0.01 / 0.217±0.01 / 0.231±0.01 | 0.261±0.01 / 0.236±0.01 / 0.245±0.01 |
| DeepAR | No ESE | RMSE | 0.249±0.01 / 0.257±0.01 / 0.255±0.01 | 0.250±0.01 / 0.266±0.01 / 0.260±0.01 | 0.277±0.01 / 0.275±0.01 / 0.299±0.01 |
| | | MAE | 0.212±0.01 / 0.211±0.01 / 0.206±0.01 | 0.223±0.01 / 0.214±0.01 / 0.226±0.01 | 0.244±0.01 / 0.241±0.01 / 0.238±0.01 |
| | With ESE | RMSE | 0.251±0.01 / 0.258±0.01 / 0.248±0.01 | 0.246±0.01 / 0.264±0.01 / 0.259±0.01 | 0.276±0.01 / 0.286±0.01 / 0.302±0.01 |
| | | MAE | 0.215±0.01 / 0.214±0.01 / 0.212±0.01 | 0.209±0.01 / 0.220±0.01 / 0.225±0.01 | 0.253±0.01 / 0.243±0.01 / 0.237±0.01 |
| PatchTST | No ESE | RMSE | 0.254±0.01 / 0.253±0.01 / 0.255±0.01 | 0.260±0.01 / 0.260±0.01 / 0.263±0.01 | 0.289±0.01 / 0.278±0.01 / 0.286±0.01 |
| | | MAE | 0.212±0.01 / 0.216±0.01 / 0.216±0.01 | 0.217±0.01 / 0.224±0.01 / 0.229±0.01 | 0.229±0.01 / 0.242±0.01 / 0.250±0.01 |
| | With ESE | RMSE | 0.253±0.01 / 0.251±0.01 / 0.258±0.01 | 0.258±0.01 / 0.258±0.01 / 0.263±0.01 | 0.290±0.01 / 0.278±0.01 / 0.283±0.01 |
| | | MAE | 0.211±0.01 / 0.216±0.01 / 0.216±0.01 | 0.217±0.01 / 0.225±0.01 / 0.228±0.01 | 0.228±0.01 / 0.243±0.01 / 0.249±0.01 |

Table 6: Comparison of prediction results for 5/10/20 systems by using synthetic data. RMSE and MAE are means of prediction results based on different input lengths (10,20,50), and the prediction target is fixed at 5 steps.

| Models | | Metric | Predicting 5 Steps | | |
| --- | --- | --- | --- | --- | --- |
| | | | Input Length = 10 | Input Length = 20 | Input Length = 50 |
| ESE | | RMSE | 0.277±0.01 / 0.270±0.01 / 0.267±0.01 | 0.286±0.01 / 0.280±0.01 / 0.289±0.01 | 0.317±0.01 / 0.312±0.01 / 0.340±0.01 |
| | | MAE | 0.223±0.01 / 0.223±0.01 / 0.232±0.01 | 0.241±0.01 / 0.256±0.01 / 0.249±0.01 | 0.266±0.01 / 0.268±0.01 / 0.267±0.01 |
| ARIMA | No ESE | RMSE | 0.280±0.01 / 0.277±0.01 / 0.271±0.02 | 0.283±0.01 / 0.295±0.01 / 0.294±0.02 | 0.337±0.02 / 0.353±0.01 / 0.304±0.02 |
| | | MAE | 0.231±0.01 / 0.230±0.01 / 0.229±0.01 | 0.243±0.01 / 0.237±0.01 / 0.259±0.01 | 0.303±0.01 / 0.264±0.01 / 0.278±0.01 |
| | With ESE | RMSE | 0.280±0.01 / 0.286±0.02 / 0.280±0.02 | 0.293±0.01 / 0.291±0.02 / 0.280±0.02 | 0.323±0.02 / 0.341±0.02 / 0.291±0.02 |
| | | MAE | 0.234±0.01 / 0.239±0.01 / 0.230±0.01 | 0.243±0.01 / 0.233±0.01 / 0.248±0.01 | 0.288±0.01 / 0.254±0.01 / 0.270±0.01 |
| LSTM | No ESE | RMSE | 0.271±0.01 / 0.261±0.01 / 0.271±0.01 | 0.292±0.01 / 0.284±0.01 / 0.286±0.01 | 0.335±0.01 / 0.351±0.01 / 0.302±0.01 |
| | | MAE | 0.232±0.01 / 0.228±0.01 / 0.232±0.01 | 0.240±0.01 / 0.241±0.01 / 0.252±0.01 | 0.275±0.01 / 0.251±0.01 / 0.268±0.01 |
| | With ESE | RMSE | 0.276±0.01 / 0.271±0.01 / 0.266±0.01 | 0.287±0.01 / 0.279±0.01 / 0.274±0.01 | 0.319±0.01 / 0.335±0.01 / 0.289±0.01 |
| | | MAE | 0.233±0.01 / 0.237±0.01 / 0.235±0.01 | 0.236±0.01 / 0.238±0.01 / 0.248±0.01 | 0.265±0.01 / 0.241±0.01 / 0.260±0.01 |
| Dlinear | No ESE | RMSE | 0.272±0.01 / 0.270±0.01 / 0.271±0.01 | 0.296±0.01 / 0.286±0.01 / 0.302±0.01 | 0.339±0.01 / 0.341±0.01 / 0.357±0.02 |
| | | MAE | 0.231±0.01 / 0.233±0.01 / 0.238±0.01 | 0.246±0.01 / 0.242±0.01 / 0.255±0.01 | 0.267±0.01 / 0.260±0.01 / 0.273±0.01 |
| | With ESE | RMSE | 0.277±0.01 / 0.275±0.01 / 0.272±0.01 | 0.293±0.01 / 0.293±0.01 / 0.289±0.01 | 0.323±0.01 / 0.327±0.01 / 0.343±0.01 |
| | | MAE | 0.234±0.01 / 0.236±0.01 / 0.238±0.01 | 0.244±0.01 / 0.238±0.01 / 0.245±0.01 | 0.256±0.01 / 0.249±0.01 / 0.262±0.01 |
| Informer | No ESE | RMSE | 0.264±0.01 / 0.274±0.01 / 0.254±0.01 | 0.278±0.01 / 0.282±0.01 / 0.311±0.01 | 0.328±0.01 / 0.322±0.01 / 0.339±0.01 |
| | | MAE | 0.280±0.01 / 0.278±0.01 / 0.279±0.01 | 0.287±0.01 / 0.281±0.01 / 0.298±0.01 | 0.318±0.01 / 0.310±0.01 / 0.327±0.01 |
| | With ESE | RMSE | 0.233±0.01 / 0.222±0.01 / 0.232±0.01 | 0.246±0.01 / 0.251±0.01 / 0.261±0.01 | 0.291±0.01 / 0.290±0.01 / 0.268±0.01 |
| | | MAE | 0.237±0.01 / 0.235±0.01 / 0.236±0.01 | 0.245±0.01 / 0.246±0.01 / 0.249±0.01 | 0.278±0.01 / 0.279±0.01 / 0.256±0.01 |
| DeepAR | No ESE | RMSE | 0.261±0.01 / 0.271±0.01 / 0.273±0.01 | 0.278±0.01 / 0.288±0.01 / 0.304±0.01 | 0.329±0.01 / 0.357±0.01 / 0.326±0.01 |
| | | MAE | 0.232±0.01 / 0.237±0.01 / 0.231±0.01 | 0.248±0.01 / 0.243±0.01 / 0.254±0.01 | 0.265±0.01 / 0.274±0.01 / 0.291±0.01 |
| | With ESE | RMSE | 0.273±0.02 / 0.262±0.02 / 0.272±0.02 | 0.278±0.02 / 0.284±0.02 / 0.287±0.02 | 0.317±0.02 / 0.343±0.02 / 0.310±0.02 |
| | | MAE | 0.234±0.01 / 0.233±0.01 / 0.231±0.02 | 0.244±0.01 / 0.241±0.01 / 0.240±0.01 | 0.255±0.01 / 0.262±0.01 / 0.278±0.01 |
| PatchTST | No ESE | RMSE | 0.277±0.01 / 0.272±0.01 / 0.274±0.01 | 0.281±0.01 / 0.293±0.01 / 0.3287±0.01 | 0.304±0.01 / 0.295±0.01 / 0.340±0.01 |
| | | MAE | 0.233±0.01 / 0.234±0.01 / 0.234±0.01 | 0.251±0.01 / 0.239±0.01 / 0.245±0.01 | 0.289±0.01 / 0.245±0.01 / 0.265±0.01 |
| | With ESE | RMSE | 0.278±0.01 / 0.274±0.01 / 0.272±0.01 | 0.284±0.01 / 0.293±0.01 / 0.287±0.01 | 0.305±0.01 / 0.297±0.01 / 0.340±0.01 |
| | | MAE | 0.233±0.01 / 0.234±0.01 / 0.235±0.01 | 0.254±0.01 / 0.239±0.01 / 0.244±0.01 | 0.289±0.01 / 0.262±0.01 / 0.267±0.01 |

F.2 THE COMPUTATIONAL COST (MINUTE) FOR PREDICTING SYNTHETIC DATA

Table 7: Comparison of Computational Cost (minute) for 5/10/20 systems by using synthetic data. The results are based on different input lengths (10,20,50), and the prediction target is fixed at 1 step.

| Models | | Computational Costs (mins) | | |
|---|---|---|---|---|
| | | $Input\ Length = 10$ | $Input\ Length = 20$ | $Input\ Length = 50$ |
| ESE | | 0.19 / 0.24 / 0.27 | 0.20 / 0.23 / 0.29 | 0.20 / 0.23 / 0.30 |
| ARIMA | No ESE | 0.04 / 0.09 / 0.17 | 0.04 / 0.09 / 0.17 | 0.04 / 0.08 / 0.18 |
| | With ESE | 0.20 / 0.25 / 0.28 | 0.21 / 0.24 / 0.30 | 0.21 / 0.24 / 0.31 |
| LSTM | No ESE | 1.27 / 2.34 / 5.00 | 1.22 / 2.47 / 4.69 | 1.18 / 2.35 / 4.94 |
| | With ESE | 0.44 / 0.47 / 0.52 | 0.44 / 0.48 / 0.53 | 0.43 / 0.47 / 0.55 |
| Dlinear | No ESE | 1.47 / 2.93 / 5.53 | 1.39 / 2.93 / 5.80 | 1.41 / 2.75 / 5.68 |
| | With ESE | 0.48 / 0.53 / 0.55 | 0.47 / 0.52 / 0.58 | 0.48 / 0.51 / 0.59 |
| Informer | No ESE | 0.90 / 1.71 / 3.44 | 0.86 / 1.69 / 3.40 | 0.83 / 1.66 / 3.38 |
| | With ESE | 0.37 / 0.41 / 0.45 | 0.37 / 0.40 / 0.46 | 0.36 / 0.40 / 0.47 |
| DeepAR | No ESE | 1.17 / 2.35 / 4.54 | 1.10 / 2.29 / 4.53 | 1.16 / 2.37 / 4.76 |
| | With ESE | 0.61 / 0.51 / 0.57 | 0.51 / 0.53 / 0.52 | 0.51 / 0.64 / 0.53 |
| PatchTST | No ESE | 0.90 / 1.87 / 3.78 | 0.90 / 1.84 / 3.70 | 0.94 / 1.83 / 3.67 |
| | With ESE | 0.37 / 0.43 / 0.46 | 0.38 / 0.41 / 0.48 | 0.38 / 0.42 / 0.49 |

Table 8: Comparison of Computational Cost (minute) for 5/10/20 systems by using synthetic data. The results are based on different input lengths (10,20,50) and the prediction target is fixed at 2 steps.

| Models | | Computational Costs (mins) | | |
|---|---|---|---|---|
| | | $Input\ Length = 10$ | $Input\ Length = 20$ | $Input\ Length = 50$ |
| ESE | | 0.18 / 0.25 / 0.28 | 0.20 / 0.24 / 0.28 | 0.19 / 0.24 / 0.31 |
| ARIMA | No ESE | 0.04 / 0.09 / 0.18 | 0.04 / 0.09 / 0.17 | 0.04 / 0.08 / 0.17 |
| | With ESE | 0.19 / 0.26 / 0.29 | 0.21 / 0.25 / 0.29 | 0.20 / 0.25 / 0.32 |
| LSTM | No ESE | 1.27 / 2.50 / 4.76 | 1.28 / 2.37 / 5.10 | 1.21 / 2.47 / 4.89 |
| | With ESE | 0.43 / 0.50 / 0.51 | 0.46 / 0.48 / 0.54 | 0.43 / 0.48 / 0.56 |
| Dlinear | No ESE | 1.47 / 2.92 / 5.89 | 1.48 / 2.90 / 5.92 | 1.36 / 2.82 / 5.71 |
| | With ESE | 0.47 / 0.54 / 0.57 | 0.50 / 0.53 / 0.58 | 0.46 / 0.52 / 0.60 |
| Informer | No ESE | 0.88 / 1.80 / 3.31 | 0.90 / 1.76 / 3.51 | 0.84 / 1.68 / 3.49 |
| | With ESE | 0.36 / 0.43 / 0.44 | 0.38 / 0.42 / 0.46 | 0.36 / 0.41 / 0.48 |
| DeepAR | No ESE | 1.14 / 2.31 / 4.68 | 1.11 / 2.41 / 4.79 | 1.15 / 2.29 / 4.41 |
| | With ESE | 0.41 / 0.48 / 0.51 | 0.42 / 0.48 / 0.52 | 0.42 / 0.47 / 0.53 |
| PatchTST | No ESE | 0.97 / 1.82 / 3.67 | 0.95 / 1.91 / 3.59 | 0.97 / 1.96 / 3.63 |
| | With ESE | 0.37 / 0.43 / 0.46 | 0.39 / 0.43 / 0.46 | 0.38 / 0.43 / 0.49 |

## G THE DATA OF COVID-19

"*NDC*" is the number of new cases on the day. "*PCR cases*" is the number of confirmed positive cases obtained through official tests. "*PCR test*" is the total number of tests on that day. "*RAT cases*" is the number of newly confirmed positive cases through rapid antigen tests. "*Hospitalisation*", "*ICU cases*" and "*On ventilation*" represent the number of cases in three statuses in hospitals. "*Active cases*" and "*Death*" are the total numbers of active cases and new deaths on that day in Victoria. Table 10 lists the collected data, which are the most direct indicators of the epidemic spreading status, all with timestamps. These data are numeric values manually collected on a daily basis, through the regular releases on the Victorian government data portals. Some of the attributes are for the entire state as well as different regions and suburbs, such as "*Active cases*". PCR cases are also categorized into different age groups. It should be noted that before February 4, 2022, the government only provided the daily regional new cases for PCR with full information, and the daily total new cases of the whole state for RAT but without the region, age, and other information. The details of processing RAT info are in **Appendix H**.

V1, V2, and V3 in Table 11 represent the vaccination rates of the first, second and third doses respectively for a particular region. In order to better quantify regional medical capacity as a regional

Table 9: Comparison of Computational Cost (minute) for 5/10/20 systems by using synthetic data. The results are based on different input lengths (10,20,50) and the prediction target is fixed at 5 steps.

| Models | | Computational Costs (mins) | | |
|---|---|---|---|---|
| | | $Input\ Length = 10$ | $Input\ Length = 20$ | $Input\ Length = 50$ |
| ESE | | 0.20 / 0.25 / 0.29 | 0.20 / 0.24 / 0.33 | 0.21 / 0.24 / 0.35 |
| ARIMA | No ESE | 0.05 / 0.09 / 0.17 | 0.05 / 0.09 / 0.17 | 0.05 / 0.09 / 0.18 |
| | With ESE | 0.21 / 0.26 / 0.29 | 0.21 / 0.24 / 0.34 | 0.22 / 0.25 / 0.36 |
| LSTM | No ESE | 1.27 / 2.56 / 5.12 | 1.27 / 2.55 / 4.93 | 1.24 / 2.38 / 4.74 |
| | With ESE | 0.45 / 0.50 / 0.54 | 0.46 / 0.49 / 0.58 | 0.46 / 0.48 / 0.59 |
| Dlinear | No ESE | 1.40 / 2.80 / 5.55 | 1.46 / 2.99 / 5.91 | 1.41 / 2.87 / 5.96 |
| | With ESE | 0.48 / 0.53 / 0.56 | 0.50 / 0.53 / 0.63 | 0.50 / 0.52 / 0.65 |
| Informer | No ESE | 0.84 / 1.77 / 3.52 | 0.88 / 1.65 / 3.35 | 0.85 / 1.82 / 3.48 |
| | With ESE | 0.37 / 0.42 / 0.46 | 0.38 / 0.40 / 0.50 | 0.38 / 0.42 / 0.52 |
| DeepAR | No ESE | 1.14 / 2.29 / 4.68 | 1.19 / 2.39 / 4.73 | 1.17 / 2.38 / 4.63 |
| | With ESE | 0.43 / 0.47 / 0.52 | 0.44 / 0.48 / 0.57 | 0.45 / 0.47 / 0.58 |
| PatchTST | No ESE | 0.90 / 1.96 / 3.77 | 0.95 / 1.93 / 3.82 | 0.92 / 1.91 / 3.73 |
| | With ESE | 0.38 / 0.44 / 0.47 | 0.39 / 0.43 / 0.52 | 0.40 / 0.43 / 0.54 |

Table 10: Daily COVID-19 Data

| Attribute | Data Type | Comments |
|---|---|---|
| Active cases | numeric | State total/By region; daily |
| NDC | numeric | State total/By region; daily |
| PCR cases | numeric | State total/By region/By age; daily |
| PCR tests | numeric | State total; daily |
| RAT cases | numeric | State total; daily |
| # Hospitalisation | numeric | State total; daily |
| # ICU cases | numeric | State total; daily |
| # On ventilation | numeric | State total; daily |
| # Death | numeric | State total; daily |

Table 11: Attribute Data Describing Local Regions

| Attribute | Data Type | Comments |
|---|---|---|
| V1,V2,V3 | percentiles | weekly |
| Acquired source of cases | text | daily |
| Population | numeric | Collected daily |
| Band (restriction level) | numeric | $[0, 10]$; Daily |
| Medical practitioners | numeric | – |
| Health care and social assistance | numeric | – |
| Private health insurance | numeric | – |
| Age group | numeric | – |
| Other health data | numeric | – |

attribute, we collected the number of medical practitioners, hospital distribution, the number of health care and social assistance by referring to the research of Munga, Yin et al Munga & Mæstad (2009); Yin et al. (2018). In addition, we collected demographic data of regions, e.g. age distribution, as prior studies have shown that there is a strong correlation between age and COVID-19, risk increasing significantly with age Li et al. (2020). "*Other health data*" in the last row includes additional relevant data of the region, e.g. the rates of obesity, hypertension and chronic diseases, as many studies have pointed out that COVID-19 infection is connected with these health problems, obesity Rychter et al. (2020); Popkin et al. (2020) and chronic diseases Fang et al. (2020); Laires et al. (2021). Other non-major data include economic data, emergencies (such as activity gathering) and government policies (such as lockdowns imposed). These data are related to the epidemic situation and cannot be ignored during the model testing.

## H PREPROCESSING OF RAT CASES

The daily figure of RAT data published before February 4 2022 lacks regional information. There-fore, we preprocessed the RAT data before that date, by modifying the definition of "*Close Contact*" associated with RAT data. In addition, on December 30, 2021, the Australian Federal Government redefined "*Close Contact*" from 15 minutes to 4 hours. As a result, cases that would be classified as "*Unknown Sources*" by the early definition are now classified as "*Close Contact*". So the number of "*Close Contact*" increases significantly. To address the above issues, RAT data are processed by arccotangent normalisation and transformation.

### H.1 ARCCOTANGENT NORMALISATION (ACN)

The purpose of this normalisation is to unify the data collected before and after February 4 2022 into the same distribution, ranging from 0 to 1, through arccotangent formulation:

$$ACN(x,y) = (\frac{2}{\pi} arccot \ x)^y \quad (x \geq 0) \tag{25}$$

where $x$ represents the number of new cases added daily of which the cause can be either "*Close Contact*" or "*Unknown Sources*". Parameter $y$ is the degree of normalisation in the transformation process. By this formula, $x$ values can be converted into $ACN(x,y)$. The value of $y$ can be obtained by the following:

$$\hat{y}_t = log_{(\frac{2}{\pi}arccot \ x_{t,15mins})}(\frac{x_{t,15mins}}{T_t}) \tag{26}$$

$$y = \frac{\sum_{i=1}^{t} \hat{y}_i}{t} \tag{27}$$

where $\hat{y}_t$ is the degree of normalisation for $t$ day, $x_{t,15mins}$ is the number of new cases caused by "*Close Contact*" and "*Unknown Sources*" at $t$ day when the definition is 15 mins. $T_t$ represents the total daily increase of all "*Close Contact*" cases at $t$ day. By aggregating all $\hat{y}_t$, we can obtain the average as $y$. With the above formulae, $ACN(x,y)$ will always be 1 if the policy of close contact is set to 0. That means that all cases are "*Close Contact*", and there is no case of "*Unknown Sources*". If the contact time is set bigger, more cases will be in the category of "*Unknown Sources*".

### H.2 ACN BASED TRANSFORMATION

The above ACN normalization ensures consistency in handling different definitions of close contact. Another source of inconsistency is the region information, which can be dealt with by ACN as well. To reduce the bias in RAT in regions, the regional RAT numbers can be computed as follows using ACN:

$$RAT_i = RAT_{total} \times \frac{ACN_i}{\sum_{j=1}^{n} ACN_j} \tag{28}$$

where $RAT_i$ is the number of RAT cases in region $i$; $ACN_i$ is the arccotangent normalization value of that region; $\sum_{j=1}^{n} ACN_j$ is the total $ACN$ of all regions. With $ACN$, we can effectively eliminate the problem of attribute change and obtain an estimated number of close contact $CC_i$ (transformed cases caused by "*Close Contact*" for region $i$) by the following formula:

$$CC_i = \begin{cases} \theta, & \theta \geq OCC + US \\ OCC, & \theta < OCC + US \end{cases} \quad \theta = \frac{OCC}{ACN(x,y)} \tag{29}$$

where $OCC_i$ and $US_i$ are the official figures of daily cases of "*Close Contact*" and "*Unknown Sources*" for region $i$ respectively. When $\theta$ is less than the sum of $OCC_i$ and $US_i$, it indicates that

$OCC_i$ is not beyond the reasonable range ($CC_i$ will be equal to $OCC_i$). If not, $CC_i$ will be equal to $\theta$. According to the $CC$ from different regions, RAT can be allocated as below:

$$RAT_i = RAT_{total} \times \frac{CC_i}{\sum_{i=1}^{n} CC_i} \tag{30}$$

where $RAT_{total}$ is the total of all daily RAT cases. $RAT_i$ is the cases that are assigned to region $i$. $CC_i$ is the estimated number of close contacts in region $i$.

## I  ANALYSIS OF EQUILIBRIUM STATE EVALUATION

ESE is first evaluated on the Equilibrium Index (EI), to validate its effectiveness in revealing patterns of COVID-19 spreading and associated events. When the Equilibrium Index (EI) is closer to zero and stabilises, that means the epidemic state $\mathcal{ST}$ is closer to the estimated value of the equilibrium state $\mathcal{ES}$, meaning the COVID-19 case distribution in Victoria is stabilized, because of certain influence in these regions, e.g. medical competency. If the EI is not stable and is approaching a value of 1, it indicates that the regional attributes are possibly not the main contributor, but some external factors, affecting the number of cases, e.g. protests and other public gathering events.

According to the EI values during the COVID-19 pandemic period in Victoria, they obviously fluctuated before January 3, 2022, and around May 22, 2022. That reconciles with the news report as there are large public gatherings occurred during both periods, the New Year celebrations held in many regions before Jan/3/22, and the election held from May/19/22, to May/23/22.

## J  FULL COMPARISON ON COVID-19 DATA

### J.1  PREDICTION PERFORMANCE ON COVID-19 DATA

Table 12: Comparing prediction performance (output step is 1) with 12 SOTA methods, in RMSE, MAE, and DILATE, with no ESE and with ESE, with input of 10, 20, 50 and 100 steps, for 20 large regions / 79 regions / 320 sub-regions.

| Models | | Metric | Predicting 1 Step | | | |
|---|---|---|---|---|---|---|
| | | | *Input Length = 10* | *Input Length = 20* | *Input Length = 50* | *Input Length = 100* |
| ESE | – | RMSE | 53.69 / 57.99 / 43.47 | 52.47 / 54.99 / 47.67 | 62.16 / 54.52 / 47.34 | 84.54 / 55.54 / 48.32 |
| | | MAE | 51.64 / 52.27 / 35.43 | 51.60 / 49.86 / 44.31 | 51.34 / 49.94 / 45.67 | 83.91 / 50.85 / 45.86 |
| | | DILATE | 96.89 / 102.42 / 79.26 | 83.45 / 95.21 / 72.10 | 77.64 / 94.52 / 78.24 | 106.96 / 96.58 / 79.64 |
| VAR | – | RMSE | 45.29 / 63.94 / 67.66 | 54.61 / 78.92 / 84.90 | 77.19 / 84.94 / 89.09 | 95.79 / 90.85 / 80.76 |
| | | MAE | 38.48 / 56.16 / 59.63 | 53.38 / 70.90 / 81.75 | 73.55 / 82.26 / 83.82 | 90.85 / 81.74 / 76.83 |
| | | DILATE | 80.59 / 104.62 / 107.49 | 87.75 / 110.31 / 111.95 | 110.48 / 118.93 / 119.60 | 122.35 / 126.52 / 114.46 |
| ARIMA | No ESE | RMSE | 18.49 / 57.69 / 54.45 | 39.44 / 70.94 / 73.41 | 69.84 / 72.56 / 76.42 | 79.70 / 77.72 / 74.41 |
| | | MAE | 17.47 / 49.22 / 50.23 | 37.64 / 64.31 / 62.45 | 68.05 / 66.87 / 65.44 | 79.59 / 67.41 / 62.41 |
| | | DILATE | 28.44 / 96.29 / 84.32 | 59.24 / 99.37 / 86.74 | 82.41 / 102.43 / 92.31 | 109.91 / 105.45 / 93.46 |
| | With ESE | RMSE | 49.34 / 56.32 / 43.81 | 50.12 / 54.28 / 46.61 | 61.34 / 55.46 / 48.97 | 75.31 / 54.15 / 47.46 |
| | | MAE | 48.41 / 51.21 / 34.67 | 49.16 / 49.12 / 43.91 | 50.34 / 50.45 / 44.74 | 73.64 / 50.65 / 44.19 |
| | | DILATE | 83.12 / 98.47 / 85.73 | 79.44 / 95.24 / 71.76 | 87.03 / 93.56 / 74.64 | 105.48 / 96.49 / 75.36 |
| LSTM | No ESE | RMSE | 16.41 / 46.60 / 45.14 | 37.01 / 59.01 / 54.33 | 57.69 / 60.83 / 55.47 | 78.45 / 61.42 / 62.30 |
| | | MAE | 15.01 / 40.02 / 35.22 | 35.96 / 50.68 / 47.49 | 47.64 / 55.87 / 52.90 | 74.96 / 57.26 / 54.78 |
| | | DILATE | 28.21 / 83.14 / 74.33 | 38.36 / 89.78 / 82.47 | 79.99 / 93.01 / 81.25 | 101.65 / 98.12 / 89.32 |
| | With ESE | RMSE | 48.01 / 56.62 / 43.04 | 48.49 / 56.31 / 45.42 | 60.20 / 55.77 / 47.31 | 72.79 / 55.42 / 46.82 |
| | | MAE | 44.15 / 51.12 / 33.49 | 44.54 / 50.32 / 42.41 | 47.37 / 50.47 / 42.48 | 69.47 / 49.93 / 43.76 |
| | | DILATE | 80.32 / 86.12 / 74.97 | 75.11 / 87.49 / 68.31 | 83.92 / 91.45 / 73.91 | 99.61 / 96.41 / 74.15 |
| Dlinear | No ESE | RMSE | 16.34 / 50.09 / 44.32 | 36.31 / 54.55 / 55.31 | 57.32 / 55.15 / 51.32 | 79.47 / 56.16 / 59.31 |
| | | MAE | 14.96 / 46.04 / 36.14 | 34.78 / 53.78 / 45.39 | 49.63 / 52.95 / 48.39 | 72.41 / 53.95 / 51.04 |
| | | DILATE | 27.33 / 79.45 / 79.41 | 57.45 / 86.14 / 79.47 | 80.41 / 94.24 / 82.43 | 102.47 / 97.22 / 87.11 |
| | With ESE | RMSE | 45.47 / 50.23 / 42.41 | 46.97 / 51.94 / 44.74 | 58.13 / 53.44 / 47.06 | 73.61 / 56.14 / 45.15 |
| | | MAE | 42.74 / 48.19 / 31.64 | 42.17 / 51.63 / 41.46 | 46.82 / 50.42 / 43.60 | 100.94 / 50.12 / 73.41 |
| | | DILATE | 79.12 / 90.23 / 73.47 | 74.62 / 91.48 / 69.15 | 71.91 / 93.33 / 72.47 | 100.94 / 96.32 / 73.41 |
| Nlinear | No ESE | RMSE | 15.01 / 43.28 / 41.32 | 34.78 / 52.69 / 53.47 | 56.74 / 54.22 / 49.74 | 74.96 / 55.62 / 54.77 |
| | | MAE | 14.90 / 43.49 / 32.47 | 33.69 / 51.53 / 45.25 | 47.41 / 51.95 / 48.01 | 70.77 / 52.23 / 49.96 |
| | | DILATE | 27.03 / 86.15 / 70.01 | 54.77 / 89.44 / 77.47 | 78.84 / 91.45 / 77.31 | 96.93 / 97.15 / 85.47 |
| | With ESE | RMSE | 45.23 / 48.32 / 41.64 | 46.03 / 51.64 / 43.94 | 58.14 / 55.01 / 47.13 | 71.64 / 55.14 / 44.75 |
| | | MAE | 40.94 / 44.31 / 30.91 | 41.31 / 50.78 / 40.19 | 45.84 / 50.34 / 42.45 | 69.98 / 51.33 / 41.86 |
| | | DILATE | 78.44 / 94.21 / 74.61 | 72.49 / 92.31 / 68.91 | 70.45 / 93.66 / 72.04 | 98.30 / 95.25 / 73.07 |
| Informer | No ESE | RMSE | 17.03 / 58.96 / 48.65 | 38.99 / 59.85 / 60.74 | 58.31 / 61.23 / 57.14 | 76.84 / 62.01 / 60.52 |
| | | MAE | 14.99 / 47.31 / 39.41 | 35.69 / 53.78 / 50.64 | 46.72 / 58.85 / 52.56 | 71.79 / 60.57 / 58.03 |
| | | DILATE | 28.64 / 89.41 / 81.78 | 58.41 / 91.44 / 83.22 | 79.45 / 95.31 / 83.50 | 99.39 / 99.85 / 89.41 |
| | With ESE | RMSE | 46.25 / 60.32 / 43.16 | 47.34 / 56.33 / 44.86 | 59.42 / 55.17 / 48.01 | 73.43 / 55.48 / 46.04 |
| | | MAE | 43.32 / 51.37 / 31.94 | 43.44 / 50.98 / 42.34 | 48.83 / 49.47 / 44.06 | 71.48 / 49.23 / 43.71 |
| | | DILATE | 79.99 / 94.49 / 73.71 | 76.74 / 94.43 / 70.06 | 72.14 / 94.77 / 72.94 | 99.43 / 94.61 / 75.62 |
| FiLM | No ESE | RMSE | 16.54 / 52.90 / 45.36 | 35.66 / 53.32 / 50.31 | 55.33 / 55.57 / 50.98 | 76.84 / 56.21 / 57.41 |
| | | MAE | 13.85 / 43.21 / 33.45 | 33.58 / 50.66 / 49.44 | 47.45 / 48.60 / 45.79 | 73.85 / 51.34 / 49.63 |
| | | DILATE | 29.12 / 90.14 / 78.64 | 57.40 / 92.45 / 84.67 | 83.64 / 95.14 / 80.77 | 98.47 / 102.36 / 93.11 |
| | With ESE | RMSE | 45.94 / 53.14 / 42.84 | 44.17 / 52.96 / 43.74 | 57.93 / 55.31 / 46.94 | 71.68 / 54.96 / 43.61 |
| | | MAE | 42.45 / 51.67 / 30.94 | 41.96 / 51.14 / 41.49 | 45.83 / 49.66 / 43.22 | 69.15 / 49.01 / 39.63 |
| | | DILATE | 80.31 / 92.11 / 71.86 | 74.14 / 94.36 / 69.61 | 69.73 / 95.54 / 72.12 | 97.74 / 96.49 / 74.64 |
| SCINet | No ESE | RMSE | 15.44 / 50.64 / 46.00 | 35.98 / 57.45 / 56.31 | 58.94 / 59.80 / 60.33 | 79.75 / 55.86 / 59.43 |
| | | MAE | 13.90 / 44.70 / 33.32 | 32.23 / 50.48 / 51.32 | 54.79 / 51.31 / 55.74 | 69.94 / 53.56 / 55.94 |
| | | DILATE | 26.49 / 86.19 / 79.68 | 54.34 / 88.35 / 88.12 | 81.74 / 95.16 / 88.12 | 97.25 / 98.41 / 90.07 |
| | With ESE | RMSE | 44.67 / 53.01 / 42.44 | 45.49 / 52.25 / 42.09 | 58.88 / 54.12 / 46.34 | 71.84 / 54.04 / 44.65 |
| | | MAE | 40.95 / 51.18 / 31.66 | 42.79 / 47.11 / 41.84 | 46.73 / 48.14 / 41.64 | 69.71 / 50.34 / 42.41 |
| | | DILATE | 77.14 / 92.44 / 72.36 | 73.94 / 91.21 / 70.63 | 71.06 / 92.12 / 72.36 | 98.71 / 93.23 / 70.94 |
| DeepAR | No ESE | RMSE | 17.63 / 51.12 / 50.11 | 39.01 / 53.48 / 57.41 | 61.78 / 54.64 / 61.74 | 83.41 / 56.34 / 62.98 |
| | | MAE | 16.41 / 47.32 / 39.41 | 34.68 / 47.54 / 49.37 | 50.74 / 50.31 / 56.41 | 74.65 / 54.32 / 50.40 |
| | | DILATE | 28.68 / 88.21 / 80.18 | 58.31 / 92.47 / 79.94 | 81.74 / 95.65 / 85.93 | 101.98 / 99.42 / 87.77 |
| | With ESE | RMSE | 47.44 / 51.95 / 43.06 | 48.41 / 52.47 / 45.74 | 60.03 / 52.43 / 48.52 | 74.82 / 50.99 / 47.92 |
| | | MAE | 46.41 / 50.33 / 33.82 | 46.94 / 49.16 / 42.93 | 48.34 / 51.02 / 43.82 | 72.91 / 50.96 / 44.61 |
| | | DILATE | 82.94 / 93.57 / 82.79 | 77.16 / 92.62 / 70.74 | 75.06 / 95.14 / 73.64 | 100.74 / 95.01 / 73.46 |
| KVAE | No ESE | RMSE | 16.21 / 48.69 / 43.96 | 35.14 / 48.14 / 47.92 | 54.36 / 52.41 / 49.74 | 70.85 / 53.12 / 52.93 |
| | | MAE | 13.43 / 47.72 / 42.66 | 31.47 / 47.23 / 45.34 | 45.96 / 50.34 / 41.90 | 64.44 / 52.71 / 49.47 |
| | | DILATE | 27.30 / 86.34 / 79.94 | 54.12 / 84.36 / 80.11 | 78.71 / 92.14 / 77.06 | 96.03 / 97.12 / 88.81 |
| | With ESE | RMSE | 44.45 / 52.10 / 42.36 | 44.94 / 51.39 / 42.41 | 58.22 / 52.11 / 47.42 | 71.94 / 52.13 / 47.67 |
| | | MAE | 41.47 / 49.02 / 31.82 | 40.46 / 48.94 / 40.76 | 46.56 / 51.32 / 43.77 | 68.41 / 51.24 / 42.66 |
| | | DILATE | 78.87 / 89.22 / 68.18 | 72.44 / 84.30 / 68.49 | 72.41 / 93.03 / 73.92 | 97.43 / 95.41 / 73.81 |
| TPGNN | No ESE | RMSE | 15.96 / 48.31 / 44.33 | 36.97 / 49.21 / 51.49 | 53.74 / 56.65 / 52.31 | 78.37 / 59.22 / 60.63 |
| | | MAE | 15.12 / 44.12 / 38.41 | 34.12 / 48.36 / 50.36 | 48.71 / 54.79 / 45.70 | 69.86 / 58.13 / 54.13 |
| | | DILATE | 25.65 / 84.36 / 75.14 | 54.31 / 89.34 / 82.16 | 83.44 / 94.65 / 79.74 | 99.66 / 97.32 / 95.41 |
| | With ESE | RMSE | 45.12 / 54.29 / 43.24 | 46.41 / 53.29 / 42.75 | 57.65 / 55.16 / 46.82 | 72.24 / 56.95 / 46.83 |
| | | MAE | 43.56 / 50.41 / 33.47 | 43.46 / 51.32 / 42.61 | 46.07 / 52.96 / 43.64 | 70.64 / 54.99 / 42.41 |
| | | DILATE | 79.16 / 93.28 / 72.66 | 74.92 / 94.12 / 70.03 | 71.62 / 93.45 / 72.49 | 98.43 / 94.35 / 73.03 |
| PatchTST | No ESE | RMSE | 15.67 / 49.46 / 44.31 | 36.23 / 53.87 / 52.34 | 55.43 / 54.49 / 50.74 | 70.01 / 56.44 / 51.13 |
| | | MAE | 14.75 / 43.94 / 33.62 | 31.98 / 47.31 / 48.21 | 49.34 / 53.35 / 42.94 | 63.52 / 52.41 / 44.33 |
| | | DILATE | 25.75 / 85.36 / 82.64 | 55.74 / 87.94 / 86.38 | 86.41 / 89.86 / 79.46 | 95.61 / 88.12 / 79.64 |
| | With ESE | RMSE | 45.82 / 58.19 / 42.38 | 46.09 / 54.66 / 43.35 | 59.12 / 52.58 / 46.54 | 70.74 / 53.71 / 45.19 |
| | | MAE | 42.44 / 51.17 / 31.86 | 41.63 / 49.89 / 41.62 | 48.49 / 47.99 / 43.51 | 65.97 / 47.25 / 43.69 |
| | | DILATE | 79.59 / 90.89 / 73.33 | 76.63 / 91.10 / 77.25 | 79.54 / 92.19 / 79.97 | 98.35 / 91.32 / 72.00 |

Table 13: Comparing prediction performance (output step is 2) with 12 SOTA methods, in RMSE, MAE, and DILATE, with no ESE and with ESE, with input of 10, 20, 50 and 100 steps, for 20 large regions / 79 regions / 320 sub-regions.

| Models | | Metric | Predicting 2 Steps | | | |
|---|---|---|---|---|---|---|
| | | | *Input Length = 10* | *Input Length = 20* | *Input Length = 50* | *Input Length = 100* |
| ESE | – | RMSE | 58.50 / 62.19 / 46.61 | 55.73 / 59.51 / 51.22 | 67.85 / 59.17 / 50.17 | 92.19 / 59.72 / 52.43 |
| | | MAE | 53.96 / 54.90 / 36.65 | 53.41 / 52.00 / 46.19 | 53.54 / 52.05 / 48.04 | 88.29 / 53.45 / 47.84 |
| | | DILATE | 105.51 / 112.01 / 87.09 | 90.53 / 103.91 / 78.72 | 89.23 / 103.42 / 85.84 | 126.82 / 104.73 / 86.14 |
| VAR | – | RMSE | 46.57 / 65.82 / 70.93 | 54.84 / 81.39 / 87.14 | 77.70 / 86.81 / 92.54 | 95.86 / 93.56 / 84.17 |
| | | MAE | 39.12 / 58.72 / 62.09 | 49.67 / 72.96 / 83.32 | 76.63 / 82.57 / 86.08 | 91.64 / 83.38 / 78.35 |
| | | DILATE | 83.85 / 105.10 / 109.76 | 79.84 / 110.95 / 116.22 | 111.45 / 121.99 / 120.35 | 124.02 / 132.47 / 115.93 |
| ARIMA | No ESE | RMSE | 19.20 / 59.31 / 56.02 | 40.38 / 73.22 / 75.78 | 71.86 / 74.51 / 78.86 | 82.07 / 80.34 / 77.21 |
| | | MAE | 18.83 / 53.64 / 54.90 | 39.82 / 69.65 / 68.21 | 64.35 / 72.12 / 70.71 | 86.79 / 73.38 / 67.76 |
| | | DILATE | 29.42 / 98.99 / 87.28 | 60.50 / 102.47 / 89.55 | 90.59 / 104.65 / 95.49 | 128.26 / 109.63 / 96.81 |
| | With ESE | RMSE | 53.74 / 59.54 / 46.13 | 52.55 / 57.58 / 48.53 | 66.92 / 58.84 / 51.62 | 80.58 / 57.66 / 50.78 |
| | | MAE | 51.61 / 53.90 / 34.84 | 50.87 / 49.13 / 44.64 | 55.33 / 50.68 / 48.38 | 75.28 / 54.36 / 47.19 |
| | | DILATE | 87.47 / 103.77 / 88.32 | 85.97 / 98.65 / 76.96 | 89.30 / 98.09 / 81.45 | 125.52 / 104.54 / 77.01 |
| LSTM | No ESE | RMSE | 17.87 / 50.30 / 49.06 | 39.20 / 64.00 / 58.71 | 62.49 / 65.48 / 59.84 | 84.85 / 66.73 / 67.75 |
| | | MAE | 15.52 / 41.36 / 36.44 | 36.76 / 52.00 / 48.31 | 49.20 / 57.05 / 54.60 | 76.35 / 59.12 / 56.22 |
| | | DILATE | 30.16 / 88.10 / 79.59 | 60.40 / 95.40 / 88.18 | 84.65 / 97.90 / 87.13 | 117.93 / 104.48 / 94.72 |
| | With ESE | RMSE | 51.24 / 56.83 / 45.04 | 48.98 / 59.34 / 47.21 | 49.92 / 54.21 / 44.75 | 70.29 / 53.19 / 44.24 |
| | | MAE | 47.15 / 54.57 / 35.19 | 45.34 / 52.11 / 44.38 | 49.92 / 54.21 / 44.75 | 70.29 / 53.19 / 44.24 |
| | | DILATE | 81.54 / 91.29 / 81.20 | 76.16 / 93.86 / 72.77 | 86.91 / 94.14 / 76.62 | 114.91 / 98.61 / 75.61 |
| Dlinear | No ESE | RMSE | 16.65 / 50.90 / 45.49 | 37.07 / 55.80 / 56.81 | 58.62 / 56.70 / 52.47 | 82.09 / 57.09 / 61.21 |
| | | MAE | 15.02 / 46.35 / 36.14 | 34.69 / 53.84 / 45.46 | 49.87 / 53.00 / 48.64 | 72.86 / 53.81 / 51.35 |
| | | DILATE | 29.56 / 85.20 / 85.07 | 60.15 / 93.12 / 85.95 | 87.59 / 102.51 / 88.80 | 117.78 / 105.72 / 93.92 |
| | With ESE | RMSE | 46.21 / 54.20 / 42.85 | 51.02 / 55.62 / 44.87 | 61.34 / 54.35 / 51.70 | 77.71 / 58.76 / 48.07 |
| | | MAE | 43.24 / 51.23 / 33.58 | 44.86 / 52.87 / 43.40 | 47.45 / 54.08 / 47.28 | 75.45 / 53.52 / 44.89 |
| | | DILATE | 82.73 / 94.48 / 76.79 | 80.49 / 91.78 / 72.14 | 88.09 / 94.13 / 73.67 | 120.97 / 103.71 / 75.08 |
| Nlinear | No ESE | RMSE | 16.50 / 47.48 / 45.25 | 37.26 / 57.54 / 58.29 | 62.28 / 59.61 / 54.79 | 81.63 / 60.84 / 59.61 |
| | | MAE | 15.60 / 44.73 / 33.78 | 34.49 / 53.75 / 47.27 | 49.49 / 53.92 / 50.45 | 74.01 / 54.71 / 52.21 |
| | | DILATE | 28.56 / 90.99 / 73.88 | 56.78 / 95.13 / 83.10 | 83.67 / 97.59 / 82.96 | 115.38 / 103.34 / 90.87 |
| | With ESE | RMSE | 45.37 / 51.11 / 41.95 | 50.56 / 53.65 / 47.86 | 62.34 / 57.70 / 47.38 | 76.42 / 59.77 / 47.25 |
| | | MAE | 43.52 / 45.64 / 32.88 | 45.28 / 53.05 / 41.36 | 48.21 / 52.46 / 46.50 | 71.59 / 52.19 / 44.59 |
| | | DILATE | 86.25 / 103.28 / 75.28 | 74.02 / 96.31 / 69.76 | 81.53 / 98.09 / 75.43 | 115.54 / 96.51 / 74.23 |
| Informer | No ESE | RMSE | 17.73 / 60.39 / 50.08 | 40.10 / 61.77 / 63.19 | 60.27 / 63.62 / 59.02 | 78.13 / 64.41 / 63.03 |
| | | MAE | 16.17 / 50.50 / 42.41 | 37.39 / 57.72 / 54.03 | 49.91 / 63.41 / 56.20 | 77.42 / 65.27 / 62.75 |
| | | DILATE | 30.94 / 95.60 / 87.27 | 61.64 / 98.45 / 90.38 | 85.69 / 103.20 / 89.42 | 117.83 / 108.05 / 96.20 |
| | With ESE | RMSE | 47.22 / 63.37 / 47.28 | 48.11 / 61.71 / 48.50 | 64.29 / 58.86 / 49.34 | 76.45 / 55.93 / 46.29 |
| | | MAE | 46.57 / 53.57 / 32.89 | 46.05 / 51.53 / 46.37 | 50.88 / 50.03 / 44.22 | 73.77 / 50.42 / 44.03 |
| | | DILATE | 82.67 / 98.44 / 74.16 | 84.29 / 102.09 / 73.43 | 72.97 / 97.78 / 78.65 | 118.20 / 96.74 / 77.62 |
| FiLM | No ESE | RMSE | 17.87 / 57.78 / 49.56 | 37.75 / 57.75 / 54.54 | 60.11 / 60.73 / 55.27 | 83.54 / 61.18 / 62.15 |
| | | MAE | 14.19 / 43.95 / 34.08 | 34.01 / 51.97 / 50.91 | 48.83 / 50.10 / 46.54 | 76.50 / 53.08 / 50.75 |
| | | DILATE | 30.26 / 93.18 / 81.28 | 59.24 / 96.12 / 88.30 | 87.07 / 99.20 / 84.22 | 106.97 / 106.26 / 96.41 |
| | With ESE | RMSE | 49.00 / 56.48 / 45.07 | 45.53 / 56.94 / 46.26 | 61.44 / 58.91 / 48.20 | 72.65 / 56.50 / 44.59 |
| | | MAE | 45.67 / 53.86 / 32.26 | 45.24 / 52.09 / 44.27 | 47.91 / 52.37 / 43.70 | 69.15 / 49.64 / 40.36 |
| | | DILATE | 82.42 / 98.78 / 73.67 | 81.53 / 94.80 / 75.79 | 83.87 / 104.05 / 75.12 | 107.99 / 102.68 / 78.01 |
| SCINet | No ESE | RMSE | 15.86 / 51.31 / 46.95 | 36.66 / 58.76 / 57.43 | 60.74 / 61.34 / 61.34 | 82.18 / 56.96 / 60.66 |
| | | MAE | 13.87 / 44.53 / 33.33 | 31.96 / 50.10 / 51.00 | 54.91 / 50.48 / 55.78 | 69.80 / 53.10 / 55.05 |
| | | DILATE | 27.98 / 91.29 / 84.32 | 56.54 / 94.15 / 94.46 | 89.02 / 101.92 / 91.47 | 114.90 / 105.33 / 96.07 |
| | With ESE | RMSE | 47.72 / 57.83 / 42.89 | 48.72 / 53.33 / 45.18 | 61.37 / 57.08 / 46.61 | 76.69 / 56.47 / 48.17 |
| | | MAE | 41.42 / 54.89 / 34.51 | 46.17 / 50.55 / 44.00 | 47.36 / 50.44 / 43.43 | 73.07 / 53.05 / 43.90 |
| | | DILATE | 83.78 / 99.32 / 76.39 | 81.23 / 95.22 / 70.73 | 87.57 / 92.89 / 75.12 | 111.99 / 99.73 / 72.44 |
| DeepAR | No ESE | RMSE | 19.68 / 57.99 / 56.29 | 42.85 / 60.17 / 64.10 | 69.39 / 61.15 / 69.41 | 94.20 / 66.58 / 63.28 |
| | | MAE | 17.15 / 54.46 / 48.29 | 40.13 / 58.14 / 59.90 | 62.15 / 56.95 / 62.41 | 91.80 / 63.03 / 61.37 |
| | | DILATE | 30.23 / 91.78 / 84.15 | 59.72 / 96.74 / 83.95 | 85.06 / 100.94 / 89.84 | 116.23 / 104.47 / 91.37 |
| | With ESE | RMSE | 49.47 / 55.85 / 46.53 | 48.83 / 52.82 / 49.17 | 61.54 / 57.59 / 50.09 | 81.55 / 54.68 / 50.88 |
| | | MAE | 47.63 / 54.90 / 36.61 | 47.12 / 51.78 / 43.46 | 51.49 / 51.40 / 44.52 | 76.76 / 52.76 / 48.56 |
| | | DILATE | 83.95 / 101.69 / 88.86 | 83.03 / 99.02 / 77.81 | 89.79 / 98.81 / 80.83 | 118.40 / 97.05 / 77.29 |
| KVAE | No ESE | RMSE | 16.91 / 48.68 / 45.58 | 39.18 / 56.30 / 55.93 | 51.80 / 57.25 / 56.06 | 85.97 / 60.00 / 59.32 |
| | | MAE | 15.95 / 45.77 / 42.24 | 34.55 / 53.34 / 51.13 | 48.42 / 53.14 / 53.22 | 76.52 / 58.67 / 58.22 |
| | | DILATE | 27.59 / 87.65 / 80.96 | 58.89 / 99.71 / 88.87 | 83.11 / 102.26 / 88.70 | 112.80 / 105.37 / 97.34 |
| | With ESE | RMSE | 45.36 / 54.48 / 43.27 | 48.52 / 56.17 / 46.09 | 61.11 / 52.72 / 48.58 | 76.96 / 54.17 / 50.50 |
| | | MAE | 44.62 / 49.37 / 32.53 | 41.34 / 49.19 / 42.30 | 47.84 / 55.87 / 46.29 | 70.77 / 55.48 / 45.98 |
| | | DILATE | 83.61 / 95.33 / 70.79 | 76.01 / 89.36 / 72.56 | 85.53 / 95.88 / 75.52 | 109.72 / 96.12 / 74.21 |
| TPGNN | No ESE | RMSE | 19.71 / 59.24 / 54.31 | 43.37 / 60.64 / 63.15 | 66.11 / 69.84 / 64.09 | 96.12 / 72.72 / 75.03 |
| | | MAE | 17.46 / 50.69 / 44.70 | 37.81 / 55.92 / 58.52 | 56.32 / 62.84 / 52.45 | 81.05 / 67.01 / 63.06 |
| | | DILATE | 28.63 / 94.41 / 84.10 | 58.36 / 100.61 / 92.43 | 83.60 / 107.48 / 89.24 | 117.49 / 110.18 / 107.79 |
| | With ESE | RMSE | 47.09 / 54.99 / 46.02 | 46.68 / 55.39 / 45.55 | 59.55 / 56.25 / 50.70 | 74.76 / 59.39 / 50.73 |
| | | MAE | 44.40 / 53.31 / 34.06 | 43.85 / 54.64 / 45.91 | 50.07 / 57.37 / 46.14 | 73.94 / 59.78 / 43.49 |
| | | DILATE | 82.36 / 96.78 / 74.55 | 81.10 / 97.30 / 71.59 | 85.64 / 94.68 / 79.46 | 119.25 / 101.71 / 78.85 |
| PatchTST | No ESE | RMSE | 16.55 / 52.02 / 45.27 | 36.89 / 57.93 / 56.49 | 56.80 / 58.07 / 52.80 | 72.20 / 61.24 / 53.23 |
| | | MAE | 14.95 / 44.68 / 35.47 | 33.67 / 51.07 / 49.29 | 52.14 / 55.93 / 46.58 | 64.19 / 56.65 / 48.17 |
| | | DILATE | 26.93 / 93.03 / 90.69 | 57.86 / 96.17 / 93.91 | 91.49 / 98.03 / 87.02 | 104.55 / 95.92 / 79.78 |
| | With ESE | RMSE | 46.59 / 61.93 / 44.12 | 46.38 / 57.12 / 45.92 | 60.21 / 52.82 / 49.49 | 71.91 / 56.34 / 48.34 |
| | | MAE | 45.38 / 54.72 / 33.20 | 43.20 / 50.77 / 44.42 | 50.34 / 51.33 / 44.50 | 70.28 / 48.84 / 44.26 |
| | | DILATE | 82.96 / 91.08 / 77.14 | 76.90 / 93.47 / 71.61 | 69.78 / 93.01 / 70.23 | 106.88 / 97.45 / 74.96 |

Table 14: Comparing prediction performance (output step is 5) with 12 SOTA methods, in RMSE, MAE, and DILATE, with no ESE and with ESE, with input of 10, 20, 50 and 100 steps, for 20 large regions / 79 regions / 320 sub-regions.

| Models | | Metric | Predicting 5 Steps | | | |
|---|---|---|---|---|---|---|
| | | | *Input Length = 10* | *Input Length = 20* | *Input Length = 50* | *Input Length = 100* |
| ESE | – | RMSE | 60.16 / 64.95 / 48.49 | 62.71 / 61.43 / 53.60 | 69.56 / 61.56 / 53.12 | 94.74 / 61.91 / 58.99 |
| | | MAE | 57.09 / 57.70 / 39.09 | 56.03 / 54.75 / 49.07 | 56.79 / 54.99 / 50.29 | 92.14 / 55.90 / 50.45 |
| | | DILATE | 107.75 / 113.92 / 88.10 | 89.80 / 106.04 / 80.29 | 86.42 / 105.95 / 86.71 | 130.64 / 107.85 / 88.45 |
| VAR | – | RMSE | 51.87 / 70.96 / 81.32 | 66.86 / 88.04 / 99.20 | 93.86 / 85.82 / 84.76 | 114.49 / 98.09 / 95.73 |
| | | MAE | 46.98 / 63.25 / 66.53 | 65.47 / 82.92 / 104.16 | 93.13 / 81.51 / 78.58 | 103.24 / 91.26 / 87.00 |
| | | DILATE | 97.17 / 123.18 / 130.30 | 91.27 / 145.75 / 135.62 | 124.07 / 144.26 / 137.58 | 148.79 / 131.93 / 129.79 |
| ARIMA | No ESE | RMSE | 24.38 / 63.88 / 60.37 | 48.50 / 78.15 / 81.15 | 76.90 / 78.30 / 84.06 | 93.27 / 85.53 / 82.38 |
| | | MAE | 22.84 / 57.97 / 59.07 | 42.39 / 75.75 / 73.31 | 69.98 / 75.64 / 76.82 | 88.08 / 79.06 / 73.57 |
| | | DILATE | 37.08 / 104.36 / 91.87 | 62.47 / 107.70 / 93.58 | 89.38 / 107.68 / 99.58 | 135.17 / 113.86 / 101.01 |
| | With ESE | RMSE | 57.41 / 62.42 / 48.39 | 59.51 / 59.68 / 56.35 | 67.52 / 61.44 / 54.23 | 86.86 / 59.99 / 54.60 |
| | | MAE | 54.11 / 60.27 / 39.88 | 57.98 / 57.56 / 50.66 | 55.01 / 53.16 / 52.50 | 82.92 / 54.38 / 52.06 |
| | | DILATE | 89.91 / 106.26 / 92.53 | 86.16 / 103.00 / 77.98 | 93.63 / 101.09 / 80.67 | 125.67 / 106.24 / 81.37 |
| LSTM | No ESE | RMSE | 21.92 / 56.56 / 54.83 | 42.85 / 71.87 / 66.10 | 70.11 / 71.72 / 67.57 | 95.02 / 74.49 / 75.65 |
| | | MAE | 21.47 / 47.61 / 42.15 | 41.09 / 60.44 / 56.65 | 57.12 / 60.51 / 63.18 | 89.23 / 68.40 / 65.66 |
| | | DILATE | 39.37 / 97.22 / 86.64 | 64.60 / 104.60 / 96.06 | 92.98 / 104.81 / 94.81 | 129.08 / 114.07 / 104.58 |
| | With ESE | RMSE | 52.66 / 59.77 / 42.62 | 53.03 / 58.57 / 49.97 | 65.99 / 61.23 / 52.09 | 79.62 / 55.23 / 51.25 |
| | | MAE | 47.76 / 56.15 / 35.70 | 48.02 / 49.10 / 45.85 | 50.94 / 52.25 / 45.83 | 74.63 / 51.82 / 47.12 |
| | | DILATE | 89.22 / 102.53 / 82.92 | 83.47 / 97.23 / 75.69 | 81.96 / 101.26 / 81.93 | 121.75 / 107.14 / 82.34 |
| Dlinear | No ESE | RMSE | 21.30 / 54.25 / 47.77 | 38.59 / 58.90 / 60.03 | 62.04 / 59.02 / 55.61 | 85.64 / 60.76 / 63.94 |
| | | MAE | 19.13 / 49.10 / 38.36 | 36.42 / 57.32 / 48.33 | 52.72 / 57.35 / 51.72 | 77.40 / 57.57 / 54.49 |
| | | DILATE | 39.08 / 94.53 / 94.62 | 64.75 / 102.68 / 94.66 | 95.47 / 102.64 / 97.96 | 129.27 / 115.71 / 103.64 |
| | With ESE | RMSE | 46.57 / 57.56 / 35.93 | 48.10 / 53.39 / 45.79 | 59.82 / 55.13 / 48.20 | 75.42 / 57.53 / 46.48 |
| | | MAE | 43.42 / 53.05 / 35.93 | 42.79 / 48.28 / 41.81 | 47.38 / 51.21 / 44.01 | 72.50 / 50.94 / 43.07 |
| | | DILATE | 88.89 / 100.75 / 82.24 | 83.88 / 102.32 / 77.69 | 80.61 / 104.85 / 81.08 | 124.76 / 108.01 / 82.25 |
| Nlinear | No ESE | RMSE | 20.57 / 52.70 / 47.29 | 38.35 / 60.04 / 61.00 | 64.50 / 60.14 / 56.52 | 85.26 / 63.34 / 62.41 |
| | | MAE | 20.14 / 49.05 / 36.64 | 36.82 / 58.16 / 51.02 | 53.38 / 58.25 / 54.40 | 80.04 / 59.30 / 56.24 |
| | | DILATE | 38.39 / 102.45 / 83.14 | 61.15 / 105.75 / 91.94 | 93.77 / 105.94 / 91.74 | 129.39 / 115.02 / 101.08 |
| | With ESE | RMSE | 48.95 / 56.44 / 45.24 | 49.87 / 56.19 / 47.84 | 62.88 / 59.55 / 51.07 | 77.96 / 59.57 / 48.55 |
| | | MAE | 44.09 / 51.50 / 36.17 | 44.23 / 52.29 / 44.06 | 49.05 / 51.01 / 45.64 | 74.83 / 54.98 / 44.79 |
| | | DILATE | 88.17 / 100.06 / 84.22 | 81.89 / 103.63 / 77.68 | 79.48 / 105.42 / 80.77 | 122.28 / 107.18 / 82.06 |
| Informer | No ESE | RMSE | 21.50 / 61.88 / 51.13 | 42.77 / 62.70 / 64.58 | 65.07 / 66.77 / 60.89 | 85.99 / 74.67 / 65.38 |
| | | MAE | 18.54 / 54.70 / 45.58 | 38.74 / 58.62 / 57.62 | 58.99 / 64.11 / 58.09 | 80.70 / 70.23 / 64.84 |
| | | DILATE | 40.38 / 104.91 / 96.05 | 65.92 / 107.38 / 97.75 | 93.12 / 107.31 / 98.00 | 129.90 / 116.92 / 105.30 |
| | With ESE | RMSE | 48.42 / 63.24 / 45.22 | 49.62 / 58.99 / 46.85 | 62.07 / 57.98 / 50.24 | 76.76 / 57.94 / 48.15 |
| | | MAE | 42.49 / 55.12 / 36.92 | 50.25 / 49.24 / 43.32 | 53.50 / 53.18 / 46.33 | 72.87 / 56.92 / 41.01 |
| | | DILATE | 89.45 / 105.26 / 82.53 | 85.57 / 105.17 / 77.82 | 80.30 / 105.99 / 81.08 | 121.86 / 105.86 / 84.24 |
| FiLM | No ESE | RMSE | 20.93 / 59.13 / 51.16 | 40.10 / 59.53 / 53.42 | 64.73 / 63.47 / 54.08 | 84.06 / 65.31 / 60.63 |
| | | MAE | 15.56 / 48.48 / 39.28 | 37.15 / 57.59 / 46.14 | 54.60 / 55.46 / 48.04 | 79.11 / 56.94 / 56.54 |
| | | DILATE | 40.35 / 104.17 / 90.42 | 63.17 / 106.24 / 97.18 | 95.98 / 106.41 / 92.64 | 118.73 / 117.78 / 107.06 |
| | With ESE | RMSE | 51.53 / 560.42 / 47.88 | 49.45 / 59.36 / 48.91 | 64.78 / 60.65 / 52.61 | 79.93 / 59.30 / 48.83 |
| | | MAE | 42.02 / 51.25 / 34.69 | 41.40 / 50.63 / 41.21 | 45.37 / 49.18 / 42.69 | 68.45 / 48.58 / 39.17 |
| | | DILATE | 87.82 / 101.15 / 78.58 | 81.00 / 103.10 / 76.02 | 76.29 / 104.45 / 78.66 | 118.06 / 105.15 / 81.96 |
| SCINet | No ESE | RMSE | 19.49 / 53.00 / 48.30 | 37.20 / 60.60 / 59.23 | 62.00 / 60.40 / 63.51 | 83.74 / 68.90 / 62.70 |
| | | MAE | 18.17 / 48.82 / 36.29 | 34.36 / 55.28 / 55.88 | 59.97 / 55.20 / 61.01 | 76.46 / 58.84 / 61.43 |
| | | DILATE | 37.90 / 102.51 / 95.06 | 60.76 / 105.31 / 105.12 | 99.91 / 105.34 / 102.51 | 128.04 / 117.78 / 107.40 |
| | With ESE | RMSE | 44.68 / 62.43 / 42.56 | 45.65 / 52.27 / 43.87 | 58.61 / 54.01 / 46.10 | 76.88 / 53.80 / 44.58 |
| | | MAE | 42.48 / 53.13 / 35.04 | 44.36 / 51.17 / 41.49 | 48.72 / 49.91 / 43.21 | 72.42 / 52.25 / 44.15 |
| | | DILATE | 87.37 / 104.97 / 81.84 | 83.91 / 102.93 / 80.34 | 80.59 / 102.47 / 80.90 | 123.38 / 106.01 / 80.12 |
| DeepAR | No ESE | RMSE | 24.58 / 59.55 / 58.21 | 41.94 / 62.23 / 66.69 | 71.72 / 62.27 / 71.75 | 97.04 / 65.39 / 73.41 |
| | | MAE | 23.25 / 55.89 / 46.39 | 37.32 / 56.11 / 57.98 | 59.70 / 55.99 / 66.32 | 87.61 / 64.01 / 59.54 |
| | | DILATE | 40.88 / 103.98 / 94.83 | 65.35 / 109.42 / 94.79 | 96.47 / 109.37 / 101.93 | 131.92 / 118.05 / 103.56 |
| | With ESE | RMSE | 49.95 / 61.98 / 41.70 | 56.22 / 60.73 / 52.95 | 69.70 / 60.76 / 52.32 | 86.63 / 51.41 / 45.94 |
| | | MAE | 47.26 / 56.17 / 35.51 | 53.23 / 51.97 / 45.55 | 47.74 / 51.30 / 48.36 | 83.60 / 50.11 / 36.72 |
| | | DILATE | 92.13 / 104.24 / 91.59 | 85.63 / 102.58 / 78.75 | 83.22 / 105.45 / 81.48 | 123.03 / 105.87 / 81.25 |
| KVAE | No ESE | RMSE | 21.98 / 52.46 / 51.89 | 41.90 / 60.88 / 60.22 | 56.62 / 61.03 / 62.23 | 93.53 / 64.77 / 65.94 |
| | | MAE | 20.17 / 50.87 / 49.62 | 37.02 / 59.10 / 56.60 | 54.27 / 59.16 / 60.58 | 84.68 / 62.54 / 52.13 |
| | | DILATE | 35.33 / 93.58 / 85.59 | 61.19 / 106.40 / 95.34 | 88.47 / 106.18 / 95.45 | 120.47 / 112.54 / 102.96 |
| | With ESE | RMSE | 52.07 / 60.96 / 49.71 | 52.62 / 59.97 / 49.44 | 67.87 / 58.98 / 53.30 | 84.38 / 59.21 / 48.71 |
| | | MAE | 44.21 / 52.23 / 33.73 | 43.04 / 51.84 / 43.39 | 49.51 / 52.70 / 46.37 | 72.53 / 54.24 / 45.21 |
| | | DILATE | 88.56 / 100.17 / 76.45 | 81.29 / 98.85 / 76.64 | 81.23 / 104.37 / 83.13 | 120.32 / 107.37 / 82.86 |
| TPGNN | No ESE | RMSE | 21.41 / 62.40 / 56.90 | 45.44 / 63.23 / 66.31 | 69.04 / 63.32 / 67.32 | 89.21 / 76.22 / 77.79 |
| | | MAE | 19.95 / 56.68 / 49.04 | 41.60 / 61.99 / 64.36 | 62.44 / 61.91 / 58.32 | 79.55 / 74.43 / 69.30 |
| | | DILATE | 35.64 / 98.12 / 86.80 | 59.77 / 103.86 / 95.59 | 96.84 / 103.64 / 92.38 | 120.44 / 112.81 / 110.70 |
| | With ESE | RMSE | 50.67 / 62.96 / 48.38 | 52.04 / 59.53 / 47.92 | 64.50 / 60.52 / 50.56 | 80.70 / 58.73 / 52.39 |
| | | MAE | 48.43 / 53.09 / 37.32 | 48.24 / 49.39 / 42.62 | 51.16 / 50.10 / 48.69 | 78.51 / 55.16 / 47.31 |
| | | DILATE | 87.19 / 102.52 / 80.03 | 82.89 / 103.45 / 76.21 | 79.21 / 102.67 / 79.81 | 119.71 / 103.75 / 80.17 |
| PatchTST | No ESE | RMSE | 19.05 / 55.33 / 49.41 | 40.10 / 62.56 / 57.38 | 58.25 / 60.75 / 57.12 | 84.02 / 66.97 / 64.18 |
| | | MAE | 15.13 / 44.89 / 46.00 | 35.49 / 50.94 / 51.06 | 52.09 / 57.68 / 49.70 | 79.02 / 57.05 / 59.37 |
| | | DILATE | 31.63 / 97.14 / 96.02 | 60.20 / 94.33 / 99.52 | 88.41 / 103.73 / 95.60 | 112.28 / 100.17 / 91.89 |
| | With ESE | RMSE | 50.60 / 65.59 / 47.53 | 52.01 / 62.50 / 48.28 | 65.69 / 60.34 / 52.03 | 79.39 / 60.65 / 49.87 |
| | | MAE | 46.77 / 56.56 / 35.83 | 47.74 / 55.90 / 45.90 | 54.87 / 54.55 / 49.83 | 73.47 / 52.27 / 42.74 |
| | | DILATE | 88.89 / 103.51 / 82.95 | 47.74 / 55.90 / 45.90 | 54.87 / 54.55 / 49.83 | 73.47 / 52.27 / 42.74 |

Table 15: Comparing prediction performance (output step is 10) with 12 SOTA methods, in RMSE, MAE, and DILATE, with no ESE and with ESE, with input of 10, 20, 50 and 100 steps, for 20 large regions / 79 regions / 320 sub-regions.

| Models | | Metric | Predicting 10 Steps | | | |
|---|---|---|---|---|---|---|
| | | | *Input Length = 10* | *Input Length = 20* | *Input Length = 50* | *Input Length = 100* |
| ESE | – | RMSE | 61.89 / 66.57 / 59.95 | 65.98 / 73.11 / 54.93 | 68.22 / 67.66 / 54.25 | 97.34 / 69.27 / 55.55 |
| | | MAE | 59.83 / 60.47 / 50.82 | 59.86 / 57.53 / 51.11 | 62.32 / 63.77 / 52.69 | 95.02 / 64.62 / 53.01 |
| | | DILATE | 109.63 / 115.76 / 89.91 | 94.54 / 113.89 / 81.56 | 92.52 / 107.09 / 88.78 | 142.99 / 117.61 / 95.13 |
| VAR | – | RMSE | 55.18 / 78.13 / 81.49 | 67.77 / 98.17 / 105.85 | 98.19 / 105.90 / 107.90 | 117.09 / 112.45 / 104.53 |
| | | MAE | 49.42 / 67.56 / 73.40 | 64.52 / 87.39 / 100.95 | 90.99 / 101.47 / 95.13 | 112.44 / 100.95 / 97.97 |
| | | DILATE | 101.81 / 133.21 / 135.42 | 99.51 / 132.47 / 136.48 | 138.84 / 152.18 / 151.91 | 163.89 / 157.76 / 137.59 |
| ARIMA | No ESE | RMSE | 25.08 / 71.01 / 66.89 | 56.28 / 87.47 / 90.50 | 86.38 / 89.27 / 94.50 | 98.31 / 87.48 / 92.08 |
| | | MAE | 24.72 / 63.19 / 64.42 | 55.49 / 82.67 / 80.34 | 87.09 / 85.86 / 83.71 | 102.05 / 82.51 / 79.81 |
| | | DILATE | 51.96 / 112.65 / 98.90 | 85.72 / 116.08 / 101.52 | 96.40 / 119.89 / 108.15 | 145.57 / 116.15 / 109.42 |
| | With ESE | RMSE | 61.57 / 64.49 / 52.11 | 63.92 / 69.95 / 54.40 | 64.62 / 63.32 / 60.57 | 91.88 / 61.79 / 53.72 |
| | | MAE | 56.90 / 60.83 / 49.34 | 57.92 / 60.82 / 51.51 | 59.57 / 59.68 / 52.30 | 89.79 / 60.15 / 51.85 |
| | | DILATE | 104.99 / 110.29 / 85.48 | 87.92 / 106.70 / 78.70 | 85.29 / 104.70 / 81.87 | 136.07 / 107.73 / 88.78 |
| LSTM | No ESE | RMSE | 23.07 / 59.44 / 57.57 | 54.39 / 75.39 / 69.31 | 73.72 / 77.31 / 70.43 | 99.83 / 75.18 / 79.42 |
| | | MAE | 20.95 / 51.13 / 44.84 | 53.15 / 64.56 / 60.39 | 60.61 / 71.12 / 67.17 | 95.77 / 64.53 / 70.03 |
| | | DILATE | 46.68 / 104.21 / 93.28 | 88.35 / 112.67 / 103.65 | 100.42 / 116.65 / 101.80 | 139.72 / 112.94 / 112.58 |
| | With ESE | RMSE | 56.40 / 63.37 / 46.01 | 56.89 / 67.97 / 52.94 | 63.76 / 66.16 / 55.26 | 87.54 / 65.82 / 54.48 |
| | | MAE | 51.04 / 52.21 / 43.76 | 51.92 / 58.83 / 48.93 | 55.10 / 59.23 / 49.08 | 83.31 / 58.74 / 50.84 |
| | | DILATE | 95.68 / 103.38 / 89.25 | 89.22 / 104.74 / 80.69 | 88.10 / 106.01 / 88.33 | 132.19 / 116.61 / 88.56 |
| Dlinear | No ESE | RMSE | 22.04 / 58.39 / 51.97 | 50.66 / 63.52 / 64.61 | 67.14 / 64.59 / 60.22 | 92.78 / 63.74 / 69.04 |
| | | MAE | 19.18 / 58.39 / 51.97 | 50.66 / 63.52 / 64.61 | 67.14 / 64.59 / 60.22 | 92.78 / 63.74 / 69.04 |
| | | DILATE | 45.45 / 99.93 / 99.96 | 87.03 / 108.59 / 99.94 | 101.20 / 118.17 / 103.51 | 136.85 / 118.43 / 109.94 |
| | With ESE | RMSE | 48.06 / 60.83 / 44.70 | 56.87 / 65.78 / 47.22 | 62.70 / 57.45 / 49.95 | 80.94 / 60.88 / 47.74 |
| | | MAE | 44.94 / 54.27 / 41.85 | 49.41 / 55.30 / 43.62 | 49.56 / 54.06 / 45.92 | 78.21 / 53.68 / 44.73 |
| | | DILATE | 94.59 / 108.73 / 87.24 | 88.64 / 109.88 / 81.47 | 85.47 / 105.71 / 85.93 | 134.28 / 116.09 / 87.14 |
| Nlinear | No ESE | RMSE | 21.87 / 52.43 / 49.68 | 49.79 / 63.18 / 64.50 | 68.31 / 64.99 / 59.76 | 89.84 / 63.35 / 66.13 |
| | | MAE | 19.06 / 50.22 / 37.63 | 47.39 / 59.69 / 52.52 | 54.71 / 60.34 / 55.42 | 81.92 / 59.71 / 58.09 |
| | | DILATE | 44.36 / 107.12 / 87.28 | 83.26 / 111.04 / 96.74 | 98.10 / 114.12 / 96.41 | 136.06 / 121.39 / 106.16 |
| | With ESE | RMSE | 49.60 / 57.33 / 44.93 | 50.42 / 62.04 / 47.75 | 64.62 / 61.12/ 51.66 | 80.85 / 61.16 / 48.89 |
| | | MAE | 42.36 / 52.52 / 40.95 | 43.05 / 53.79 / 42.45 | 47.97 / 53.50 / 44.04 | 75.94 / 54.23 / 43.40 |
| | | DILATE | 92.78 / 112.62 / 88.26 | 85.55 / 109.70 / 81.02 | 82.77 / 104.71 / 84.96 | 130.25 / 113.95 / 86.01 |
| Informer | No ESE | RMSE | 22.92 / 72.13 / 59.61 | 56.44 / 73.54 / 74.51 | 71.49 / 74.89 / 70.10 | 93.52 / 73.26 / 74.13 |
| | | MAE | 20.71 / 59.39 / 49.49 | 53.23 / 67.45 / 63.64 | 62.63 / 73.56 / 65.62 | 90.27 / 67.35 / 72.68 |
| | | DILATE | 48.32 / 113.47 / 104.16 | 88.81 / 116.67 / 106.36 | 101.45 / 121.24 / 106.10 | 140.17 / 123.48 / 113.73 |
| | With ESE | RMSE | 48.91 / 60.33 / 45.33 | 58.05 / 63.38 / 47.24 | 64.36 / 59.42 / 50.85 | 80.67 / 59.92 / 48.64 |
| | | MAE | 46.65 / 56.45 / 43.03 | 52.96 / 55.95 / 45.59 | 53.44 / 53.93 / 47.36 | 80.24 / 53.83 / 47.08 |
| | | DILATE | 96.76 / 115.66 / 89.05 | 92.64 / 114.93 / 84.22 | 87.00 / 105.64 / 88.17 | 134.14 / 115.22 / 91.27 |
| FiLM | No ESE | RMSE | 23.37 / 67.77 / 58.14 | 52.91 / 68.19 / 64.47 | 70.69 / 70.95 / 65.43 | 98.40 / 68.25 / 73.76 |
| | | MAE | 17.00 / 47.94 / 37.09 | 46.15 / 56.51 / 54.90 | 52.49 / 54.18 / 50.92 | 82.39 / 56.29 / 54.97 |
| | | DILATE | 46.93 / 110.09 / 96.07 | 85.45 / 112.35 / 103.06 | 101.78 / 115.29 / 98.14 | 125.44 / 122.49 / 113.50 |
| | With ESE | RMSE | 53.83 / 61.01 / 50.01 | 55.65 / 62.54 / 51.02 | 59.12 / 65.58 / 55.16 | 86.48 / 65.61 / 50.67 |
| | | MAE | 42.11 / 52.43 / 39.40 | 49.64 / 51.79 / 41.04 | 46.08 / 50.41 / 43.20 | 71.67 / 49.26 / 39.06 |
| | | DILATE | 92.53 / 107.13 / 82.48 | 85.16 / 110.01 / 79.91 | 79.95 / 103.14 / 83.09 | 126.04 / 112.30 / 86.10 |
| SCINet | No ESE | RMSE | 20.48 / 60.85 / 55.10 | 53.32 / 69.22 / 68.01 | 71.08 / 72.02 / 72.32 | 96.06 / 69.10 / 71.77 |
| | | MAE | 19.36 / 56.65 / 42.04 | 50.05 / 63.89 / 64.95 | 69.14 / 64.99 / 70.29 | 88.44 / 63.85 / 70.98 |
| | | DILATE | 43.96 / 108.74 / 100.85 | 83.15 / 111.04 / 110.93 | 105.47 / 119.92 / 108.09 | 134.81 / 121.46 / 113.48 |
| | With ESE | RMSE | 43.69 / 62.07 / 44.21 | 54.73 / 63.95 / 48.83 | 59.43 / 54.41 / 45.66 | 83.61 / 54.32 / 46.69 |
| | | MAE | 42.12 / 59.91 / 38.27 | 52.05 / 51.63 / 42.97 | 48.95 / 50.42 / 42.81 | 74.86 / 52.71 / 43.78 |
| | | DILATE | 92.20 / 107.98 / 86.00 | 87.98 / 110.07 / 84.23 | 84.57 / 106.37 / 85.26 | 132.25 / 112.62 / 84.60 |
| DeepAR | No ESE | RMSE | 24.88 / 61.90 / 61.21 | 55.00 / 66.93 / 66.71 | 75.73 / 73.09 / 69.34 | 99.18 / 67.14 / 73.78 |
| | | MAE | 19.40 / 52.81 / 56.06 | 48.34 / 62.01 / 59.71 | 60.37 / 66.31 / 54.99 | 84.99 / 62.18 / 64.88 |
| | | DILATE | 47.25 / 119.99 / 111.05 | 87.39 / 117.57 / 111.08 | 109.62 / 127.79 / 107.27 | 139.40 / 124.31 / 114.05 |
| | With ESE | RMSE | 52.55 / 63.69 / 45.47 | 59.99 / 60.65 / 49.65 | 63.25 / 60.91 / 54.13 | 87.15 / 58.46 / 53.12 |
| | | MAE | 49.54 / 59.83 / 39.27 | 54.76 / 54.08 / 43.92 | 52.39 / 56.94 / 45.44 | 91.87 / 56.68 / 46.61 |
| | | DILATE | 94.16 /107.97 / 94.09 | 90.70 / 106.95 / 78.64 | 84.46 / 105.02 / 82.79 | 129.77 / 110.14 / 81.88 |
| KVAE | No ESE | RMSE | 23.27 / 60.72 / 57.66 | 58.95 / 70.89 / 69.58 | 65.42 / 71.62 / 70.30 | 98.45 / 70.75 / 74.00 |
| | | MAE | 22.32 / 58.54 / 52.38 | 53.05 / 67.86 / 64.93 | 62.51 / 67.81 / 71.92 | 97.92 / 68.02 / 74.63 |
| | | DILATE | 40.05 / 96.89 / 88.48 | 82.51 / 109.64 / 98.68 | 91.43 / 112.65 / 98.33 | 125.39 / 118.89 / 105.96 |
| | With ESE | RMSE | 50.00 / 61.58 / 47.22 | 60.46 / 59.77 / 47.14 | 62.97 / 60.81 / 54.10 | 85.18 / 60.70 / 54.49 |
| | | MAE | 42.43 / 59.51 / 40.00 | 51.24 / 52.44 / 41.47 | 49.07 / 55.56 / 45.52 | 78.06 / 55.50 / 44.42 |
| | | DILATE | 94.94 / 109.02 / 80.13 | 85.39 / 108.19 / 80.32 | 85.63 / 103.92 / 87.88 | 134.26 / 117.88 / 87.54 |
| TPGNN | No ESE | RMSE | 23.25 / 64.05 / 58.41 | 56.21 / 64.98 / 68.01 | 71.15 / 75.09 / 69.00 | 96.70 / 65.08 / 80.21 |
| | | MAE | 21.61 / 57.41 / 49.71 | 51.98 / 62.90 / 65.43 | 63.23 / 71.22 / 59.37 | 90.85 / 62.77 / 69.98 |
| | | DILATE | 41.34 / 103.43 / 92.21 | 82.13 / 109.08 / 100.27 | 101.92 / 116.20 / 97.59 | 128.65 / 119.48 / 116.65 |
| | With ESE | RMSE | 53.04 / 64.88 / 50.49 | 59.39 / 61.65 / 49.79 | 65.28 / 64.04 / 55.30 | 87.54 / 64.02 / 54.93 |
| | | MAE | 49.84 / 59.39 / 37.98 | 53.98 / 52.68 / 48.78 | 52.98 / 59.87 / 50.12 | 84.26 / 59.44 / 48.48 |
| | | DILATE | 82.24 / 109.59 / 84.07 | 87.16 / 110.38 / 80.47 | 82.47 / 104.30 / 83.55 | 128.09 / 110.19 / 84.32 |
| PatchTST | No ESE | RMSE | 21.10 / 67.19 / 58.08 | 50.31 / 70.11 / 69.40 | 77.25 / 76.29 / 70.66 | 92.65 / 78.95 / 68.08 |
| | | MAE | 20.53 / 61.18 / 46.43 | 43.36 / 64.12 / 62.86 | 68.05 / 72.54 / 59.04 | 85.67 / 73.31 / 61.05 |
| | | DILATE | 35.13 / 112.75 / 114.12 | 76.29 / 116.47 / 113.85 | 119.00 / 122.54 / 105.50 | 127.33 / 118.50 / 107.17 |
| | With ESE | RMSE | 53.04 / 51.74 / 68.66 | 53.37 / 64.31 / 50.17 | 69.18 / 61.00 / 55.05 | 83.42 / 60.67 / 50.41 |
| | | MAE | 48.56 / 59.60 / 36.25 | 48.91 / 57.96 / 47.02 | 55.34 / 53.04 / 49.14 | 76.01 / 55.41 / 48.36 |
| | | DILATE | 94.00 / 106.89 / 86.48 | 87.27 / 108.79 / 74.21 | 79.16 / 108.51 / 82.13 | 111.36 / 103.62 / 81.61 |

## J.2 PREDICTION COST ON COVID-19 DATA

Table 16: Comparing computational cost (output step is 1) with 12 SOTA methods, with no ESE and with ESE, with 10, 20, 50, 100 steps of input, for 20 large regions / 79 regions / 320 sub-regions.

| Models | | Computational Costs (mins) | | | |
|---|---|---|---|---|---|
| | | *Input Length = 10* | *Input Length = 20* | *Input Length = 50* | *Input Length = 100* |
| ESE | – | 1.19 / 1.49 / 1.71 | 1.23 / 1.43 / 1.82 | 1.22 / 1.45 / 1.97 | 1.31 / 2.10 / 2.28 |
| ARIMA | No ESE | 0.18 / 0.70 / 2.81 | 0.22 / 0.89 / 3.59 | 0.27 / 1.09 / 4.11 | 0.33 / 1.31 / 5.07 |
| | With ESE | 1.20 / 1.50 / 1.72 | 1.24 / 1.44 / 1.83 | 1.23 / 1.46 / 1.99 | 1.33 / 2.12 / 2.30 |
| LSTM | No ESE | 5.06 / 20.68 / 86.88 | 6.14 / 27.76 / 109.29 | 7.77 / 33.81 / 131.74 | 9.23 / 39.96 / 149.26 |
| | With ESE | 1.46 / 1.77 / 1.98 | 1.57 / 1.74 / 2.13 | 1.61 / 1.84 / 2.41 | 1.80 / 2.59 / 2.72 |
| Dlinear | No ESE | 6.20 / 23.96 / 96.44 | 7.07 / 31.61 / 132.31 | 10.28 / 38.58 / 159.46 | 10.40 / 40.98 / 163.57 |
| | With ESE | 1.48 / 1.79 / 2.03 | 1.62 / 1.81 / 2.18 | 1.69 / 1.90 / 2.49 | 1.86 / 2.68 / 2.87 |
| Nlinear | No ESE | 6.04 / 24.91 / 99.01 | 7.40 / 31.37 / 135.11 | 9.57 / 37.83 / 155.00 | 11.56 / 45.67 / 160.09 |
| | With ESE | 1.50 / 1.81 / 2.01 | 1.60 / 1.80 / 2.23 | 1.72 / 1.93 / 2.41 | 1.87 / 2.70 / 2.86 |
| Informer | No ESE | 3.52 / 13.24 / 57.48 | 4.56 / 17.16 / 74.00 | 5.38 / 20.58 / 93.92 | 5.95 / 26.97 / 100.08 |
| | With ESE | 1.36 / 1.67 / 1.88 | 1.43 / 1.67 / 2.03 | 1.48 / 1.73 / 2.25 | 1.66 / 2.40 / 2.62 |
| FiLM | No ESE | 6.63 / 26.22 / 108.97 | 7.62 / 34.91 / 143.09 | 9.66 / 40.59 / 171.36 | 11.70 / 48.55 / 181.06 |
| | With ESE | 1.50 / 1.82 / 2.02 | 1.65 / 1.87 / 2.24 | 1.73 / 1.96 / 2.44 | 1.86 / 2.74 / 2.84 |
| SCINet | No ESE | 7.98 / 30.62 / 127.32 | 10.37 / 40.89 / 151.48 | 12.24 / 49.96 / 189.39 | 14.18 / 62.27 / 206.06 |
| | With ESE | 1.57 / 1.88 / 2.12 | 1.70 / 1.92 / 2.31 | 1.78 / 2.11 / 2.60 | 2.04 / 2.82 / 2.94 |
| DeepAR | No ESE | 5.11 / 19.57 / 76.72 | 6.16 / 23.56 / 94.99 | 7.34 / 31.55 / 131.00 | 9.22 / 37.25 / 130.67 |
| | With ESE | 1.43 / 1.74 / 1.95 | 1.52 / 1.72 / 2.15 | 1.63 / 1.87 / 2.39 | 1.77 / 2.52 / 2.73 |
| KVAE | No ESE | 4.37 / 17.03 / 67.34 | 5.21 / 22.17 / 90.23 | 7.19 / 28.31 / 96.23 | 6.80 / 32.41 / 109.62 |
| | With ESE | 1.41 / 1.71 / 1.92 | 1.49 / 1.69 / 2.07 | 1.53 / 1.78 / 2.30 | 1.67 / 2.50 / 2.63 |
| TPGNN | No ESE | 5.87 / 23.56 / 97.40 | 7.90 / 27.60 / 119.72 | 9.85 / 35.00 / 152.22 | 10.58 / 42.92 / 158.84 |
| | With ESE | 1.49 / 1.80 / 2.00 | 1.62 / 1.82 / 2.19 | 1.72 / 1.91 / 2.48 | 1.81 / 2.65 / 2.86 |
| PatchTST | No ESE | 3.71 / 15.55 / 61.08 | 4.60 / 18.68 / 82.90 | 5.85 / 23.96 / 94.47 | 6.92 / 27.70 / 109.89 |
| | With ESE | 1.38 / 1.69 / 1.90 | 1.48 / 1.69 / 2.08 | 1.53 / 1.78 / 2.29 | 1.79 / 2.46 / 2.66 |

Table 17: Comparing computational cost (output step is 2) with 12 SOTA methods, with no ESE and with ESE, with 10, 20, 50, 100 steps of input, for 20 large regions / 79 regions / 320 sub-regions.

| Models | | Computational Costs (mins) | | | |
|---|---|---|---|---|---|
| | | *Input Length = 10* | *Input Length = 20* | *Input Length = 50* | *Input Length = 100* |
| ESE | | 1.20 / 1.50 / 1.72 | 1.23 / 1.44 / 1.81 | 1.22 / 1.46 / 1.99 | 1.32 / 2.11 / 2.30 |
| ARIMA | No ESE | 0.18 / 0.70 / 2.71 | 0.21 / 0.94 / 3.78 | 0.29 / 1.02 / 4.79 | 0.35 / 1.30 / 5.32 |
| | With ESE | 1.21 / 1.51 / 1.73 | 1.24 / 1.45 / 1.84 | 1.24 / 1.47 / 1.99 | 1.34 / 2.13 / 2.30 |
| LSTM | No ESE | 5.06 / 20.24 / 88.66 | 7.04 / 25.26 / 104.36 | 7.83 / 33.44 / 132.97 | 10.29 / 41.51 / 153.76 |
| | With ESE | 1.47 / 1.76 / 1.96 | 1.55 / 1.75 / 2.15 | 1.61 / 1.89 / 2.41 | 1.77 / 2.59 / 2.75 |
| Dlinear | No ESE | 6.02 / 24.55 / 95.88 | 7.02 / 32.21 / 127.01 | 10.14 / 36.63 / 140.45 | 10.77 / 48.41 / 165.12 |
| | With ESE | 1.49 / 1.80 / 2.02 | 1.62 / 1.81 / 2.22 | 1.69 / 1.91 / 2.51 | 1.88 / 2.71 / 2.88 |
| Nlinear | No ESE | 6.18 / 23.48 / 98.89 | 7.86 / 32.94 / 124.18 | 10.26 / 37.00 / 150.42 | 11.08 / 43.64 / 186.56 |
| | With ESE | 1.48 / 1.80 / 2.03 | 1.65 / 1.83 / 2.19 | 1.67 / 1.93 / 2.50 | 1.91 / 2.67 / 2.86 |
| Informer | No ESE | 3.53 / 14.52 / 58.83 | 4.67 / 16.95 / 71.12 | 5.40 / 22.88 / 88.41 | 6.54 / 24.66 / 93.47 |
| | With ESE | 1.36 / 1.68 / 1.89 | 1.43 / 1.65 / 2.05 | 1.50 / 1.75 / 2.27 | 1.62 / 2.41 / 2.62 |
| FiLM | No ESE | 6.49 / 25.28 / 102.23 | 7.62 / 32.35 / 138.70 | 10.62 / 42.84 / 167.44 | 11.61 / 43.21 / 198.15 |
| | With ESE | 1.52 / 1.81 / 2.06 | 1.66 / 1.85 / 2.25 | 1.76 / 2.01 / 2.46 | 1.95 / 2.65 / 2.84 |
| SCINet | No ESE | 8.09 / 30.57 / 130.87 | 9.43 / 41.49 / 158.16 | 12.55 / 45.86 / 211.56 | 15.92 / 55.12 / 236.50 |
| | With ESE | 1.59 / 1.91 / 2.10 | 1.70 / 1.91 / 2.34 | 1.83 / 2.03 / 2.64 | 2.09 / 2.89 / 3.05 |
| DeepAR | No ESE | 4.75 / 19.20 / 79.48 | 6.58 / 24.60 / 95.80 | 7.70 / 29.92 / 126.18 | 9.03 / 31.82 / 151.75 |
| | With ESE | 1.45 / 1.73 / 1.96 | 1.54 / 1.76 / 2.14 | 1.63 / 1.81 / 2.34 | 1.73 / 2.55 / 2.71 |
| KVAE | No ESE | 4.02 / 16.90 / 65.41 | 5.57 / 22.04 / 84.81 | 6.08 / 27.94 / 113.13 | 7.36 / 32.06 / 133.49 |
| | With ESE | 1.40 / 1.72 / 1.92 | 1.47 / 1.72 / 2.09 | 1.57 / 1.80 / 2.31 | 1.70 / 2.50 / 2.71 |
| TPGNN | No ESE | 6.09 / 22.68 / 99.67 | 7.21 / 28.38 / 118.50 | 9.36 / 37.78 / 152.74 | 10.76 / 44.22 / 155.24 |
| | With ESE | 1.48 / 1.80 / 2.02 | 1.59 / 1.80 / 2.24 | 1.67 / 1.94 / 2.43 | 1.86 / 2.63 / 2.87 |
| PatchTST | No ESE | 3.77 / 14.77 / 60.33 | 4.83 / 18.83 / 81.41 | 6.31 / 24.19 / 101.68 | 7.29 / 28.69 / 103.71 |
| | With ESE | 1.38 / 1.69 / 1.91 | 1.47 / 1.67 / 2.08 | 1.54 / 1.76 / 2.31 | 1.69 / 2.47 / 2.61 |

Table 18: Comparing computational cost (output step is 5) with 12 SOTA methods, with no ESE and with ESE, with 10, 20, 50, 100 steps of input, for 20 large regions / 79 regions / 320 sub-regions.

| Models | | Computational Costs (mins) | | | |
| --- | --- | --- | --- | --- | --- |
| | | *Input Length = 10* | *Input Length = 20* | *Input Length = 50* | *Input Length = 100* |
| ESE | | 1.25 / 1.53 / 1.91 | 1.31 / 1.61 / 1.90 | 1.23 / 1.54 / 2.05 | 1.40 / 2.41 / 2.38 |
| ARIMA | No ESE | 0.18 / 0.83 / 3.22 | 0.24 / 1.02 / 3.64 | 0.30 / 1.19 / 4.55 | 0.34 / 1.46 / 5.81 |
| | With ESE | 1.24 / 1.68 / 1.91 | 1.36 / 1.64 / 1.92 | 1.36 / 1.49 / 2.10 | 1.51 / 2.14 / 2.33 |
| LSTM | No ESE | 5.68 / 21.82 / 91.98 | 7.45 / 27.28 / 112.66 | 8.79 / 32.04 / 138.17 | 11.39 / 41.16 / 158.84 |
| | With ESE | 1.67 / 2.04 / 2.23 | 1.68 / 2.08 / 2.49 | 1.92 / 2.07 / 2.76 | 2.01 / 2.70 / 3.24 |
| Dlinear | No ESE | 6.05 / 23.42 / 113.54 | 7.03 / 35.35 / 124.01 | 10.35 / 41.71 / 160.66 | 11.36 / 40.65 / 189.59 |
| | With ESE | 1.57 / 1.88 / 2.22 | 1.78 / 1.85 / 2.34 | 1.83 / 2.11 / 2.68 | 1.97 / 2.64 / 3.05 |
| Nlinear | No ESE | 6.59 / 27.11 / 103.52 | 7.77 / 32.62 / 146.44 | 10.50 / 39.99 / 168.24 | 12.11 / 43.00 / 177.14 |
| | With ESE | 1.56 / 1.88 / 2.23 | 1.73 / 1.84 / 2.48 | 1.78 / 2.07 / 2.72 | 1.95 / 2.75 / 3.01 |
| Informer | No ESE | 4.08 / 14.90 / 64.72 | 4.35 / 17.16 / 76.05 | 5.96 / 24.74 / 99.45 | 6.86 / 26.33 / 108.50 |
| | With ESE | 1.53 / 1.90 / 2.08 | 1.59 / 1.66 / 2.22 | 1.56 / 1.80 / 2.55 | 1.73 / 2.74 / 2.64 |
| FiLM | No ESE | 6.65 / 27.88 / 115.14 | 7.84 / 33.46 / 140.27 | 10.58 / 45.99 / 193.35 | 11.37 / 49.80 / 177.95 |
| | With ESE | 1.55 / 2.07 / 2.24 | 1.87 / 2.05 / 2.45 | 1.93 / 2.00 / 2.52 | 1.94 / 3.01 / 3.13 |
| SCINet | No ESE | 8.52 / 31.75 / 144.09 | 11.46 / 40.04 / 183.36 | 12.63 / 50.63 / 222.10 | 14.04 / 62.88 / 262.88 |
| | With ESE | 1.68 / 2.11 / 2.37 | 1.74 / 1.96 / 2.66 | 1.89 / 2.39 / 2.94 | 1.97 / 3.12 / 3.44 |
| DeepAR | No ESE | 5.07 / 18.75 / 82.37 | 6.46 / 25.93 / 102.85 | 7.53 / 29.79 / 123.68 | 9.52 / 39.91 / 143.83 |
| | With ESE | 1.51 / 1.90 / 1.96 | 1.65 / 1.76 / 2.29 | 1.82 / 2.07 / 2.69 | 1.97 / 2.83 / 3.06 |
| KVAE | No ESE | 4.64 / 19.12 / 66.30 | 5.56 / 20.49 / 91.55 | 7.02 / 24.84 / 120.19 | 9.01 / 31.98 / 153.34 |
| | With ESE | 1.41 / 1.84 / 2.20 | 1.71 / 1.78 / 2.15 | 1.54 / 1.79 / 2.63 | 1.76 / 2.77 / 2.86 |
| TPGNN | No ESE | 6.13 / 26.30 / 111.59 | 8.85 / 34.59 / 123.10 | 9.10 / 37.87 / 167.70 | 10.93 / 47.32 / 167.42 |
| | With ESE | 1.67 / 2.06 / 2.08 | 1.82 / 1.83 / 2.22 | 1.78 / 2.09 / 2.52 | 2.02 / 2.65 / 2.96 |
| PatchTST | No ESE | 4.30 / 16.03 / 67.92 | 5.10 / 21.31 / 91.14 | 6.46 / 25.72 / 117.19 | 7.36 / 27.97 / 132.35 |
| | With ESE | 1.57 / 1.70 / 2.08 | 1.55 / 1.84 / 2.23 | 1.53 / 1.78 / 2.43 | 1.81 / 2.61 / 2.70 |

Table 19: Comparing computational cost (output step is 10) with 12 SOTA methods, with no ESE and with ESE, with 10, 20, 50, 100 steps of input, for 20 large regions / 79 regions / 320 sub-regions.

| Models | | Computational Costs (mins) | | | |
| --- | --- | --- | --- | --- | --- |
| | | *Input Length = 10* | *Input Length = 20* | *Input Length = 50* | *Input Length = 100* |
| ESE | | 1.20 / 1.77 / 2.00 | 1.30 / 1.65 / 2.03 | 1.22 / 1.66 / 1.98 | 1.36 / 2.11 / 2.66 |
| ARIMA | No ESE | 0.20 / 0.77 / 3.11 | 0.23 / 0.97 / 3.86 | 0.33 / 1.29 / 4.64 | 0.31 / 1.47 / 5.27 |
| | With ESE | 1.32 / 1.78 / 1.89 | 1.37 / 1.54 / 2.12 | 1.40 / 1.51 / 2.03 | 1.51 / 2.15 / 2.57 |
| LSTM | No ESE | 5.39 / 23.63 / 93.81 | 6.64 / 27.15 / 130.60 | 8.34 / 35.48 / 156.14 | 11.54 / 41.35 / 183.91 |
| | With ESE | 1.60 / 2.03 / 2.17 | 1.65 / 2.04 / 2.47 | 1.77 / 2.09 / 2.65 | 1.84 / 3.00 / 2.98 |
| Dlinear | No ESE | 7.17 / 25.10 / 102.81 | 8.49 / 33.99 / 149.76 | 9.38 / 38.68 / 180.71 | 12.24 / 43.94 / 190.21 |
| | With ESE | 1.68 / 2.10 / 2.35 | 1.70 / 2.05 / 2.33 | 1.94 / 2.23 / 2.51 | 1.96 / 2.75 / 3.27 |
| Nlinear | No ESE | 6.49 / 26.95 / 104.37 | 7.99 / 37.47 / 136.83 | 9.96 / 44.62 / 156.76 | 11.51 / 48.25 / 171.26 |
| | With ESE | 1.58 / 1.99 / 2.07 | 1.85 / 1.82 / 2.45 | 1.84 / 2.12 / 2.45 | 1.96 / 2.72 / 3.04 |
| Informer | No ESE | 3.71 / 15.70 / 62.75 | 4.41 / 20.75 / 86.88 | 5.85 / 23.84 / 89.99 | 6.69 / 27.92 / 97.35 |
| | With ESE | 1.52 / 1.80 / 2.00 | 1.61 / 1.98 / 2.46 | 1.72 / 1.93 / 2.38 | 1.66 / 2.89 / 2.76 |
| FiLM | No ESE | 7.00 / 26.69 / 115.83 | 9.40 / 36.31 / 138.15 | 10.87 / 44.67 / 172.86 | 12.54 / 57.13 / 218.43 |
| | With ESE | 1.76 / 2.19 / 2.14 | 1.97 / 1.99 / 2.55 | 1.81 / 2.00 / 2.63 | 2.00 / 2.99 / 3.26 |
| SCINet | No ESE | 9.11 / 31.92 / 133.02 | 11.90 / 38.60 / 161.36 | 13.38 / 49.13 / 214.69 | 14.99 / 64.88 / 239.29 |
| | With ESE | 1.87 / 1.96 / 2.26 | 1.82 / 2.15 / 2.35 | 1.91 / 2.30 / 2.64 | 2.15 / 3.02 / 3.58 |
| DeepAR | No ESE | 5.75 / 22.47 / 90.77 | 6.66 / 27.13 / 103.54 | 8.19 / 33.24 / 131.73 | 9.22 / 37.33 / 172.25 |
| | With ESE | 1.68 / 1.88 / 2.32 | 1.59 / 2.01 / 2.25 | 1.65 / 2.24 / 2.71 | 1.93 / 2.89 / 2.93 |
| KVAE | No ESE | 4.33 / 18.79 / 73.88 | 4.90 / 25.86 / 97.79 | 7.79 / 30.02 / 123.96 | 7.83 / 28.27 / 144.77 |
| | With ESE | 1.39 / 1.71 / 2.25 | 1.52 / 2.00 / 2.42 | 1.63 / 1.76 / 2.54 | 1.71 / 2.96 / 2.74 |
| TPGNN | No ESE | 5.97 / 24.32 / 104.31 | 7.44 / 34.46 / 155.33 | 10.28 / 45.28 / 144.83 | 12.46 / 50.40 / 172.18 |
| | With ESE | 1.72 / 2.03 / 2.25 | 1.62 / 2.09 / 2.58 | 1.70 / 2.10 / 2.74 | 2.11 / 2.89 / 3.38 |
| PatchTST | No ESE | 4.39 / 15.63 / 66.70 | 4.99 / 20.69 / 89.16 | 7.09 / 24.81 / 109.28 | 6.74 / 33.07 / 118.99 |
| | With ESE | 1.51 / 1.77 / 1.94 | 1.70 / 1.78 / 2.12 | 1.63 / 2.00 / 2.56 | 1.66 / 2.56 / 3.14 |

