# OpenReview forum: "Integrated Multi-system Prediction via Equilibrium State Evaluation"
_ICLR.cc/2025/Conference — Submitted to ICLR 2025_

### Official Review · Reviewer_i1oB · 2024-11-02

**Soundness:** 1
**Presentation:** 1
**Contribution:** 1
**Rating:** 1
**Confidence:** 5

**Summary:**

This paper presents a new prediction approach called Equilibrium State Evaluation for handling multi-system prediction problems where multiple interacting systems must be predicted simultaneously. Where the authors treat each system as independent time series and consider multiple-interacting systems as components of a "super-system". The method maintains an Equilibrium Index (EI) that measures the distance between the current state and the equilibrium state.

**Strengths:**

There aren't any significant strengths in the paper

**Weaknesses:**

- The paper is confusing to follow. The introduction is poorly written. The mathematical notation is very dense without sufficient explanation.
- Complex method workflow is not fully explained
- It is not apparent to me why proportions must follow zero-sum rule. Doesn't explain how to handles growing-shrinking total system values.
- It is unclear why is specific normalization is chosen λᵢ,ⱼ = (αᵢ,ⱼ - mean(atⱼ)) / (max(atⱼ) - min(atⱼ)), why is this normalization beneficial as compared to other ways to normalize?
- Algorithm 1 is vaguely described, and testing and training process aren't well defined. It doesn't map succinctly with the figure 2 of the paper.

**Questions:**

The authors could provide a more detailed algorithm description and better justification of experimental and methodological design

---

> ### Author Response · Authors · 2024-11-20
> **Please reconsider the comments on weaknesses**
>
> > ***This paper presents a new prediction approach called Equilibrium State Evaluation for handling multi-system prediction problems where multiple interacting systems must be predicted simultaneously.***
>
> We really appreciate your review summary which precisely describes the problem that we are working on in this paper.  In the meantime, we are sorry to hear that the mathematical formulation of the task and the description of our methodology are not understood.  Below please see our further explanations below, which hopefully can clear up your doubts and concerns.
>
> > ***There aren't any significant strengths in the paper***
>
> Sorry we don't understand this comment.  Our experiments clearly show ESE is at least as accurate as, but much faster than **12 SOTA time series prediction algorithms**.  When combined with other prediction algorithms, ESE can achieve the highest accuracy in almost all cases (Tables 1, 2, 3 in the main paper + Tables 4 - 9, 12 - 19 in the appendix).  In terms of cost, ESE can be **70+ times** faster (results bold in Table 3).
>
> > ***Weakness 1 - confusing to follow.***
>
> We are quite puzzled to see the introduction being commented as poorly written.  We would really appreciate specific comments and suggestions so improvement can be actioned upon accordingly.
>
> Regarding mathematical notions, we acknowledge that there are many equations in this paper, but for good and valid reasons.  We have gone through extensive mathematical rigor to formulate the multi-system prediction problem and our methodologies, as precise as possible.  This is to establish a solid theoretical foundation for this study.  Without these mathematical notations and equations, it would be more likely to cause ambiguities and misunderstandings of our work.  We believe strong theoretical foundation with solid mathematical formulation is what ICLR and sister venues are looking for.
>
> In addition, many equations are derived from game theory with small variations, hence should not be too difficult to follow for readers that are familiar with the field.  For the benefit of readers, we have added citations to classic game theory books.  Furthermore, we intentionally placed some of the deductive processes in the appendix to reduce the amount of equations.
>
> > ***Weakness 2 - Complex method workflow is not fully explained***
>
> Similar to Weakness 1, we are quite puzzled to see the comment and would be really grateful if you could be more specific so we would be able to address and/or action accordingly, e.g. which part is not explained? Which part needs more clarification?
>
> In order to present this new paradigm of prediction, we not only carefully established the mathematical model, but also used figures to illustrate the method.  In addition, our source code is fully viewable, so can be checked and verified.
>
> > ***Weakness 3 - not apparent to me why proportions must follow zero-sum rule ... how to handles growing-shrinking total system values.***
>
> Zero-sum is the foundation of equilibrium, hence the basis of our ESE method, as described in Definition 2 and Constraints 1 & 2.  ESE would NOT be directly suitable for situations where the assumption of zero-sum does not hold.  With this zero-sum rule, all changes in a multi-system can be viewed as interactions between the systems within.  Hence our method can apply.
>
> In regard to the growing shrinking of total values, that is normally the case in multi-systems.  Hence we deal with proportions instead of absolute values, as shown in Constraint 2 (Eq. 4) and Appendix A (Eq. 16).  The total changes in proportion are always 0, and the total proportion is always 1, making the multi-system align well with zero-sum conditions described above.
>
> > ***Weakness 4 - unclear why is specific normalization is chosen...***
>
> We are not 100% sure what you are referring to.  Would it be Eq. 9?   If so, the explanation can be found at the top of Page 6, from Lines 271 to 274. “***Because the system is assumed to be a zero-sum system, the aggregation of all $\lambda_{i,j}$, e.g. the feature of a particular attribute of all systems, is always summed to zero, e.g. $\sum_{j=1}^m\lambda_{i,j}=0$. That is applicable for all features/attributes, hence we can have Equation 10.***”
>
> In addition, Eq. 9 normalizes all attribute values between -1 and 1, making it easy to see the impact of an attribute, either positive or negative. If that value is greater than 0, we would know this attribute has a positive impact on the system.  Likewise, a negative value indicates the attribute’s impact being negative.  To make this point clear, we have added one sentence in the text in Line 274: “***In addition, the value of $\lambda_{i,j}$ would indicate the magnitude of the change in $s_i$ and whether the change is positive or negative.***”

---

> ### Author Response · Authors · 2024-11-20
> **Addressing the question, Algorithm 1**
>
> > ***Algorithm 1 is vaguely described, and testing and training process aren't well defined. It doesn't map succinctly with the figure 2 of the paper.***
>
> Algorithm 1 is actually quite straightforward so we thought it would be self-explanatory.  Please see the source code below, which is also viewable in our anonymous repository. It simply iteratively progresses with all the time steps $i$, Steps 3-5, until a cointegration condition is satisfied (Step 2).  Note, this algorithm is the equilibrium training process, a small part of our ESE method.  Hence it is just a small part of Fig 2, the “***Equilibrium Training***” block in between the rings of $\mathcal{ES}^{[0]} $ and $\mathcal{ST} _{t+h}$.  We have updated the main paper to highlight that.
>
> ```
> # https://anonymous.4open.science/r/ESE-6432/code/LongRunTraining.py
>
> # Equilibrium Training Function
>
> def long_run_equilibrium_l(esps_0,spss):         # esps: equilibrium state parameter set;
>                                                  # spss: State Parameter SetS
>
>     i = 0
>     no_part = len(esps_0)                        ### number of parts, esps_0 as the initial esps
>     l = np.ones(no_part)                         ### initialise all values of vector l to 1
>
>     esps = esps_0
>
>     while True:
>         for i in range(len(spss)):
>             l = (esps - spss[i] + l) / 2
>
>         esps = esps - (l / 2)
>
>         if cointegration(esps,spss) == False:    ### break if esps and spss are in cointegration!
>             break                                ### Note: False means the rejection of the null hypothesis,
>                                                  ### e.g. in cointegration.
>         i+=1
>     return esps
> ```

---

### Official Review · Reviewer_ddNh · 2024-11-03

**Soundness:** 2
**Presentation:** 3
**Contribution:** 2
**Rating:** 5
**Confidence:** 4

**Summary:**

The paper presents a new model called Equilibrium State Evaluation (ESE) for predicting the collective behavior of interconnected multi-system structures. ESE is designed to manage systems with multiple interdependent subsystems, each affecting and being affected by the others. The goal of the model is to determine an equilibrium state for the entire multi-system by applying concepts from Nash equilibrium along with a zero-sum constraint.
In ESE, each subsystem is given a payoff function that assesses its "benefit" based on the states of all the subsystems. The equilibrium state is defined as the configuration in which each subsystem achieves its optimal state (maximum payoff), given the conditions of the others, consistent with Nash equilibrium principles. The zero-sum constraint enforces a balance across subsystems, ensuring that any gain in one subsystem is countered by a loss in another, thereby maintaining overall stability.

To estimate the long-term equilibrium trends of the system, the model utilizes cointegration, an econometric technique that identifies shared equilibrium paths among interdependent variables. Additionally, an equilibrium index (EI) is introduced to quantify how closely the system's current state aligns with equilibrium, providing a measure of system stability.

The paper validates the ESE model using multi-regional COVID-19 transmission data, demonstrating that it achieves competitive prediction accuracy while maintaining linear computational complexity. The authors contend that ESE's equilibrium-based approach enables scalable and efficient prediction of complex multi-system dynamics, making it potentially valuable in high-dimensional environments where subsystems are closely interconnected.

**Strengths:**

The primary strength of this paper lies in its effort to apply a novel conceptual framework for modeling interconnected systems. It draws on ideas from Nash equilibrium, zero-sum constraints, and cointegration. This approach offers an interesting perspective by reinterpreting multi-system predictions through game-theoretic concepts, particularly utilizing Nash equilibrium principles and payoff functions to illustrate the mutual influences among subsystems. While these concepts are mainly interpretative, they provide a fresh way to think about multi-system interactions, which could have significant implications for interdisciplinary applications across fields such as economics, epidemiology, and regional forecasting. The paper demonstrates the practical feasibility of the proposed model, ESE, by applying it to a real-world COVID-19 dataset. This highlights the model's potential to capture complex interdependencies and offers insights into multi-region transmission dynamics. The presentation is generally clear, with structured explanations of equilibrium concepts and the role of each component, such as the equilibrium index. Furthermore, if the model’s empirical performance in terms of accuracy and computational efficiency is further substantiated, it suggests that ESE could serve as a competitive alternative for high-dimensional applications where subsystems are highly interconnected.

**Weaknesses:**

A significant weakness of the paper lies in its insufficient support for the claim that the system will reach a true equilibrium state. While the model employs Nash equilibrium and zero-sum constraints to define static equilibrium conditions, these concepts alone do not provide a mechanism to ensure that the system will naturally progress toward equilibrium over time. Without a dynamic framework or time-dependent interactions—such as differential equations or explicit stability conditions—there is no mathematical basis for assuming that the system will move from an arbitrary initial state toward equilibrium. Furthermore, although the paper incorporates cointegration in its training process, this is a statistical technique that assumes the existence of a long-term equilibrium relationship rather than actively guiding the system toward it. This distinction weakens the model’s theoretical foundation, as it does not establish how equilibrium is achieved dynamically. Another critical limitation is that the equilibrium definition based on Nash equilibrium and payoff functions does not exclude the possibility of stable oscillatory behavior. This means the system could theoretically settle into a stable oscillation rather than converging to a true steady state. Additionally, the ESE model shows strong mathematical similarities to traditional multi-compartment models, which use conservation principles to maintain balance across subsystems. While the introduction of game-theoretic concepts like Nash equilibrium and payoff functions offers a fresh interpretive layer, it does not constitute a substantial mathematical advancement over established multi-compartment models. To improve the model’s robustness, the authors would need to provide a formal analysis of convergence conditions, explicitly address the exclusion of oscillatory states, and better differentiate their approach from conventional multi-compartment modeling frameworks.

**Questions:**

1. Convergence Mechanism: Since Nash equilibrium and the zero-sum constraint do not inherently ensure convergence to equilibrium, does the model include any dynamic process to guide the system toward equilibrium from any initial state? Would the authors consider adding time-based dynamics or stability conditions to ensure convergence?
2. Exclusion of Oscillatory and Chaotic Behavior: In traditional multi-compartment models, stability conditions can be defined to ensure convergence to equilibrium. However, the equilibrium condition in this paper does not seem to exclude stable oscillatory or chaotic behavior. Could the authors clarify if such behaviors are possible in their model? If so, how might these be addressed to ensure the system reaches a steady-state equilibrium?
3. Role of Cointegration: Cointegration is typically used to identify statistical relationships between non-stationary variables, assuming they share a long-term equilibrium path. Is cointegration in this model intended merely as a statistical tool for parameter estimation, or is it expected to actively drive the system toward equilibrium? Could the authors clarify how cointegration contributes to achieving or maintaining equilibrium in the system?
4. Distinction from Multi-Compartment Models: The ESE model shows strong similarities to multi-compartment models, which also use conservation principles to maintain balance. Beyond the conceptual framing of Nash equilibrium and payoff functions, is there a fundamental structural difference that distinguishes ESE from these existing models? Specifically, does ESE offer any new insights or capabilities that go beyond what is possible with traditional multi-compartment approaches?
5. Computational Complexity: The paper claims linear computational complexity but does not provide a formal proof. Could the authors offer a theoretical complexity analysis to substantiate this claim? Additionally, how does the model handle scenarios with extensive interactions or feedback loops between subsystems, which might increase computational demands?
6. Applicability to Non-Stationary or Oscillatory Systems: If the system exhibits non-stationary or oscillatory behavior, how reliable is the equilibrium index (EI) as a measure of stability? Would the model need adjustments to handle such cases, or does it implicitly assume stationarity and convergence to a single steady state?

---

> ### Author Response · Authors · 2024-11-20
> **Responses to four weaknesses - part 1/2**
>
> We are very grateful for your thorough review and insightful comments, especially your deep understanding of our work and the recognition of its novelty.  It would be wonderful if one day we could engage in an in-person conversation about this study.  Nevertheless, please see our explanation to the points you raised, which are not weaknesses of this study.
>
> > ***Weakness 1 - insufficient support for ... true equilibrium state ... dynamic framework  ...***
>
> Please allow us to clarify a critical point that we do NOT claim that the system will reach a true equilibrium state.  ESE does not require multi-system $\mathcal{MS}$ to reach a true equilibrium state.  Instead, we estimate what the equilibrium state would look like (Eq. 11, Eq. 12 and Algorithm 1).  Then predictions can be performed based on measuring how far the current state is to the estimated equilibrium state. Hence there is no need to have a mechanism to ensure that the system will progress towards true equilibrium, but to measure how likely a system would progress toward estimated equilibrium.  As shown by our extensive experimental results, this ESE approach is quite effective in prediction.
>
> As you rightly pointed out, “*there is no mathematical basis for assuming that the system will move toward equilibrium*”.  Whether $\mathcal{MS}$ is actually in an equilibrium state or not would not matter for the prediction process.  If $\mathcal{MS}$ is indeed in equilibrium, then the prediction outcome would be likely to be “no change” in these systems.  All of the systems stay in their status quo.  Accordingly, we have added a brief note in the main paper to emphasize that point and to avoid a similar interpretation of ESE. “***ESE does not require $\mathcal{MS}$ to reach equilibrium but estimates its equilibrium state.***” (Section 3, Line 209)
>
> Regarding “*a dynamic framework or time-dependent interactions*”, ESE is actually such a system, similar to the time-varying VAR model described in (Gao et. al 2024) (added at Lines 238-239).  As shown in Lines 286 – 289, including Eq. 12, where $t$ is the time factor.  The progression of $\mathcal{L}$ is time dependent, establishing the status of $t+1$ based on the status at time $t$.
>
> Another aspect of dynamic is the composition of $\mathcal{MS}$. In this study, we assume that will not change, meaning no system joining or leaving.  The challenge of dynamic composition, for example, a stock market with new IPOs and bad stocks being delisted, is being investigated in our extended work.
>
> It is great to see that *differential equation* was mentioned.  We have indeed investigated differential equations for obtaining the equilibrium state while developing ESE (see our citations at Lines 101-102, Bai et al. 2019, 2020, 2022).  We found this classic approach unsuitable, mainly due to two reasons.
> * Firstly, it is not suitable for a large number of systems. Our early-stage experiments show that when the number of systems exceeds $15$, the computational cost becomes prohibitively high. Because the cost to obtain inflection points that simultaneously satisfy all systems is exponential to the number of systems.
> * Secondly, when the input length is short, such as less than 20 time points, differential equations is less effective.  Our experiments show that ESE is much more accurate than differential equation approaches, even with a small number of systems.  If necessary, we can include that part of the study in the appendix.  Due to the deterministic nature of classic differential equations, they cannot effectively deal with stochastic variations in the data.  Introducing a stochastic mechanism in differential equations may help, but that is beyond the scope of this study.
>
> Worth mentioning that, we have also investigated other alternatives, e.g. non-linear regression methods and Markov models.  None of them is as effective as ESE.
>
> The two references mentioned above have been added to the paper (see full details below).
> * Jiti Gao, Bin Peng, and Yayi Yan. Estimation, inference, and empirical analysis for time-varying var models. Journal of Business & Economic Statistics, 42(1): 310–321, 2024
> * Wendell H Fleming and Raymond W Rishel. Deterministic and stochastic optimal control, volume 1. Springer Science & Business Media, 2012.
>
> > ***Weakness 2 - cointegration  ... how equilibrium is achieved dynamically.***
>
> As you rightly pointed out, cointegration is not to guide the progression toward equilibrium, but just a tool we use to verify whether the state we obtain is in long-run equilibrium.  If not, the training cycle will continue (Step 2 of Algorithm 1).  Otherwise, the training will stop as the existence of a long-run equilibrium is true.  To clarify that point, we have updated the main text, from Lines 306 to 310. Note, compared to short-run equilibrium, we prefer long-run equilibrium (see more below).

---

> ### Author Response · Authors · 2024-11-20
> **Responses to four weaknesses - part 2/2**
>
> > ***Weakness 3 - possibility of stable oscillatory behavior .... settle into a stable oscillation rather than converging to a true steady state.***
>
> That is a super insightful comment!! Thank you!  Please note that we do not aim to converge to a true equilibrium, but rather obtain a converged state of estimated equilibrium, so subsequent predictions can proceed. With ESE, oscillation behavior, even when it occurs in $\mathcal{MS}$, would not have an impact on the estimated equilibrium state, and subsequently on the predictions.
>
> Please see Steps 4 & 6 in Algorithm 1, the calculation there is to advance toward a state of long-run equilibrium.  The algorithm itself would not oscillate but converge from the state at the current time point. Even if $\mathcal{MS}$ itself exhibits oscillatory behavior, the calculation of Algorithm 1 is not affected, as it only concerns the current state (Morin 2008).
>
> To further illustrate the above, we have inserted a new section, **Section C**, in the appendix,  ***ESE Convergence Process***.  That shows the convergence from the current state toward a state of long-run equilibrium, for both synthetic data and COVID data.  Note, the existence of cointegration is determined by a null hypothesis with a p-value of 0.05.  An actual equilibrium training process will stop when the p-value reaches 0.05.  For illustration purposes, we allow the p-value to go lower than 0.05.  Nevertheless, we can see a clear trend of convergence.
>
> * David Morin. Introduction to classical mechanics: with problems and solutions. Cambridge University Press, 2008.
>
> > ***Weakness 4 - multi-compartment models, ... convergence conditions ...***
>
> Thank you for raising multi-compartment models.  Our understanding of multi-compartment models is quite different to the multi-system models used in this paper.  Each compartment aggregates targets of one particular category.  Different compartments contain targets of different types.  The three compartments in the study of Nakata et al. refer to stem cells, mature cells, and progenitor cells (Nakata et al. 2012).  In our study, all systems in $\mathcal{MS}$ are of the same type, having the same set of attributes, but different attribute values (See Definition 1), similar to multiple objects created from the same class in Object Oriented Programming.
>
> Another widely used multi-compartment models in Pharmacokinetics divides the body into different compartments to study the distribution of drugs in the body (Aldo et al. 1960).  Again, different compartments here are different in characteristics where systems in our $\mathcal{MS}$ are in the same type.
>
> Nevertheless, the number of compartments analyzed in multi-compartment studies is commonly two to three.  For such a small number, differential equations can be quite effective.  However when the number of compartments increases, e.g. beyond 10, multi-compartment would become very costly to compute by differential equations (see response to **Weakness 1**).
>
> Regarding the analysis of convergence, thank you for the suggestion.  We have added one more section, **Section C**, in the appendix,  ***ESE Convergence Process***, as discussed for **Weakness 3**.  We can observe a steady and consistent pattern of convergence toward estimated long-run equilibrium, in both synthetic data and COVID data.
>
> * Yukihiko Nakata, Philipp Getto, Anna Marciniak-Czochra, and Tom´as Alarc´on. Stability analysis of multi-compartment models for cell production systems. Journal of biological dynamics, 6(sup1):2–18, 2012.
>
> * Aldo Rescigno. Synthesis of a multi compartmented biological model. Biochimica et Biophysica Acta, 37(3): 463–468, 1960.

---

> ### Author Response · Authors · 2024-11-20
> **Responses to 6 questions**
>
> > ***Q1. Convergence Mechanism:***
>
> As discussed throughout the responses to Weaknesses, we do not require $\mathcal{MS}$ to converge to a true equilibrium.  ESE estimates the long-run equilibrium at a specific input length during training as shown in Algorithm 1.  The analysis of its convergence process has been added as **Section C** in the appendix.  As mentioned above, we have investigated other approaches like differential equations, and found ESE to be the most suitable for the multi-system prediction problem.
>
> > ***Q2. Exclusion of Oscillatory and Chaotic Behavior:***
>
> An excellent point! Oscillation and stochastic behaviours are important in multi-system prediction.  Because of them, conventional deterministic differential equations do not perform well in the scenarios that we studied.   In comparison, ESE can achieve good performance on par or better than SOTA methods, but much faster.  ESE does not require $\mathcal{MS}$ to converge to a steady-state equilibrium.  More details are in the response to **Weakness 3**.
>
> > ***Q3. Role of Cointegration:***
>
> Another very insightful question! Please see our response to **Weakness 2** and **Weakness 3**, plus the updated paper, especially **Section C** of the appendix.  Cointegration in ESE is just a statistical tool to test the existence of long-run equilibrium.
>
>
> > ***Q4. Distinction from Multi-Compartment Models:***
>
> Thank you for raising this point.  Please see our response to **Weakness 4**.
>
> > ***Q5. Computational Complexity:***
>
> Thank you for asking for further clarification on this key point.  We thought the equations of ESE could already show the linear complexity as they are linear if other parameters remain constant.
>
> To clearly prove the linear complexity of ESE, we uncommented out the detailed form of Eq. 11, and updated Section 6 in the main paper: “***As shown in Eq. 11, $\psi_j$ and $\lambda_{i,j}$ reflect the number of systems and attributes respectively. Also, as shown in Line 3 of Algorithm 1, the number of iterations is proportional to time step $t$. That means Algorithm 1 is of linear complexity. The computational cost is linear to the number of systems, the number of attributes and the time steps. That is consistent with the analysis using COVID data, shown in Fig. 5”***.
>
> > ***Q5.2 ... extensive interactions and feedback loops...:***
>
> See our response in a separate block below.
>
> > ***Q6. Applicability to Non-Stationary or Oscillatory Systems:***
>
> Thank you again for your deep understanding of our work.  Non-stationary or oscillatory systems can be often observed in the real world. Our COVID data does have some characteristics of that.  For example, the number of deaths varies significantly, while the number of daily new cases exhibits a certain degree of oscillation.  *ECM (Error Correction Model)* is a common strategy for dealing with these issues (Stock et al 2020).  However, on COVID-19 data, ECM is not effective when used with differential equations.  In comparison, ESE can perform well without ECM, showing its strong applicability in dealing with non-stationary or oscillatory behaviours.
>
> * Stock, James H and Watson, Mark W. Introduction to econometrics. Pearson. 2020.

---

> ### Author Response · Authors · 2024-11-22
> **Response to Q5.2 - extensive interactions and feedback loops**
>
> > ***how does the model handle scenarios with extensive interactions or feedback loops between subsystems, which might increase computational demands?***
>
> Thank you for yet another super insightful question! This was indeed a challenge that we faced at the early stage of our ESE study.  For this very reason, we introduced the transformation from Eq. 7 to Eq. 8, Lines 204 - 210, under the zero-sum assumption, so the interaction between systems can be simplified, and computational complexity reduced.  For the sake of readability, that part of our work is placed in the appendix, ***Appendix B***.
>
> Without the zero-sum assumption, attributes will be dependent to each other, extensive interaction and feedback loop then have to be considered in equilibrium calculation.  The complexity of the computation can be as high as $O(N^N)$, making it prohibitively costly when the number of systems go beyond 10.
>
> With the zero-sum simplification, we treat all attributes independent, interaction and feedback between systems are only considered for the same attribute.  In ESE, the total change of one attribute in $\mathcal{MS}$ is always 0, because of the use of proportions instead of absolute values.  Then the impact of interaction and feedback is further controlled.  As a result, ESE can be of linear complexity.
> ﻿
> In light of your comment, we have updated Appendix B to make it clearer.  The main text is also slightly updated at Lines 204-207, when introducing the simplification.
>
> Much appreciated.

---

### Official Review · Reviewer_WBf8 · 2024-11-03

**Soundness:** 1
**Presentation:** 1
**Contribution:** 1
**Rating:** 1
**Confidence:** 1

**Summary:**

I believe this paper either has a high chance of being generated by a language model, or was rushed into submission. In both cases, I recommend a rejection.

**Strengths:**

-

**Weaknesses:**

**I stopped reviewing this paper, as there were multiple issues. The objective and the problem setting are not clear, assumptions are not clearly stated, related work are missing, and writing is very poor.**

The definition of a "system" in the first place is not clear. The authors casually drop "properties" of this said system/multi-system, which vaguely correspond to some concepts, such as Nash equilibria, but not clear at all if those are assumptions/setting the authors consider.

The paper is almost impossible to read, with abuse of notation used absolutely for no reason. It is not really clear what problem the paper is dealing with and it is very poorly formulated throughout.

It is not clear how the COVID example fits into the Constraints 1 & 2. Why Eq. (4) needs to hold for the COVID case for instance?

Related work section mentions mostly datasets and equilibrium examples from different domains, but I could not spot any ML related work that consider a similar problem.

Why do you define ${\cal M} {\cal S}$ to be the sum of the target variables in systems, whereas it was first defined as the set of systems? The notation should be improved across the board, but this is just one example.

Why does Eq. (1) hold *in general*? Is that a setting you consider?

You cannot refer to Constraint 1 when defining Constraint 1 itself?

Line 158 - Please refer to where did you describe your ESE method as claimed.

What is the takeaway from Figure 1 exactly?

The connection to the Nash equilibrium is never formally introduced or motivated. It is not clear where  Lemma 1 comes from. There are no proper citations as well.

**Questions:**

-

**Details Of Ethics Concerns:**

need to be checked for LLM-generated content.

---

> ### Author Response · Authors · 2024-11-20
> **Please reconsider - part 1/2**
>
> > ***I believe ... being generated by a language model, or was rushed into submission...***
>
> We are quite concerned about what made you think this work has no strengths and is either generated by LLM or prepared in haste.  Both are serious academic integrity accusations that none of us can afford to bear. This manuscript is purely hand-typed and has been manually polished for several rounds.  Actually, it is a near-miss of another top venue similar to ICLR.  The feedback from there was purely technical, enhancing experiments and adding complexity analysis, which has been incorporated in this version.   We are willing to be scrutinized in whatever way it takes to prove innocent.
>
> > ***I stopped reviewing ... objective and the problem are not clear, assumptions are not clearly stated, ... very poor.***
>
> Very sorry and saddened to see this.  The objective and the problem are stated in the abstract and at the beginning of the introduction. Lines 031 – 034, “ ***This study establishes a new prediction method to address tasks involving multiple systems that may interact with each other. While predicting individually for each system is possible, we enable a holistic approach based on the concept of equilibrium, so all of the systems can be predicted at once without any repetition.***”  We thought what we wrote was clear, especially with the illustration in Fig 1.
>
> The objective is to perform one prediction when multiple predictions of the same type are needed.  For the sake of example, if we need to predict the sales of multiple shops, we can use conventional time series prediction methods to predict the sales for one shop and repeat the process for every shop.  In comparison, our aim is to establish a new method that can predict the sales of all the shops at once.
>
> The new method in the paper is not an incremental study but opens a new frontier.  There is no prior or similar work as we are aware of.  However, there are many related works, e.g. prediction methods, and equilibrium and game theory studies.  They have been included in the bibliography.
>
> Regarding writing, we would highly appreciate specific comments and suggestions on how our writing can be improved.  Below we address your specific comments one by one.
>
> > ***The definition of a "system" ... not clear. ...***
>
> We are quite puzzled by the concerns raised here.  The term “system” follows common usage, simply referring to a group of interrelated elements and can be described with a set of attributes, for example, a region or a company.  We can certainly add a formal common knowledge definition of system, $H(x(t)) = y(t)$. Prediction for a system is usually the future state of $y(t+i)$ where $t$ is the current time and $t+i$ is a future time point.
>
> The definition of multi-system in this study is at Line 123, Definition 1.  It is also illustrated in Figure 1, where $s_1,s_2,\dots,s_6$ are all individual systems.  Together they form a multi-system $\mathcal{MS}$.  Each system can be described in three attributes, Attributes 1 to 3.  Neither system nor multi-system is an assumption but a basic definition set in this study.
>
> > ***impossible to read ...***
>
> We acknowledge that there are many notations and equations in this paper, but certainly not “***used absolutely for no reason***”.   We have gone through extensive mathematic rigor to formulate the multi-system prediction problem as well as the methodology as precise as possible, as rigorous as possible.
>
> The problem that the paper is dealing is stated in the abstract and at the beginning of the introduction. Lines 031 – 034, “ ***This study establishes a new prediction method to address tasks involving multiple systems that may interact with each other. While predicting individually for each system is possible, we enable a holistic approach based on the concept of equilibrium, so all of the systems can be predicted at once without any repetition.***” As shown in the case study, to predict daily new cases of each region, we could use conventional methods to predict one region at a time and repeat the process for every region.  What we offered here is a method that can predict all regions in one run, hence no need to repeat.  It is not just more convenient, but also much more efficient and at least equally accurate, if not more accurate, than SOTA prediction methods.
>
> > ***not clear how the COVID example fits ...***
>
> Constraints 1 and 2 are constraints of zero-sum games in Nash equilibrium.  It is the basis of applying the equilibrium concept for prediction.  In the case of COVID, that simply means there is no region will be added or removed during the prediction.  The total number of regions remains the same.  Constraint 2 on COVID says the total change of all regions is treated as zero so we can apply our equilibrium state evaluation, e.g. ESE method. The total proportion is always retreated as 1 hence the total change is always zero.  So COVID case can be applied with our ESE method.

---

> ### Author Response · Authors · 2024-11-20
> **- part 2/2**
>
> > ***Why do you define MS to be the sum...***
>
> For simplicity reasons, we did not introduce another symbol to denote the sum, but just used $\mathcal{MS}$, as explained in Line 128 and Eq. 2.  The sum can reflect the overall status of the multi-system, so the symbol $\mathcal{MS}$ is used interchangeably.  We have updated the paper to emphasize the connection.
>
> > ***Why does Eq. (1) hold in general?***
>
> Equation 1 is the fundamental concept of zero-sum game in Game Theory. The equation simply says the changes of one participant are derived from the sum of the other participants. That is the basis, or basic assumption, of our multi-system prediction, the change in one system is the result of changes from other systems. For readers’ benefit, we have added the classic Von Neumann’s *“Theory of games and economic behavior, the 60th anniversary commemorative edition”* in the citation.
>
> > ***cannot refer to Constraint 1 when defining Constraint 1 itself?***
>
> That is seemingly a confusion.  “*If Constraint 1 is satisfied,*” at Line 152 is not a part of the definition of Constraint 1, but to describe what happens mathematically if the constraint is satisfied, e.g. Eq. 3.
>
> >  ***Line 158 - Please refer to where did you describe your ESE method as claimed.***
>
> That is mainly referring to Section 1, in particular the second paragraph, where the basic idea and the key principle of ESE is briefly described.  For the reader’s benefit, we have updated that to “***As described in the introduction section, our ....*** ”  at Line 174.
>
> > ***the takeaway from Figure 1 exactly?***
>
> Fig. 1 is to illustrate multi-system prediction visually, aiming to help readers establish a basic understanding of the problem that we are working on. In the example $\mathcal{MS}$, there are at least 6 systems, $s_1$ to $s_6$.  Every system is associated with the same set of three attributes, Attributes 1 to 3. The attribute set of system $s_4$ is denoted as $\mathcal{A}_4$.  The task is to predict the future state of all systems in $\mathcal{MS}$ simultaneously.  For the reader’s benefit, we have added the above text in the caption of Fig. 1.
>
> > ***connection to the Nash equilibrium is never formally introduced or motivated. It is not clear where Lemma 1 comes from. There are no proper citations as well.***
>
> Lemma 1 is a variation of the basic payoff function in Nash Equilibrium, which we assumed was a common concept in Game Theory and zero-sum games.  For readers’ benefit, we have also cited Von Neumann’s “*Theory of games and economic behavior*” for these fundamentals.  Here, based on the concept of zero-sum games, we can have Eq 6, which can be extended to Eq. 7 for $\mathcal{MS}$.  Then we can deduce from Eq. 7 into Eq. 8. The full deductive reasoning process is provided in Appendix B, from Eq. 17 to Eq. 20.

---

### Author Response · Authors · 2024-11-21
**Paper updated according to the comments**

Dear Reviewers, thank you for your time and effort, especially for these very insightful comments regarding convergence, cointegration and oscillation.

***Please examine our responses and the updated paper, and reconsider your view on this study***.  Your concerns have been carefully addressed, with detailed clarifications and further explanations in reply to your comments.  The corresponding changes in the updated PDF file are ***highlighted in red***.

Please note, this study is NOT incremental research, but aims to open a new paradigm for prediction, leveraging game theory and equilibrium which are not typical predictive models.  There is no direct prior work that we are aware of. Hence it may appear to lack references in ML.  However, this study does have many related works in game theory, econometrics and applied math (see our long list of bibliography).

ESE's novelty and significance first lie in the new mathematical formulation of ESE, which turns equilibrium estimation of zero-sum games into predictive modelling.  As a result, there is no need to repeat the prediction for multiple systems of the same type, if they can be viewed together as a ``**multi-system**''.

Note this approach does not work for every kind of time series prediction, as discussed at the end of the main paper.  Because most time series tasks/benchmarks cannot be viewed as a zero-sum game, so equilibrium does not apply.  Also, they do not have attribute data other than the target variables, making them unsuitable for estimating the equilibrium states, even if equilibrium can apply.  However the target problem studied in this paper, ``**multi-system prediction**'' can often be observed in the real world, for example, the spreading of viruses like COVID over multiple regions, the business activities of multiple companies under one market, and the economic development of multiple countries of one continent.  Hence the impact and the value of this research are foreseeable and significant.

ESE's novelty and significance can be verified by its effectiveness, demonstrated through extensive experiments on a wide range of scenarios, presented in 17 tables (Tables 1-9, 12-19).  ESE can be at least comparable in prediction accuracy, in comparison with 12 SOTA time series prediction models.  The highlight is that ESE can be combined with these models to achieve the best accuracy in most cases.

Another key advantage of ESE is its **simplicity, high speed and low complexity**.  It does not need to be repeated, no matter if there are 79 systems or 320 systems or more.  The complexity is as low as linear.  The prediction for all systems can be obtained in one run.  ***The speed increase can be as large as 70+ times (bold in Table 3).***

In addition, we have studied other methods before ESE, e.g. differential equations, non-linear regression and Markov models, on the targeted multi-system prediction task.  None is suitable or as effective and efficient as ESE.  We would love to hear suggestions on alternative methods, that may be better than ESE for multisystem prediction tasks.  Our data are openly accessible.  We welcome separate studies.

In short, we view this work as a novel and significant advancement in predictive modelling, as well as an expansion of game theory applications, deserving of decent recognition in the field.

Thank you in advance
(**Authors of 12376**)

---

### Author Response · Authors · 2024-11-25
**Please comment on our responses as early as possible.  Thank you in advance.**

Dear Reviewers, could you please comment on our responses at your earliest convenience?  Our responses have been uploaded since last Thursday, 6 days ago.  We are eager to address any further questions or concerns before the discussion period deadline, which is fast approaching.

Looking forward to hearing from you.

Thank you in advance (**Authors of 12376**)

---

> ### Author Response · Authors · 2024-11-29
> **Please respond as soon as possible**
>
> Dear Reviewers, could you please respond? We are keen to discuss the rebuttals before the extended deadline.
>
> Waiting to hear from you.
>
> Thank you in advance (**Authors of 12376**)

---

### Meta-Review · Area_Chair_3gcv · 2024-12-21

**Metareview:**

The authors propose a Equilibrium State Evaluation (ESE) to model interconnected subsystems. The model uses econometrics techniques to model long term equilibrium trends. Reviewers have noted several methodological and foundational issues particularly in terms of assumptions made of existence of such an equilibrium and the ability of the method to achieve the equilibrium. Considering also the presentation issues, the paper would require significant revisions and another round of peer review before being ready for publication. As such despite the authors partially addressing the concerns raised, I am recommending a reject.

Please note that one of the reviewers made the claim that the paper is generated by a generative AI. I have not assumed it as such while going over the paper and reviewer comments.

**Additional Comments On Reviewer Discussion:**

No additional discussions in the discussion period.

---

### Decision · Program_Chairs · 2025-01-22

Reject